# Specialized astrocytes mediate glutamatergic gliotransmission in the CNS

Roberta de Ceglia[1], Ada Ledonne[1,2,10], David Gregory Litvin[1,3,10], Barbara Lykke Lind[1,4], Giovanni Carriero[1], Emanuele Claudio Latagliata[2], Erika Bindocci[1], Maria Amalia Di Castro[5], Iaroslav Savtchouk[1,9], Ilaria Vitali[1], Anurag Ranjak[1], Mauro Congiu[1], Tara Canonica[1], William Wisden[6], Kenneth Harris[7], Manuel Mameli[1], Nicola Mercuri[2,8], Ludovic Telley[1✉] & Andrea Volterra[1,3✉]

Multimodal astrocyte–neuron communications govern brain circuitry assembly and function[1]. For example, through rapid glutamate release, astrocytes can control excitability, plasticity and synchronous activity[2,3] of synaptic networks, while also contributing to their dysregulation in neuropsychiatric conditions[4–7]. For astrocytes to communicate through fast focal glutamate release, they should possess an apparatus for $Ca^{2+}$-dependent exocytosis similar to neurons[8–10]. However, the existence of this mechanism has been questioned[11–13] owing to inconsistent data[14–17] and a lack of direct supporting evidence. Here we revisited the astrocyte glutamate exocytosis hypothesis by considering the emerging molecular heterogeneity of astrocytes[18–21] and using molecular, bioinformatic and imaging approaches, together with cell-specific genetic tools that interfere with glutamate exocytosis in vivo. By analysing existing single-cell RNA-sequencing databases and our patch-seq data, we identified nine molecularly distinct clusters of hippocampal astrocytes, among which we found a notable subpopulation that selectively expressed synaptic-like glutamate-release machinery and localized to discrete hippocampal sites. Using GluSnFR-based glutamate imaging[22] in situ and in vivo, we identified a corresponding astrocyte subgroup that responds reliably to astrocyte-selective stimulations with subsecond glutamate release events at spatially precise hotspots, which were suppressed by astrocyte-targeted deletion of vesicular glutamate transporter 1 (VGLUT1). Furthermore, deletion of this transporter or its isoform VGLUT2 revealed specific contributions of glutamatergic astrocytes in cortico-hippocampal and nigrostriatal circuits during normal behaviour and pathological processes. By uncovering this atypical subpopulation of specialized astrocytes in the adult brain, we provide insights into the complex roles of astrocytes in central nervous system (CNS) physiology and diseases, and identify a potential therapeutic target.

To begin re-examining the astrocyte glutamate exocytosis hypothesis, we first turned to single-cell transcriptomic analysis. We integrated eight diverse single-cell RNA-sequencing (scRNA-seq) and single-nucleus RNA-seq databases from mouse brain, each containing hippocampal cells, including high-quality individual astrocytes[19,23–29] (Fig. 1a, Methods and Extended Data Fig. 1a,b). Analysis of cellular transcriptional identities (Methods) revealed the presence of 15 clusters, which corresponded to the main hippocampal cell types (Extended Data Fig. 1c). To specifically annotate each individual cluster, we initially trained a deep neural network classifier (Methods) using a reference hippocampal database[29] (Extended Data Fig. 1d,e). The model performance was confirmed by applying cross-validation to each dataset (Methods and Extended Data Fig. 1f). The predicted clusters encompassed different populations of glutamatergic and GABAergic neurons, as well as several types of non-neuronal cells, including clusters identified as astrocytes (Fig. 1a and Extended Data Fig. 1d). We confirmed the correct prediction of astrocytes by checking the uniform manifold approximation and projection (UMAP) distribution of several known astrocyte markers, including *Slc1a2* (encoding GLT1), *Slc1a3* (encoding GLAST), *Gja1* (encoding CX43) and *Aqp4*, and confirmed

[1]Department of Fundamental Neuroscience, University of Lausanne, Lausanne, Switzerland. [2]Department of Experimental Neuroscience, IRCCS Santa Lucia Foundation, Rome, Italy. [3]Wyss Center for Bio and Neuro Engineering, Campus Biotech, Geneva, Switzerland. [4]Department of Neuroscience, Faculty of Health and Medical Sciences, University of Copenhagen, Copenhagen, Denmark. [5]Department of Physiology and Pharmacology, Sapienza University, Rome, Italy. [6]Department of Life Sciences and UK Dementia Research Institute, Imperial College London, London, UK. [7]UCL Queen Square Institute of Neurology, University College London, London, UK. [8]Department of Systems Medicine, University of Rome "Tor Vergata", Rome, Italy. [9]Present address: Department of Biomedical Sciences, Marquette University, Milwaukee, WI, USA. [10]These authors contributed equally: Ada Ledonne, David Gregory Litvin. ✉e-mail: ludovic.telley@unil.ch; andrea.volterra@unil.ch

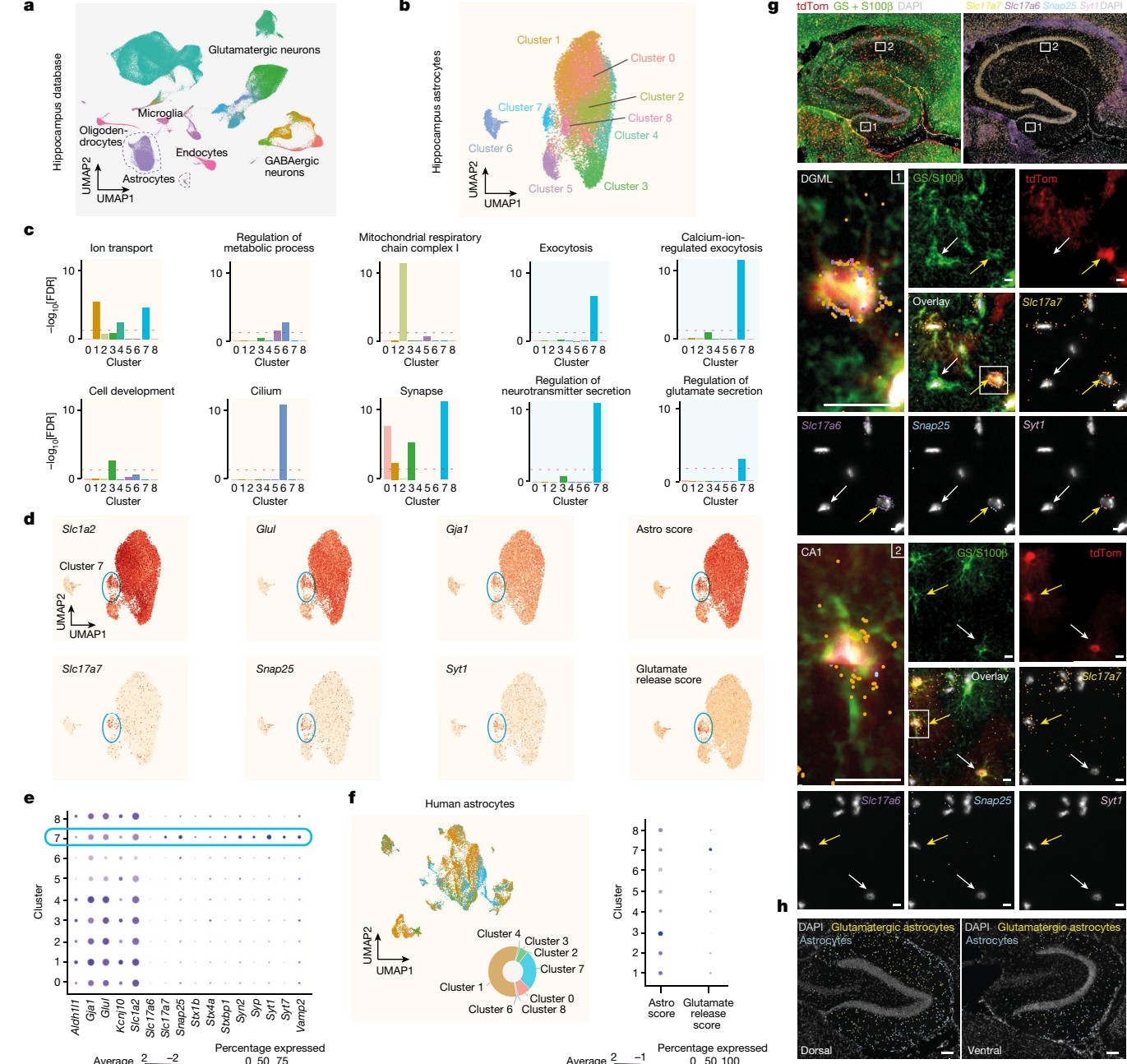

**Fig. 1 | scRNA-seq and RNAscope HiPlex identification of a subpopulation of glutamatergic astrocytes in the mouse and human hippocampus.**
**a**, UMAP representation of eight integrated hippocampus scRNA-seq datasets annotated using a neural network classifier trained on a comprehensive database[29]. **b**, Cluster analysis of the subset astrocyte population revealing nine transcriptionally distinct clusters. **c**, GO analysis of differentially expressed genes highlighting specific term enrichments for each cluster. The red dashed line shows the threshold (1, $-\log_{10}$-transformed) for significant enrichments. **d**, The expression level for canonical astrocytic markers and their respective combinatorial astro score, notably in cluster 7 (top). Bottom, the expression level for glutamate exocytosis markers and the glutamate-release score, notably in cluster 7. **e**, Expression of selected marker genes related to astrocytic identity, vesicular trafficking and glutamate-regulated exocytosis for each predicted astrocyte cluster. **f**, UMAP analysis of integrated human hippocampus scRNA-seq data, classified using our integrated astrocyte database as a reference (left). The pie chart shows the distribution across predicted clusters. Right, dot plot of canonical astrocytic or glutamate exocytosis combinatorial

score for predicted astrocyte clusters. **g**,**h**, RNAscope HiPlex assay combined with immunohistochemistry. $n = 12$ slices, 2 mice. **g**, Low-magnification dorsal hippocampus slice from mice expressing tdTomato under the *GFAP* promoter (red; Methods) showing immunohistochemistry staining for combined GS and S100β (green), and DAPI (white) (top left). Top right, in the same slice, HiPlex analysis of *Slc17a7* (yellow), *Slc17a6* (violet), *Snap25* (blue) and *Syt1* (pink). Middle, magnified images of the DGML (indicated by the white rectangle 1 in the top images), showing expression of all of the astrocytic markers and glutamate exocytosis markers listed in the top images. A glutamatergic astrocyte (yellow arrow) and a non-glutamatergic astrocyte (white arrow) are indicated. Inset (left): magnified view of the glutamatergic astrocyte. Bottom, as described for the middle images, but from the CA1 stratum radiatum region (CA1, white rectangle 2 in the top images). Scale bars, 10 μm. **h**, The proportion of glutamatergic (segmented in yellow) versus non-glutamatergic (azure) astrocytes along the dorsal–ventral axis of the hippocampus. Glutamatergic astrocytes are more abundant in a dorsal slice (left) compared with in a ventral slice (right). Scale bars, 100 μm.

their selective expression in clusters identified as astrocytes (Extended Data Fig. 1g and Supplementary Table 1). An analysis of differentially expressed genes identified genes that are enriched in astrocytes compared with in all of the other hippocampal cells, providing 'pan markers' for hippocampal astrocytes.

## Astrocytes with a glutamatergic signature

We next performed dimensionality reduction and graph-based clustering analysis of the extracted astrocyte population, identifying nine astrocytic clusters (Fig. 1b). These clusters were uniformly represented within each dataset, which indicated robust integration (Extended Data Fig. 2a,b). An analysis of differentially expressed genes among clusters pinpointed several significant type-enriched transcripts (Extended Data Fig. 2c and Supplementary Table 2). Using gene set enrichment analysis (GSEA) databases (Methods), we found significant Gene Ontology (GO) enrichment for each cluster (Extended Data Fig. 3), except for cluster 8, which showed no clear transcriptional signature. On the basis of available biological information and specific GO term analysis of the divergent genes, we noticed the emergence of discrete core biological functions (Fig. 1c), consistent with astrocytic types described in other brain regions[20]. For example, GO terms related to ion transport were significantly enriched in clusters 1, 4 and 7, whereas GO terms related to metabolic processes were found in clusters 5 and 6 (Fig. 1c and Extended Data Fig. 3). Cluster 2 GO terms were highly enriched in genes related to mitochondrial function, whereas cluster 3 showed specific enrichment for genes involved in cell development. Cluster 6 exhibited characteristics of neural precursors, including cell cycle gene expression, and typical ependymal cell cilium function, reminiscent of neurogenic niche cells in the hippocampal dentate gyrus (DG; Fig. 1c and Extended Data Fig. 2h). However, cluster 6 also showed strong expression of *Ifitm3* and *Vim* (Extended Data Fig. 2i), which were linked to a specific, immune-related, astrocytic cluster[30]. Notably, we identified clusters with GO enriched terms related to synapse (clusters 0, 1, 3 and 7), with cluster 7 exhibiting top divergent genes specifically related to exocytosis, calcium-ion-regulated exocytosis, regulation of neurotransmitter secretion and regulation of glutamate secretion (Fig. 1c). In cluster 7 we found strong enrichment of essential transcripts for $Ca^{2+}$-regulated synaptic glutamate exocytosis such as *Slc17a7* (encoding VGLUT1), *Snap25* and *Syt1*, which was confirmed by their related combinatorial score, the 'glutamate release score' (Fig. 1d). We then confirmed that pan-astrocytic markers, including *Slc1a2* (encoding GLT1), *Glul* (encoding glutamine synthetase (GS)) and *Gja1* (encoding CX43), and their combinatorial score, the 'astro score', were uniformly labelled in this cluster (Fig. 1d), along with several other typical astrocytic genes (Fig. 1e and Extended Data Fig. 2d,e). Moreover, among the most significantly enriched transcripts, cluster 7 exhibited several other genes related to vesicular transport, regulated exocytosis and synaptic functions (Fig. 1e and Extended Data Fig. 2f,g). We ruled out the neuronal origin of these transcripts due to synaptic engulfment, as this would require enrichment of phagocytosis-associated transcripts such as *Megf10* and *Merkt*[31], which was not observed in cluster 7 (Extended Data Fig. 2i). Moreover, cluster 7 was unambiguously detected in a database obtained through single-nucleus sorting[23], which excludes the enrichment of cytoplasmic mRNA. Overall, we found this synaptic glutamate exocytosis cluster in all mouse hippocampal databases (Extended Data Fig. 2b). Notably, cross-species investigation by label transfer of three human hippocampal cell databases[32–34] revealed the presence of this cluster also in human (Fig. 1f).

To confirm the presence of typical neuronal synaptic transcripts in cluster 7 astrocytes, we performed multiplex fluorescence in situ hybridization (RNAscope HiPlex assay) analysis of hippocampal slices from adult mice conditionally expressing red tdTomato reporter in astrocytes, and co-immunostained for two additional astrocytic markers, GS and S100β (Fig. 1g). We targeted four typical neuronal

genes involved in glutamatergic vesicular exocytosis (*Slc17a7*, *Slc17a6* (encoding VGLUT2), *Snap25* and *Syt1*) and found that they were strongly expressed not only in glutamatergic neurons[35] (Fig. 1g (top)) but also in a subset of GS/S100β-positive cells that belonged to the GFAP lineage (tdTomato⁺) and had visibly isolated nuclei, excluding any overlap with neurons. This confirmed the presence of an astrocytic synaptic glutamate exocytosis population (Fig. 1g (middle, bottom)). We observed the expression of the synaptic glutamatergic markers in several isolated astrocytes located in various regions of the hippocampus, including the CA1, CA2, CA3 (both stratum oriens and stratum radiatum) and the DG (both molecular layer and hilus; Fig. 1g and Extended Data Fig. 2j). Notably, the density of this population was differentially distributed along the dorsal–ventral axis with, for example, the dorsal molecular layer of the DG (DGML) displaying a significantly higher proportion compared with the ventral region (Fig. 1h).

Overall, the experimental data demonstrate a hippocampal subpopulation of cells with morphological, immunohistochemical and transcriptional features typical of astrocytes that contain transcripts required for glutamatergic regulated secretion. Accordingly, we refer to these cells as glutamatergic astrocytes.

## Imaging glutamate-secreting astrocytes

We next sought to complement the molecular evidence for glutamatergic astrocytes with direct observations of astrocytic glutamate exocytosis in situ. We performed two-photon imaging studies in the dorsal DGML, a region that is predicted to contain significant proportions of glutamate-secreting astrocytes active in synaptic modulatory functions[8–10,36]. We used astrocyte-specific expression of the glutamate sensor, superfolder GFP iGluSnFR (SF-iGluSnFR)[22], to visualize release events from individual DGML astrocytes (Fig. 2a (left) and 2b). To mimic $Ca^{2+}$-dependent glutamatergic gliotransmission evoked by native $G_q$ G-protein-coupled receptors ($G_q$-GPCRs)[2], we co-expressed in astrocytes a designer receptor exclusively activated by designer drugs ($G_q$-DREADD; Fig. 2a (left) and 2c), and used chemogenetic stimulation by the designer drug clozapine *N*-oxide (CNO). To minimize any potential source of neuronal glutamate release, we perfused hippocampal slices with a synaptic blocker mixture containing, among others, tetrodotoxin and voltage-gated $Ca^{2+}$ channel blockers (Methods). In light of previous contrasting results in astrocytes[3,17], we adopted robust experimental and analytical protocols (Methods and Extended Data Fig. 5a–h). We applied CNO locally through short puffs, repeated its application six times during each experiment and, afterwards, applied L-glutamate (L-Glut) as a positive control (Fig. 2a (middle and right)). Of 24 tested astrocytes, all responded to L-Glut (Fig. 2c and Extended Data Fig. 6a), but only nine showed reliable subsecond SF-iGluSnFR fluorescence responses to CNO (Fig. 2c–e,p, Methods, Extended Data Fig. 6a,b and Supplementary Video 1). Subsequent control experiments enabled us to exclude such responses as artefacts (Extended Data Fig. 5k) or independent of $G_q$-DREADD signalling (Extended Data Fig. 5l). Notably, the CNO-evoked glutamate responses (Fig. 2c,d) occupied a small fraction of the field of view (FOV) responding to L-Glut (Fig. 2c and Extended Data Fig. 4a,f), and consistently localized to groupings, probably representing hotspots of glutamate release (Fig. 2d,q). The other 15 tested cells showed small or null SF-iGluSnFR signals in response to CNO (Extended Data Figs. 4g–i and 6a), and were classified as non-responders (Methods). In mice co-expressing astrocyte-specific $G_q$-DREADD with the $Ca^{2+}$ indicator GCaMP6f[37], CNO evoked $Ca^{2+}$ responses in all tested cells (Extended Data Fig. 5i,j). This implies that $G_q$-DREADD stimulation evokes $Ca^{2+}$ signalling in all astrocytes, but only a subgroup has the appropriate downstream machinery for secreting glutamate.

To determine whether astrocyte release occurred through exocytosis, we sought to impede glutamate filling in vesicles. We took advantage of a recently developed *Slc17a7*$^{fl/fl}$ mouse line[38] carrying

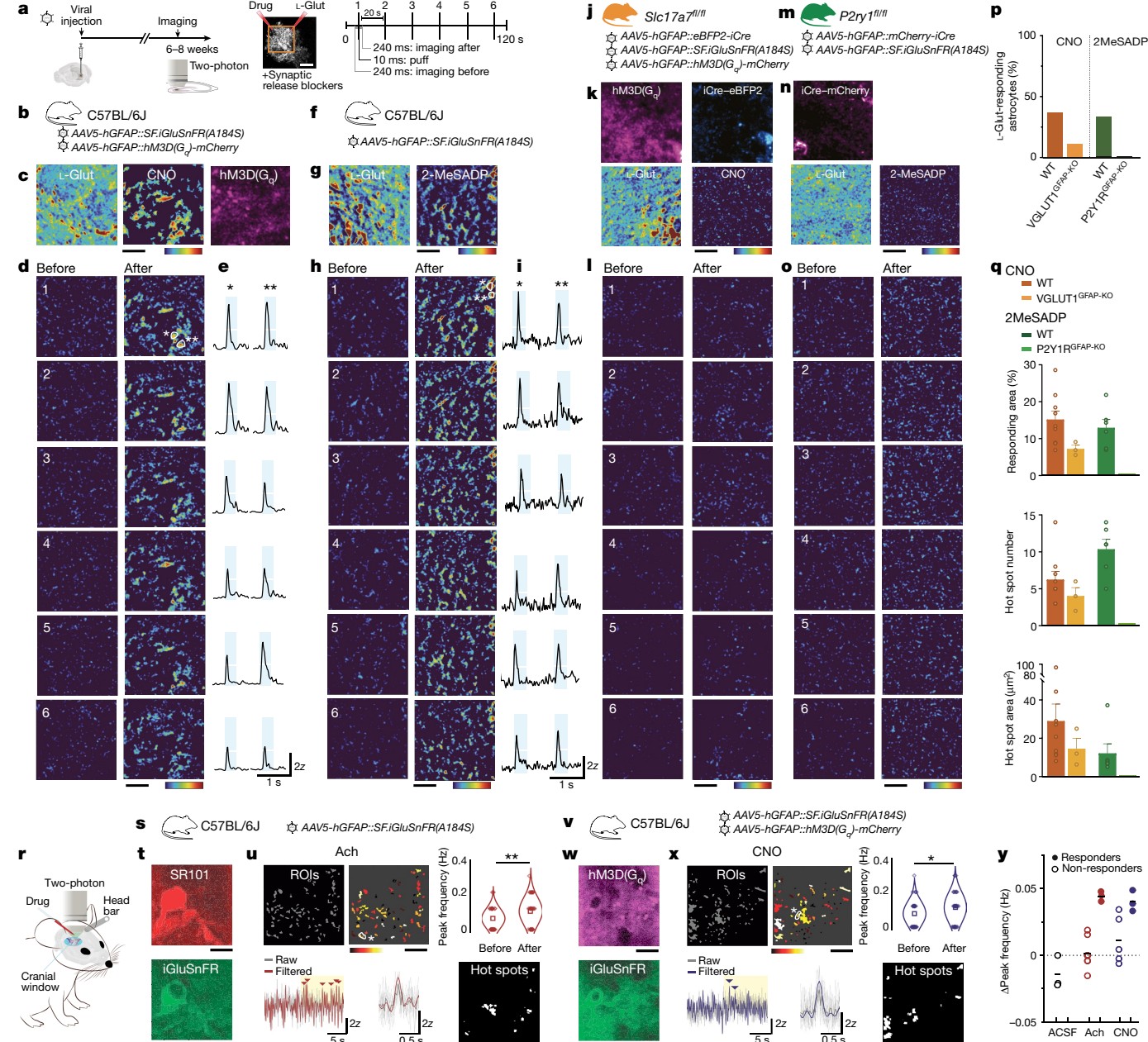

**Fig. 2 | See next page for caption.**

a conditional allele of *Slc17a7*, encoding VGLUT1, the main VGLUT isoform in hippocampus[35], expressed also in glutamatergic astrocytes (Fig. 1e,g; Extended Data Figs. 2f,g and 8f and Supplementary Table 2). To delete VGLUT1 selectively in astrocytes, we injected into the DG a viral construct expressing iCre recombinase and a fluorescent reporter under the human *GFAP* promoter, which enabled us to visualize cells undergoing recombination (Extended Data Fig. 4c). We confirmed that these cells were astrocytes, and that whole-brain samples and individually collected fluorescent astrocytes displayed recombined *Slc17a7*[fl/fl] loci, indicating VGLUT1 deletion (VGLUT1[GFAP-KO]; Extended Data Fig. 4c). To determine the effect of VGLUT1[GFAP-KO] on $G_q$-DREADD-evoked glutamate release from astrocytes, we co-infected DGML astrocytes of *Slc17a7*[fl/fl] mice with a viral vector combination expressing iCre, SF-iGluSnFR and $G_q$-DREADD (Fig. 2j). Out of 23 tested VGLUT1[GFAP-KO] astrocytes, all showing L-Glut responses comparable to wild-type (WT) astrocytes (Fig. 2k and Extended Data Figs. 4d,f and 6c,d), 20 did not respond to CNO (Fig. 2k,l and Extended Data Fig. 6c,d)

and 3 showed SF-iGluSnFR responses smaller than in WT astrocytes (Fig. 2q and Extended Data Figs. 4m–o and 6c,d). Given that injection of the same viral cocktail in WT mice did not change the proportion of responding astrocytes (Methods), the substantial reduction observed in *Slc17a7*[fl/fl] mice (Fig. 2p,k) appears to be specific to VGLUT1[GFAP-KO], consistent with VGLUT1-dependent exocytosis being the main release mechanism of glutamatergic astrocytes. The few CNO responders in the VGLUT1[GFAP-KO] group may be *Slc17a6*-expressing cells[8] (Fig. 1g and Extended Data Fig. 8f).

To demonstrate that glutamate release in a subgroup of DGML astrocytes occurs through a natural signalling mechanism, we moved from artificial stimulations to endogenous $G_q$-GPCR stimulations. We exposed hippocampal slices from WT mice expressing SF-iGluSnFR (Fig. 2f) to the same stimulation protocol used with CNO, applying instead 2-methylthio-adenosine-5′-diphosphate (2MeSADP), an agonist of purinergic P2Y1 receptors. 2MeSADP was previously shown to elicit $Ca^{2+}$ signalling in DGML astrocytes followed by modulatory

**Fig. 2 | Fast glutamate secretion at hotspots in a subgroup of astrocytes after selective chemogenetic or endogenous receptor stimulation in situ and in vivo. a**, Schematic of two-photon SF-iGluSnFR glutamate imaging experiments in hippocampal slices from virally injected WT or transgenic mice (details are provided in **b**,**f**,**j** and **m**) (left). Middle, typical FOV imaged from a DGML astrocyte. Drugs (CNO, 100 µM; 2MeSADP, 10 µM) and L-Glut (1 mM), all in Alexa-594 solution, were locally delivered through two puff pipettes. The slices were incubated with a cocktail of synaptic blockers (Methods). Right, the stimulation protocol used for drug applications. Ten-millisecond puff applications were performed six times, one every 20 s, during 120 s imaging acquisitions. 'Before' and 'after' correspond to the 240 ms imaging periods before and after each drug application shown in **d**,**h**,**l** and **o** as individual mean projections of the SF-iGluSnFR signal. Corresponding L-Glut-evoked responses are shown in Extended Data Fig. 4a,b,d,e. The whole-brain image is from the Allen Mouse Brain Connectivity Atlas (https://mouse.brain-map.org/). **b**–**e**, SF-iGluSnFR responses to chemogenetic stimulations in a representative astrocyte. **b**, Experiments in WT mice expressing SF-iGluSnFR and G$_q$-DREADD (hM3D(G$_q$)) in DGML astrocytes. **c**, Mean projection of G$_q$-DREADD–mCherry expression (right). Middle and left, s.d. projection of SF-iGluSnFR signal variance across 6 CNO (middle) or L-Glut (left) applications (high-variance spots represent repeatedly responding regions, that is, hotspots). **d**, Individual responses to six CNO applications. **e**, Traces corresponding to two hotspot regions in **d** (indicated by asterisks; white line, 2$z$; azure, 240 ms post-puff period). **f**–**i**, SF-iGluSnFR responses to endogenous P2Y1R stimulations in a representative astrocyte. **f**, Experiments in WT mice expressing SF-iGluSnFR in astrocytes. **g**, s.d. projection of SF-iGluSnFR signal variance across six applications of the P2Y1R agonist 2MeSADP (right) or L-Glut (left). **h**, Individual responses to six 2MeSADP applications. **i**, Traces corresponding to two hot spot regions. Details are as described in **e**. **j**–**l**, Lack of SF-iGluSnFR responses to CNO in a representative astrocyte with deleted VGLUT1 (VGLUT1$^{GFAP-KO}$). **j**, *Slc17a7$^{fl/fl}$* mice were injected with viral vectors inducing SF-iGluSnFR and G$_q$-DREADD expression, and iCre-mediated VGLUT1 deletion in triple-fluorescent astrocytes. **k**, Mean projections of G$_q$-DREADD–mCherry (top left) and nuclear iCre–eBFP2 (top right) expression, and s.d. projections of SF-iGluSnFR signal variance across six CNO (bottom right) or L-Glut (bottom left) applications. **l**, Individual responses to six CNO applications. **m**–**o**, A lack of SF-iGluSnFR responses to 2MeSADP in a representative astrocyte with deleted *P2y1r* (P2Y1R$^{GFAP-KO}$). **m**, *Glast$^{creERT2}$P2ry1R$^{fl/fl}$* mice were injected with viruses to express SF-iGluSnFR and induce iCre-mediated *P2y1r* deletion in astrocytes. **n**, Mean projection of iCre–mCherry expression (top) and s.d. projection of SF-iGluSnFR signal variance across six 2MeSADP (bottom right)

and L-Glut (bottom left) applications. **o**, Individual responses to 2MeSADP applications. For **c**,**d**,**g**,**h**,**k**,**l**,**n** and **o**, the z-score scale is colour-coded from 0 (dark blue) to 6 (red). **p**,**q**, Quantitative analysis of SF-iGluSnFR responses to drugs in DGML astrocytes. **p**, The proportion of astrocytes responding to (1) CNO in WT (23 cells, 5 mice) and VGLUT1$^{GFAP-KO}$ (24 cells, 5 mice) mice; and (2) 2MeSADP in WT (18 cells, 2 mice) and P2Y1R$^{GFAP-KO}$ (20 cells, 2 mice) mice. All individual cell responses are shown in Extended Data Fig. 6. **q**, Features of SF-iGluSnFR responses evoked by CNO (WT, $n$ = 9 out of 24 cells; VGLUT1$^{GFAP-KO}$, $n$ = 3 out of 24 cells) and 2MeSADP (WT, $n$ = 6 out of 18 cells; P2Y1R$^{GLAST-KO}$, $n$ = 0 out of 20 cells). Top, the percentage of L-Glut-responding FOVs that respond to CNO or to 2MeSADP (the same mouse groups as in **p**). The number (middle) and area (bottom) of individual hotspots per FOV for CNO and 2MeSADP are shown. **r**, Schematic of in vivo two-photon SF-iGluSnFR glutamate imaging experiments in the visual cortex of awake mice in the presence of synaptic blockers (details are provided in **s** and **v** and the Methods). **s**–**u**, SF-iGluSnFR responses to Ach in a representative astrocyte (110 µm below the surface). **s**, Experiments in WT mice injected with virus to express SF-iGluSnFR in visual cortex astrocytes. **t**, The red SR-101 signal highlights the astrocyte in the FOV (top). Bottom, cumulative SF-iGluSnFR fluorescence throughout the acquisition from the same astrocyte ($n$ = 8 cells, 3 mice). **u**, 50 selected ROIs (top left) (Methods), the peak frequency variations of SF-iGluSnFR signal in individual ROIs (colour scale: white (+0.25 Hz) to black (−0.1 Hz)) (top middle) and the mean frequency change in the 50 ROIs (top right) after Ach (10–50 mM) application (Wilcoxon rank-sum test, \*\*$P$ = 0.0059). Bottom left, SF-iGluSnFR traces from a representative ROI (asterisk in the top middle image), before and after (yellow) the Ach puff; the arrowheads indicate SF-iGluSnFR activity peaks. Bottom middle, the averaged kinetics of SF-iGluSnFR events from the bottom left plot, aligned to peak time. Bottom right, hotspot ROIs responding to two Ach applications. **v**–**x**, SF-iGluSnFR responses to chemogenetic stimulation in a representative astrocyte (137 µm below the surface). **v**, Experiments in mice expressing SF-iGluSnFR and G$_q$-DREADD in visual cortex astrocytes. **w**, Mean projection of G$_q$-DREADD–mCherry expression (top). Bottom, cumulative SF-iGluSnFR fluorescence throughout the acquisition from the same astrocyte as in **t** ($n$ = 11 cells, 3 mice). **x**, As described in **u**, but for CNO (0.1–1 mM) infusion. Note the mean frequency change of 50 ROIs after CNO (top right) (Wilcoxon rank-sum test, \*\*$P$ = 0.0282). Bottom right, hotspot ROIs responding to two CNO applications. **y**, The mean peak frequency changes in SF-iGluSnFR signal after stimulus (Ach, CNO or ACSF) in responding and non-responding astrocytes (individual data are shown in Extended Data Fig. 7g). Scale bars, 10 µm (**a**,**c**,**d**,**g**,**h**,**k**,**l**,**n**,**o**,**t**,**u**,**w** and **x**).

glutamatergic gliotransmission at DG excitatory synapses[9,36]. Of 18 tested astrocytes, all responded to L-Glut (Fig. 2g and Extended Data Figs. 4b,f,k and 6e,f), but only six reliably responded to 2MeSADP (Fig. 2g–i,p and Extended Data Fig. 6e,f), while the remainder were classified as non-responders (Extended Data Figs. 4b,j–l and 6e). The 2MeSADP-evoked SF-iGluSnFR responses for kinetics (Fig. 2i and Extended Data Fig. 6f) and spatial properties were similar to the CNO-evoked responses. Notably, they also displayed specific hotspots of release (Fig. 2g,h), which, contrastingly, were smaller and more numerous than with CNO (Fig. 2q).

To confirm the cell specificity of the 2MeSADP-evoked P2Y1R signalling inducing glutamate release in astrocytes, we replicated the experiments in mice[39] with induced *P2y1r* deletion selectively in DGML astrocytes (P2Y1R$^{GFAP-KO}$; Fig. 2m,n and Methods). None of the 20 tested P2Y1R$^{GFAP-KO}$ cells exhibited a significant SF-iGluSnFR response to 2MeSADP (Fig. 2n–p and Extended Data Fig. 6g), despite all of them exhibiting L-Glut responses comparable to WT astrocytes (Fig. 2n and Extended Data Figs. 4e,f and 6g,h). Thus, chemogenetic and endogenous G$_q$-GPCR stimulation in situ both evoke hotspots of fast glutamate release in a subpopulation of DGML astrocytes.

To assess the relevance of glutamate-secreting astrocytes in vivo, we turned to experiments in awake mice. At first, we performed fibre photometry recordings in mice expressing G$_q$-DREADD and SF-iGluSnFR in DGML astrocytes. Using an optofluid cannula implanted above the

DG (Extended Data Fig. 7a), we first locally infused vehicle and then CNO solutions in the presence of a synaptic blocker mixture adapted to in vivo experiments (Methods). In all of the tested mice, application of CNO, and not the vehicle, produced a significant small transient elevation in the basal SF-iGluSnFR fluorescence (Extended Data Fig. 7b).

We next moved to higher-resolution two-photon astrocyte SF-iGluSnFR imaging. We focused on the primary visual cortex, a region in which cholinergic afferents were reported to control the excitatory circuit through astrocyte glutamate signalling[40]. Preliminarily, we analysed three integrated visual cortex scRNA-seq databases from mouse, macaque and human, and confirmed the presence of a subpopulation of glutamatergic astrocytes (Methods and Extended Data Fig. 7c). We next imaged SF-iGluSnFR signals in awake mice, using acute cranial windows that also enabled local drug delivery (Fig. 2r–t and Methods). The intrinsic visual cortex signal that we recorded reported natural extracellular glutamate fluctuations sensed by the astrocytes (Extended Data Fig. 7d,e). Pharmacological inhibition of neuronal activity[41] (Methods) strongly suppressed this signal, most notably its synchronized components[42] (Extended Data Fig. 7e,f). What remained was slow-frequency asynchronous activity, which probably reflected spontaneous, local glutamate release events. To investigate a possible astrocytic origin of this release, we introduced acetylcholine (Ach), a physiologically relevant stimulus for visual cortex astrocytes. We infused the neuromodulator locally[40] and evaluated its effect on the

frequency of asynchronous SF-iGluSnFR events observed within regions of interest (ROIs) for each astrocyte (Fig. 2u and Methods). In 3 out of 11 imaged astrocytes, Ach significantly increased the mean SF-iGluSnFR event frequency within ROIs (Fig. 2u). We classified these astrocytes as responders (Fig. 2y and Extended Data Fig. 7g (top)). To assess the specificity of the Ach effect, we performed analogous stimulations with artificial cerebrospinal fluid (ACSF), which produced no response in 3 out of 3 tested astrocytes (Fig. 2y and Extended Data Fig. 7g (bottom)). Notably, in one Ach-responding astrocyte, restimulation with the neuromodulator induced responses in several of the ROIs that already responded to the first Ach challenge (Fig. 2u), which is consistent with the existence of hotspots of glutamate release. To support the astrocytic origin of Ach-evoked glutamate release, we compared SF-iGluSnFR responses to Ach with responses elicited by cell-selective chemogenetic stimulation of visual cortex astrocytes (Fig. 2v,w). CNO significantly increased SF-iGluSnFR peak frequency in 3 out of 8 imaged astrocytes expressing $G_q$-DREADD (Fig. 2x,y and Extended Data Fig. 7g (middle)). When reapplied to the three responding cells, CNO, like Ach, produced spatially consistent SF-iGluSnFR responses (Fig. 2x (bottom right)).

Taken together, our data in brain slices and awake mice show that both chemogenetic and natural stimulations in the presence of synaptic blockers trigger local subsecond SF-iGluSnFR signal elevations in astrocytes. The responses in situ were suppressed by astrocyte-selective deletion of *P2y1r* (2MeSADP-evoked responses) or *Slc17a7* (CNO-evoked responses), indicating that glutamate release is from astrocytes, occurs after astrocyte $G_q$-GPCR activation and involves a vesicular exocytosis pathway. Glutamate release responses always took place at specific hotspots of an astrocyte and only subpopulations of astrocytes were responders. These findings provide direct functional evidence for the existence of a specialized population of glutamatergic astrocytes predicted by transcriptomic studies.

## Matching molecular and functional profiles

To determine whether glutamate-secreting astrocytes in functional experiments corresponded to the transcriptomically predicted glutamatergic astrocytes, we combined SF-iGluSnFR imaging in situ with scRNA-seq analysis of the imaged cells using patch-seq. We first set up astrocyte patch-seq[43,44] in the DGML using hippocampal slices from mice conditionally expressing tdTomato under the astrocyte *GFAP* promoter (Extended Data Fig. 8a–c and Supplementary Video 2). A total of 65 whole-cell patched red-fluorescent cells displaying morphology (Extended Data Fig. 8c) and electrical properties typical of astrocytes (Extended Data Fig. 8d) and 20 additional cells whole-cell patched after SF-iGluSnFR imaging (same protocol as in Fig. 2a) were retained after quality control. These 85 cells were molecularly examined using our integrated hippocampal astrocytic database as a reference (Methods). We confirmed that the patch-seq cells had transcriptional features typical of astrocytes and corresponded molecularly to several of the clusters that were previously identified, including glutamatergic astrocytes (28 out of 85 cells; Extended Data Fig. 8e,f). As expected, the subpopulation of patched cells classified as cluster 7 was enriched in transcripts for VGLUTs (*Slc17a7* and, to a lesser extent, *Slc17a6*), core SNARE proteins (*Snap25*, *Stx1b*, *Stx4a* and *Vamp2*) and $Ca^{2+}$ sensors (*Syt1* and *Syt7*) among others (Extended Data Fig. 8f). However, this subpopulation did not differ electrophysiologically from the patch-seq population overall (Extended Data Fig. 8d), confirming typical astrocytic features. Among the 20 patch-seq cells that underwent glutamate imaging, 4 were functionally classified as responders and 16 as non-responders (Extended Data Fig. 8g). Transcriptomic annotation correctly predicted 75% of the responders and 88% of the non-responders, reaching statistical significance for correct prediction (Extended Data Fig. 8h). These data indicate a robust correlation between our physiological and molecular identification of glutamatergic astrocytes.

## Roles in hippocampal function and dysfunction

To investigate a potential role for glutamatergic astrocytes in synaptic functions and behaviour, we used an inducible transgenic mouse model (*GFAP*$^{creERT2}$*Slc17a7*$^{fl/fl}$*tdTomato*$^{lsl/lsl}$ mice) enabling selective *Slc17a7* gene deletion in astrocytes after tamoxifen (TAM) administration (Fig. 3a and Methods). We first confirmed that *cre*-recombined astrocytes in our mice (tdTomato$^+$; Fig. 3c and Extended Data Fig. 9c) had a deleted *Slc17a7* locus (VGLUT1$^{GFAP-KO}$; Fig. 3b and Extended Data Fig. 9a), as well as that *cre* recombination occurred in a strictly TAM-dependent (Extended Data Fig. 9b) and cell-specific (Fig. 3c and Extended Data Fig. 9c) manner. We next examined putative roles for glutamatergic astrocytes in synaptic plasticity (Fig. 3d–f) using hippocampal slices of *GFAP*$^{creERT2}$*Slc17a7*$^{fl/fl}$*tdTomato*$^{lsl/lsl}$ mice and focusing on DG perforant path–granule cell (PP–GC) synapses, which reportedly are under presynaptic control by glutamatergic gliotransmission[9,36]. By exposing mice to a short TAM injection protocol (Methods), we triggered sparse *cre* recombination in astrocytes[37]. This enabled us to compare theta-burst-evoked long-term potentiation (Θ-LTP), induced by medial PP fibre stimulation, in pairs of synaptic fields around 200 μm apart, containing a VGLUT1$^{GFAP-KO}$ astrocyte (Astro tdTom$^+$) and an unrecombined astrocyte (Astro, Fig. 3d, Methods and Extended Data Fig. 9e). On average, the magnitude of Θ-LTP was significantly lower in the synaptic fields containing a VGLUT1$^{GFAP-KO}$ tdTomato$^+$ astrocyte (Fig. 3e,f). This result contrasted with what we observed in analogous WT experiments, in which the magnitude of Θ-LTP in pairs of synaptic fields also around 200 μm apart was identical (Extended Data Fig. 9d). We could not attribute this reduction to a change in the baseline value or the excitability of the synapses localized to the fields containing VGLUT1$^{GFAP-KO}$ astrocytes (Extended Data Fig. 9f), nor to impaired $Ca^{2+}$ signalling in VGLUT1$^{GFAP-KO}$ astrocytes (Extended Data Fig. 9g–l). Thus, the decreased Θ-LTP magnitude appears to depend specifically on astrocyte VGLUT1 deletion, implying that glutamatergic astrocytes exert a VGLUT1-dependent positive control on Θ-LTP of PP–GC synapses residing within their territory.

We next evaluated whether astrocyte VGLUT1 deletion could affect hippocampal memory processing. For this, we ran the contextual fear-conditioning (CFC) memory test and evaluated the performance of VGLUT1$^{GFAP-KO}$ mice compared with two control groups, VGLUT1$^{GFAP-WT}$ and VGLUT1$^{WT-TAM}$ mice (Fig. 3g–i and Methods). Preliminarily, we excluded any confounding effect of TAM-induced *cre* recombination on motor function, exploration, anxiety and emotional state of VGLUT1$^{GFAP-KO}$ mice (Extended Data Fig. 9m). During fear conditioning, all three mouse groups learned proficiently, exhibiting similar levels of conditioned fear at the end of the session (Fig. 3h). Moreover, all of the groups moved the same distance during electroshocks (Extended Data Fig. 9m), which excluded the possibility that VGLUT1$^{GFAP-KO}$ mice differed in sensitivity from the controls. When we measured contextual memory expression 24 h later, only VGLUT1$^{GFAP-WT}$ and VGLUT1$^{WT-TAM}$ mice exhibited fear levels resembling those acquired during conditioning. VGLUT1$^{GFAP-KO}$ mice expressed significantly less fear than the control mice (Fig. 3i). Then, 48 h after conditioning, all of the groups exhibited low levels of residual fear, with the level in VGLUT1$^{GFAP-KO}$ mice still significantly lower than in VGLUT1$^{GFAP-WT}$ controls (Fig 3i). The contextual memory defect observed after astrocyte VGLUT1 deletion indicates that glutamatergic astrocytes have a function in physiological memory processing.

We then examined whether glutamatergic astrocytes could contribute to altered cortico-hippocampal circuitry function. We focused on epileptic seizures, given the proposed roles for astrocyte glutamate release in seizure initiation or amplification[5,45]. We triggered acute seizures in vivo by subcutaneous administration of a single-dose of kainate and compared the responses in VGLUT1$^{GFAP-KO}$ mice with those in VGLUT1$^{GFAP-WT}$ and VGLUT1$^{WT-TAM}$ controls (Fig 3j,k and Methods). In all of the mouse groups, the time of onset for the first seizure (Fig. 3l) and the total period in which the mice experienced seizures (Fig. 3n) were similar. However, once the

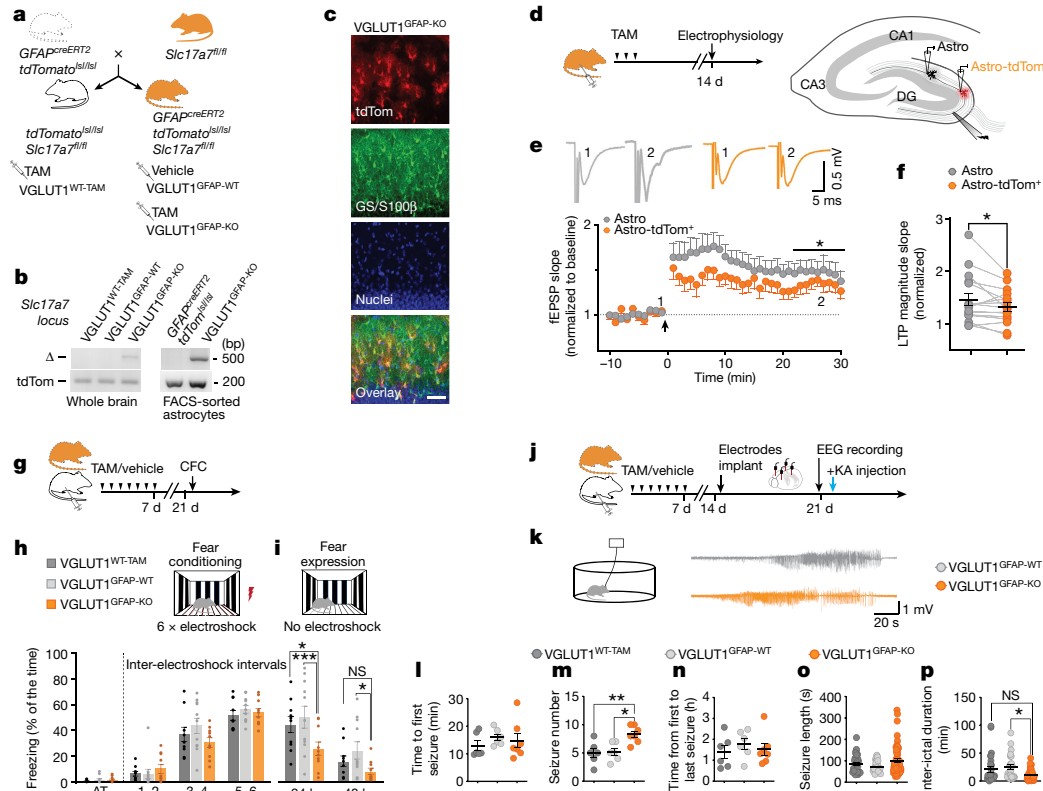

**Fig. 3 | VGLUT1 deletion in astrocytes leads to changes in LTP, memory and acute seizure patterns in the cortico-hippocampal circuitry. a**, The breeding scheme for generating astrocyte-specific conditional VGLUT1 mice (VGLUT1[GFAP-KO] after TAM-induced *cre* recombination) and the related controls: VGLUT1[GFAP-WT] mice controlling for *cre* leakage, and VGLUT1[WT-TAM] mice controlling for TAM-induced *cre*-unrelated effects (Methods). **b**, Validation of *Slc17a7* locus genetic deletion (Δ band) in whole-brain homogenates (2 mice per group) and FACS-sorted astrocytes (2 independent experiments, 5 mice per group) from VGLUT1[GFAP-KO] mice. **c**, Representative images (*n* = 2) showing *cre*-recombination reporter expression (tdTomato), astrocyte labelling (combined GS and S100β) and nuclear staining in the DGML of VGLUT1[GFAP-KO] mice. The overlay shows reporter co-localization with astrocytes. Scale bar, 50 μm. **d**, The experimental paradigm for generating sparse VGLUT1[GFAP-KO] astrocytes (Astro-tdTom[+], left) and comparatively studying the Θ-LTP in two neighbouring DGML synaptic fields containing an Astro-tdTom[+] and a WT (Astro) astrocyte (right; Extended Data Fig. 9e). **e**, Representative fEPSP traces and the time-course of the fEPSP slope before and after Θ-LTP induction (arrow) in synaptic field pairs containing Astro versus Astro-tdTom[+] (16 slices, 12 mice) (bottom). The mean LTP was lower in Astro-tdTom[+] fields (two-tailed paired *t*-test; *P = 0.044). **f**, The normalized Θ-LTP magnitude of individual pairs in **e** (two-tailed paired *t*-test;

*P = 0.044). **g**, The experimental paradigm and timeline of mouse treatments and behavioural testing (Methods). **h,i**, The contextual fear conditioning test was performed in VGLUT1[GFAP-KO] (*n* = 10), VGLUT1[GFAP-WT] (*n* = 11) and VGLUT1[WT-TAM] (*n* = 10) mice. **h**, Mice were exposed to an activity test (AT) followed by contextual fear conditioning. All mouse groups showed comparable learning (two-way analysis of variance (ANOVA) with Fisher's least significant difference (LSD) test; *P* = 0.60). **i**, Fear expression was evaluated 24 h and 48 h after the conditioning test: VGLUT1[GFAP-KO] mice show reduced performance compared with the control mice (two-way ANOVA with Fisher's LSD test; *P = 0.0101 (24 h), ***P = 0.0007 (24 h), *P = 0.0215 (48 h)). **j**, The experimental paradigm and timeline of mouse treatments, electroencephalogram (EEG) recordings and induction of acute seizures (Methods). **k**, Representative EEG traces of seizures recorded from a VGLUT1[GFAP-WT] and a VGLUT1[GFAP-KO] mouse after injection of kainic acid (KA, 10 mg per kg). **l–p**, Seizure parameters were analysed in VGLUT1[GFAP-KO] (*n* = 7), VGLUT1[GFAP-WT] (*n* = 6) and VGLUT1[WT-TAM] (*n* = 7) mice. Analysis of the specific differences between VGLUT1[GFAP-KO] and the control mice on the basis of the time to the first seizure (**l**); the total seizure number per mouse (one-way ANOVA with Tukey's test; **P = 0.0083, *P = 0.0120) (**m**); the time from first to last seizure (**n**); individual seizure length (**o**); and the inter-ictal duration (Kruskal–Wallis with Dunn's test; *P = 0.0232) (**p**).

seizures started, VGLUT1[GFAP-KO] mice underwent more episodes compared with the control groups (Fig. 3m) with individual episodes tending to be longer lasting (Fig. 3o), which resulted in significantly reduced inter-ictal periods (Fig. 3p). Overall, these data show that glutamatergic astrocytes have active roles not only in physiological processes but also in pathological processes. More specifically, they reveal a protective function of astrocyte VGLUT1-dependent signalling against kainate-induced acute seizures in vivo, notably opposing the mechanisms causing seizure amplification. This function is worth examining further in chronic epilepsy models for possible therapeutic perspectives.

## Roles in the nigrostriatal circuitry

We next investigated whether glutamatergic astrocytes regulate additional brain circuits. We focused on the mesencephalic dopaminergic

(DA) circuit connecting the substantia nigra pars compacta (SNpc) to the dorsal striatum (dST). This circuit is a key pathway for the control of voluntary movement[46], and its degeneration is the hallmark of Parkinson's disease[47]. First, we integrated three existing substantia nigra databases (Methods) to obtain scRNA-seq data of substantia nigra astrocytes and established a correlation with hippocampal astrocyte cluster 7 through label transfer (Extended Data Fig. 10a). Interrogation of the entire substantia nigra database led us to also confirm that VGLUT2 prevails over VGLUT1 in the substantia nigra[35]. Thus, for studying glutamatergic astrocytes in the SNpc, we generated an astrocyte-targeted VGLUT2-related line, *GFAP[creERT2]Slc17a6[fl/fl]tdTomato[lsl/lsl]* mice, obtained from existing *Slc17a6[fl/fl]* mice[48] (Fig. 4a). We confirmed that *cre* recombination in these mice was TAM specific, astrocyte selective and led to the deletion of the *Slc17a6[fl/fl]* sequence in astrocytes (VGLUT2[GFAP-KO]; Extended Data Fig. 10b–d). We then performed electrophysiology

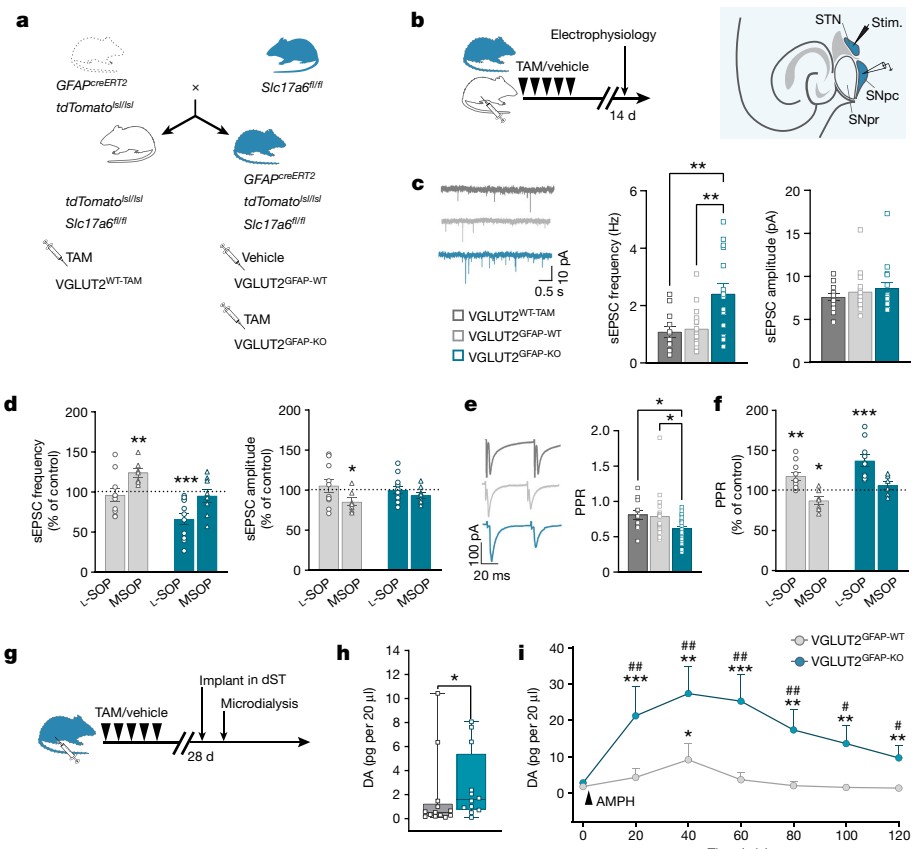

**Fig. 4 | VGLUT2 deletion in astrocytes alters nigrostriatal circuit function in situ and DA levels in vivo. a**, The breeding scheme for generating astrocyte-specific conditional VGLUT2 mice and related controls (details as in Fig. 3a, but for VGLUT2). **b**, The experimental paradigm and timeline of mouse treatments for electrophysiology recordings (left). Right, schematic of midbrain slices showing the STN, SNpc and substantia nigra pars reticulata (SNpr) with the position of the stimulating and recording electrodes. **c**, sEPSCs recorded in SNpc DA neurons of VGLUT2[GFAP-KO] (15 cells, 6 mice), VGLUT2[GFAP-WT] (20 cells, 7 mice) and VGLUT2[WT-TAM] (13 cells, 5 mice) mice. Representative current traces (left), and histograms showing, in VGLUT2[GFAP-KO] mice, increased sEPSC frequency (middle; one-way ANOVA with Tukey's test; **$P = 0.00187$ (bottom), **$P = 0.00233$ (top)) and unchanged amplitude compared with the controls (right). **d**, Group III mGluR agents differently affect sEPSCs in VGLUT2[GFAP-KO] mice compared with the control mice (two-tailed paired $t$-test). The histograms show the percentage change induced by group III mGluR agonist L-SOP (10 μM) and antagonist MSOP (10 μM) on the baseline sEPSC frequency (left) and amplitude (right) in VGLUT2[GFAP-KO] mice (L-SOP: 12 cells, 5 mice, ***$P = 0.0005$; MSOP: 8 cells, 3 mice) and VGLUT2[GFAP-WT] mice (L-SOP: 10 cells, 5 mice; MSOP: 7 cells, 4 mice; **$P = 0.0041$, *$P = 0.024$). **e**, EPSCs evoked in SNpc DA neurons by STN stimulation in VGLUT2[GFAP-KO] (24 cells, 7 mice), VGLUT2[GFAP-WT] (19 cells, 7 mice) and VGLUT2[WT-TAM] (12 cells, 5 mice) mice. Left, representative traces of paired pulse-evoked EPSCs. Right, histograms showing a reduced PPR in

VGLUT2[GFAP-KO] mice compared with in the control mice (one-way ANOVA with Fisher's test; *$P = 0.020$ (top), *$P = 0.023$ (bottom)). **f**, Differential effects (expressed as the percentage change versus the control) induced by group III mGluRs agents on PPR in VGLUT2[GFAP-KO] (L-SOP: 10 cells, 4 mice; ***$P = 0.00065$; MSOP: 6 cells, 4 mice) compared with in VGLUT2[GFAP-WT] (L-SOP: 10 cells, 5 mice; **$P = 0.0079$; MSOP: 6 cells, 4 mice; *$P = 0.038$) mice. Statistical analysis was performed using two-tailed paired $t$-tests. **g**, The experimental paradigm and timeline of mouse treatments for in vivo microdialysis measures of DA levels in the dST. **h**, The baseline DA levels in VGLUT2[GFAP-KO] mice ($n = 12$) compared with in VGLUT2[GFAP-WT] mice ($n = 13$; Kolmogorov–Smirnov test; *$P = 0.039$). For the box plots, the box limits show the 25th to 75th percentiles, the centre lines are medians, and the whiskers show the minimum to maximum values. **i**, Time course of DA levels after amphetamine challenge (AMPH, 2 mg per kg; arrow). The DA levels were significantly (Friedman ANOVA with Wilcoxon signed rank test) increased only at 40 min in VGLUT2[GFAP-WT] mice (*$P = 0.0175$), whereas DA levels were significantly increased at 20, 40, 60, 80, 100 and 120 min in VGLUT2[GFAP-KO] mice (##$P = 0.00253$, ##$P = 0.00253$, ##$P = 0.0025$, ##$P = 0.0042$, #$P = 0.010$, #$P = 0.03$, respectively). The amphetamine-induced increase was higher in VGLUT2[GFAP-KO] mice compared with in VGLUT2[GFAP-WT] mice at any tested time (Kolmogorov–Smirnov test; ***$P = 0.00049$, **$P = 0.009$, ***$P = 0.00049$, **$P = 0.00231$, **$P = 0.00913$, **$P = 0.00231$). All data are mean ± s.e.m.

studies in midbrain slices from VGLUT1[GFAP-KO] and VGLUT2[GFAP-KO] mice and their corresponding VGLUT[GFAP-WT] and VGLUT[WT-TAM] controls. We initially observed that *cre* recombination in astrocytes did not affect the basic physiology of SNpc DA neurons (Extended Data Fig. 10e,f; TAM protocols are shown in Fig. 4b and Methods). However, when we recorded synaptic transmission in these neurons, we found that spontaneous excitatory synaptic currents (sEPSCs) occurred at a significantly higher frequency in VGLUT2[GFAP-KO] (but not VGLUT1[GFAP-KO]) mice compared with in their controls (Fig. 4c and Extended Data Fig. 10g). These data not only confirmed the predominant role of VGLUT2 in the SNpc circuit, but also suggested an inhibitory role for astrocyte

VGLUT2 in controlling the excitatory synaptic input to SNpc DA neurons. This input largely depends on glutamatergic afferents from the subthalamic nucleus (STN)[49] and, consistent with the above hypothesis, their stimulation evoked significantly larger EPSCs (with a reduced paired-pulse ratio (PPR)) in SNpc DA neurons of VGLUT2[GFAP-KO] mice compared with in their controls (Fig. 4e). Given the reported presence of inhibitory presynaptic group III metabotropic glutamate receptors (mGluRs) on STN afferents[50], we sought to determine whether astrocyte VGLUT2-dependent signalling could be an endogenous activator of these receptors. Accordingly, we interfered pharmacologically with group III mGluRs function, using either an agonist (*O*-phospho-L-serine,

L-SOP) or an antagonist (α-methylserine-O-phosphate, MSOP), and compared the effects of the drugs on EPSCs in SNpc DA neurons of VGLUT2[GFAP-KO] mice and VGLUT2[GFAP-WT] controls (Fig. 4d,f). The presence of L-SOP significantly reduced the increased frequency of spontaneous events observed in VGLUT2[GFAP-KO] mice to levels comparable with those in the control mice (Fig. 4d). However, L-SOP did not modify the frequency of the events in VGLUT2[GFAP-WT] controls, or their amplitude in either mouse group. By contrast, MSOP had opposing effects, causing increased sEPSC frequency in controls, without altering the frequency in VGLUT2[GFAP-KO] mice (Fig. 4d). When we recorded evoked events (eEPSCs; Fig. 4f), L-SOP enhanced PPR more in VGLUT2[GFAP-KO] mice compared with in the controls, whereas MSOP reduced the PPR in the controls but not in the VGLUT2[GFAP-KO] mice. Overall, these results strongly support an endogenous regulatory function of astrocyte VGLUT2-dependent signalling in shaping glutamatergic synaptic transmission onto nigral DA neurons through the activation of presynaptic group III mGluRs.

To evaluate the relevance of astrocyte-mediated inhibitory control on the nigrostriatal circuit function in vivo, we measured DA levels in the dST of VGLUT2[GFAP-KO] mice and VGLUT2[GFAP-WT] controls by microdialysis (Fig. 4g). Measurements were performed under basal conditions and after amphetamine challenge (Methods). Basal DA levels in VGLUT2[GFAP-KO] mice were significantly higher compared with in the control mice (Fig. 4h). Moreover, amphetamine produced a greater and more prolonged increase in extracellular DA in VGLUT2[GFAP-KO] mice compared with in the controls (Fig. 4i). These data are consistent with a loss of presynaptic inhibition to SNpc DA neurons in VGLUT2[GFAP-KO] mice, while not excluding an effect on additional astrocyte controls in the dST[51]. Independent of the specific mechanism(s), they reveal that astrocyte VGLUT2-dependent signalling regulates nigrostriatal DA pathway function in vivo. Considering the reported efficacy of group III mGluR agonists in improving motor symptoms in Parkinson's disease animal models[52], astrocyte VGLUT2-dependent signalling represents a potential therapeutic target for Parkinson's disease.

## Conclusions

The case for $Ca^{2+}$-dependent glutamate exocytosis from astrocytes and glutamatergic gliotransmission has long been controversial[11–13] owing to the coexistence of supporting[2,8–10,36] and opposing evidences[14–17]. Our study provides key information to resolve the debate. We describe a subpopulation of specialized astrocytes with a discrete molecular signature resembling that of glutamatergic synapses, defined anatomical distribution and functional competence for VGLUT-dependent glutamate release in situ and in vivo. These data not only demonstrate that astrocyte glutamate exocytosis exists in the adult brain, but can explain why previous reports[15,17] were unable to find expression of VGLUTs and regulated exocytosis proteins in astrocytes. Negative results came mainly from bulk RNA-seq studies of entire brain regions that dilute the contribution of minority, unequally distributed, subpopulations like glutamatergic astrocytes (Fig. 1g,h). As a consequence, the levels of their differentially expressed genes may appear to be negligible when scaled to genes expressed by the whole sampled population. However, this minority contribution was detected by single-cell transcriptomic analyses and identified here in all of the single-cell and single-nucleus RNA-seq databases that we analysed, including our custom patch-seq dataset, and was corroborated by direct visualization of the glutamatergic subpopulation in RNAscope HiPlex experiments. If glutamatergic gliotransmission is a specialized function of peculiar astrocytes with defined anatomical locations, our data may also explain discrepancies among previous functional studies that did not consider in their protocols the intricacies identified here. Moreover, our study adds to the understanding of astrocyte diversity[18–21,23], suggesting that different groups of specialized astrocytes have distinct roles in brain function. By using astrocyte-targeted genetic VGLUT deletion, we revealed that glutamatergic astrocytes contribute to cortico-hippocampal and

nigrostriatal circuit function during normal behaviour and pathological processes. The identified actions—strengthening LTP and hippocampal memories, opposing hyperexcitation during seizures and, conceivably, STN overactivation in Parkinson's disease—testify to the functional relevance of these specialized astrocytes, despite their relative numerical paucity, and highlight their potential as targets for CNS protective therapies.

Future studies are expected to generate CNS-wide maps that will help to define the overall distribution of glutamatergic astrocytes and their full range of actions, and to better understand why this atypical astrocytic population exists and by which specific modalities it integrates anatomically and functionally into CNS circuits, as well as if and how its altered properties contribute to defined pathological CNS conditions.

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

## Methods

### Reagents

A list of the reagents used in this study is provided in Supplementary Table 3.

### Animals

C57BL/6JRj (WT, from Janvier) mice and transgenic mouse lines were housed at two to five animals per cage under a 12 h–12 h light–dark cycle (lights on from 07:00 to 19:00) at a constant temperature (23 °C) and humidity (~50%) with ad libitum access to food and water. All animal protocols in the present study were approved by the Swiss Federal and Cantonal authorities (VD1873.1, VD2982, VD3053.1, VD3115.1) or by the Council Directive of the European Communities (2010/63/EU), and the Animal Care Committee of Italian Ministry of Health (375/2018-PR). Mice were used at different postnatal (P) ages according to experimental type (specified in corresponding sections).

### Transgenic animal models

We used several transgenic mouse lines, some of which were generated within the present study. Mice carrying the inducible version of *cre* (*cre-ERT2*) under the human glial fibrillary acidic protein (*GFAP*) promoter[53] (*GFAP*^creERT2; Tg(GFAP-cre/ERT2)1Fki) were cross-bred with a conditional tdTomato reporter mouse line (*tdTomato*^lsl/lsl; B6.Cg-Gt(ROSA)26Sort m14(CAG-tdTomato)Hze/J; Ai14, Jackson, 007914) for two generations to obtain *GFAP*^creERT2*tdTomato*^lsl/lsl mice. The *GFAP*^creERT2 gene was always maintained in heterozygosis. To produce a conditional allele of the mouse *Slc17a7* gene encoding VGLUT1, *GFAP*^creERT2*tdTomato*^lsl/lsl mice were back-crossed with *Slc17a7*^fl/fl mice[38] to obtain *GFAP*^creERT2*Slc17a7*^fl/fl *tdTomato*^lsl/lsl mice and *Slc17a7*^fl/fl*tdTomato*^lsl/lsl littermates. Likewise, to produce *GFAP*^CreERT2*Slc17a6*^fl/fl*tdTomato*^lsl/lsl mice, we back-crossed *GFAP*^creERT2*tdTomato*^lsl/lsl mice to VGLUT2-flox mice (*Slc17a6*^fl/fl; B6;129/Sv-Slc17a6tm1.1Edw, Jackson, 636372[48]). To achieve gene recombination in the *cre*-inducible lines and their littermate controls, mice were administered TAM (100 mg per kg Sigma-Aldrich, T5648, dissolved in corn oil) or vehicle (Sigma-Aldrich, C8267), according to different protocols depending on the type of experiment. The injection protocol used in each type of experiment as well as the interval observed from the first TAM or vehicle injection to the experiment are specified in the specific method's section for each experiment as well as in main and extended data figures. For simplicity, we called *GFAP*^creERT2*Slc17a7*^fl/fl*tdTomato*^lsl/lsl mice treated with TAM and their controls, that is, *GFAP*^creERT2*Slc17a7*^fl/fl*tdTomato*^lsl/lsl mice treated with vehicle and *Slc17a7*^fl/fl *tdTomato*^lsl/lsl mice treated with TAM, respectively, VGLUT1^GFAP-KO, VGLUT1^GFAP-WT and VGLUT1^TAM-WT. Likewise, *GFAP*^creERT2*Slc17a6*^fl/fl *tdTomato*^lsl/lsl mice treated with TAM, *GFAP*^creERT2*Slc17a6*^fl/fl*tdTomato*^lsl/lsl mice treated with vehicle and *Slc17a6*^fl/fl*tdTomato*^lsl/lsl mice treated with TAM, for simplicity were called VGLUT2^GFAP-KO, VGLUT2^GFAP-WT and VGLUT2^TAM-WT, respectively. In all experiments using littermate mice in different pharmacological treatments, animals were randomized in the various groups to avoid cage, litter and batch effects. Recombination efficacy and specificity were evaluated by genomic PCR analysis and by tdTomato reporter expression (Fig. 3b and Extended Data Figs. 4 and 10). Transgenic lines were screened by PCR analysis for the presence of the transgenes in genomic DNA purified from digital biopsies (5–11 days after birth). The primers used were as follows: *hGFAP*^creERT2: 500 bp, cre-sense 5′-CAGGTTGGAGAGGAGACGCATCA-3′ and cre-antisense 5′-CGTTGCATCGACCGGTAATGCAGGC-3′; *tdTomato*^lsl/lsl: 196 bp, IMR 9103 5′-GGCATTAAAGCAGCGTATCC-3′; IMR9105 5′-CTGTTCCTGTACGGCATGG3′; *Slc17a7*^fl/fl: 270 bp WT-367 bp flox, 60483flp-KHA1 5′-GAAATTGGAGTTGTGTGTGGTGGAGC-3′; 60484flp-KHA1 5′-CCACAATGGCAAAGCCAAAGACC; *Slc17a6*^fl/f: 190 bp WT, 380 bp flox, 1176 sense, 5′-CAGTGTGCTGTAACTGAGATAGT-3′; 1346-antisense, 5′-TCTTTTGGGGTGCCATTTCAACACT-3′. In a limited set of imaging experiments, we used *GFAP*^creERT2*GCaMP6f*^fl/fl mice (B6; Tg (GFAP-cre/ERT2)1Fki crossed with B6;129S-Gt(ROSA)26Sor<tm95.1 (CAG-GCaMP6f)Hze (Ai95D, Jackson, 024105) (Extended Data Fig. 5), previously generated in our laboratory and described in ref. 37, and knock-in *GLAST*^creERT2 mice (*Slc1a3*^tm1(cre/ERT2)Mgoe (MGI: 3830051) crossbred to P2Y1 receptor flox mice (*P2ry1*^fl/fl, from C. Gachet), that is, *GLAST*^creERT2 *P2ry1*^fl/fl mice[39] (Fig. 2 and Extended Data Figs. 4 and 6).

### Preparation of a single-cell suspension from mouse brain regions

Separate batches of cortical and midbrain astrocytes from VGLUT1^GFAP-KO and *GFAP*^creERT2*tdTomato*^lsl/lsl mice (3–5 months old) treated with TAM (1 intraperitoneal (i.p.) injection per day for 8 days, long protocol) and VGLUT1^GFAP-WT treated with vehicle 30–90 days before, were prepared at equivalent circadian times, using multiple mouse litters as described previously[19]. In brief, cortices and midbrains were quickly and carefully dissected in cold Hanks' balanced salt solution (HBSS) buffer without Ca²⁺ and Mg²⁺, under a dissection microscope. Myelinated parts were discarded, to decrease the debris in the final cell suspension. Each cell suspension was prepared starting from 5 animals. Tissue dissociation was run using the neural tissue dissociation kit (P) (Miltenyi Biotec). Tissue was digested at 37 °C using papain, supplemented with DNase I and then mechanically dissociated using three rounds of trituration with 5 ml serological pipettes. The resulting suspension was filtered through a 20 μm strainer (RUAG) to remove any remaining clumps. Contamination by myelin and cell debris was removed by equilibrium density centrifugation. 90% Percoll PLUS (Life Sciences) in 1× HBSS with Ca²⁺ and Mg²⁺ (Sigma-Aldrich) was added to the suspension to produce a final concentration of 24% Percoll. Further DNase I (Worthington) was added (125 U per 1 ml) before centrifugation of the cell suspension at 300*g* for 11 min at room temperature (with minimal centrifuge braking). The resulting cell pellet was resuspended in Dulbecco's phosphate-buffered saline (dPBS) (without Ca²⁺ and Mg²⁺) containing 0.5% bovine serum albumin (BSA) (Sigma-Aldrich). The supernatants were centrifuged again at 300*g* for 10 min at room temperature. Any pelleted cells were resuspended in 0.5% BSA/dPBS (without Ca²⁺ and Mg²⁺).

### FACS isolation of astrocytes and genomic PCR

To exclude dead cells during FACS, the vital dye DAPI (1:100 dilution, Invitrogen) was added to the single-cell suspension and filtered through a 20 μm Nitex mesh. FACS analysis was performed on the BD FACSAria III (BD FACSDiva v.8.0.1) system using a 100 μm nozzle. Compensations were done on single-colour control (tdTomato) and gates were set on control samples (from VGLUT1^GFAP-WT mice). Forward scatter/side scatter gatings were used to remove clumps of cells and debris (plots produced with FlowingSoftware v.2.5.1). After sorting, cells were centrifuged at 300*g* for 15 min at 4 °C, the supernatants were discarded, and the pellet was snap-frozen in dry ice and stored at −80 °C. DNA was extracted from the pelleted cells, as well as from the whole brain control samples, using QIAamp DNA kit according to the manufacturer's instructions. PCR reactions were performed using the Go taq polymerase hot start kit (Promega) with the same primers used for genotyping to identify the floxed genes. To identify *Slc17a7* and *Slc17a6* gene deletions, the following primers were used: VGLUT1Δ: 508 bp, 60453bct-KHA15′-TCCTTTTTCTGGGGCTACATTGTCACTC-3′; 60454bct-KHA1 5′-CACCTAGTACCCGCCATTCTTAAACTCC-3′; VGLUT2Δ: 240 bp, 1176 sense-5′-CAGTGTGCTGTAACTGAGATAGT-3′; 1175-antisense 5′-AAAGGTCCTGGATCAGAGCAGG-3′ (Fig. 3b and Extended Data Figs. 4c and 10b).

### Single-astrocyte DNA analysis

Single astrocytes in brain slices of *Slc17a7*^fl/fl mice virally injected (see the 'Stereotaxic viral injections' section) in the hippocampus with *AAV5-hGFAP-eBFP2-iCre* were whole-cell patched. To validate DNA recombination of the *Slc17a7* loci, we collected their intracellular content as described in the 'Patch-seq analysis of astrocytes

from mouse hippocampal DG' section. Nested PCR was then performed using the CellsDirect One-Step qRT-PCR Kit (Thermo Fisher Scientific) according to the manufacturer's instructions with minor modifications. The external primers used for the nested PCR were as follows: VGLUT1 external: 509 bp WT -606 bp flox, Ext-60483flp 5′-AGACTGCTGGCCTACTACATGGCTCC-3′, Ext-60484flp-KAH1 5′-AGCAGGGTTAATGGGGCAGGCTTTACCT-3′; VGLUT1Δ external: 717 bp, Ext-60453bct-KHA1 5′-TGCTGATTGGTAGAGGGTAGAGTCTG GG-3′, Ext-60454bct-KHA1 5′-CCAAAGTCTAGACACACCCACAGCAAT AG-3′. An ExoSAP-IT PCR Product Cleanup (Affymetrix) step to eliminate residual primers was performed before the second PCR step using the VGLUT1Δ and VGLUT2Δ primers that are listed in the 'FACS isolation of astrocytes and genomic PCR' section (Extended Data Fig. 4c). Full gel scans are provided in the Supplementary Data.

### Immunohistochemistry and image analysis

Immunohistochemistry experiments (Fig. 3c and Extended Data Figs. 4c, 9b,c and 10d) were performed in slice preparations from (1) VGLUT1$^{GFAP-KO}$, VGLUT1$^{GFAP-WT}$ and VGLUT1$^{TAM-WT}$ mice (injected at 2 months of age with TAM, vehicle, TAM, respectively, 1 i.p. injection per day for 8 days, long protocol) to evaluate cell-specific recombination in the hippocampus and cerebral cortex; and (2) VGLUT2$^{GFAP-KO}$, VGLUT2$^{GFAP-WT}$ and VGLUT2$^{TAM-WT}$ mice (P21–25; TAM, vehicle, TAM treatment, respectively, 2 i.p. injections per day for 5 days, alternative long protocol) to evaluate cell-specific recombination in the SNpc. In all cases, mice were euthanized with pentobarbital 21 days after the first TAM or vehicle injection, perfused with 4% paraformaldehyde and brains were fixed overnight (4% paraformaldehyde in 1× PBS) at 4 °C. Then, 40-μm-thick sagittal brain slices from the three VGLUT1 mouse groups and horizontal brain slices from the three VGLUT2 mouse groups were cut with a vibratome (Leica Microsystems) and stored at −20 °C in a solution containing ethylene glycol (30%) and glycerol (30%) in 0.05 M phosphate buffer (pH 7.4) until further processing. For immunohistochemistry, slices rinsed in PBS (3 × 10 min) were permeabilized with 0.3% Triton X-100 (10 min), incubated with blocking solution (0.3% Triton X-100, 10% horse serum, 1% BSA in PBS, for 2 h) and with primary antibodies on a horizontal shaker (48 h, 4 °C), then washed in 1× PBS (3 × 10 min) and incubated with secondary antibodies in 0.3% Triton X-100 in 1× PBS at room temperature for 2 h. Next, slices were washed (2 × 10 min) in 1× PBS and incubated with Hoechst33342 (Invitrogen) to label nuclei and mounted onto glass slides using FluoSave reagent (Merk Millipore) for analysis using epifluorescence and confocal microscopy. Primary antibodies used were as follows: anti-S100ß (1:500), anti-GS (1:500), anti-NeuN (1:500), rabbit anti-OLIG2 (1:500), mouse anti-OLIG2 (1:100), anti-IBA1 (1:500), anti-tyrosine hydroxylase (TH, 1:200) and anti-Cre (1:500). Antibodies were revealed with Alexa Fluor 488 or 633 or 555 (1:500) secondary antibodies (details are provided in Supplementary Table 3). The images were acquired using the Leica Axioplan stereomicroscope (×20 objective, Leica Microsystems). In all of the other cases, the images were acquired using the Leica SP5 confocal microscope (Leica Microsystems), using a ×20 oil-immersion objective. For each fluorophore, confocal acquisition consisted of a z-stack (12–20 μm; step size, 0.5–1 μm; frame average, 2; scan speed, 400 Hz; resolution, 1,024 × 1,024 pixels). Laser-excitation wavelength was set at 405 nm for DAPI; 488 nm with an argon laser for Alexa Fluor 488; and 543 nm and 633 nm with a He/Ne laser for tdTomato and Alexa Fluor 633, respectively. Images were visualized using the LAS X software (v.3.7.4., Leica Microsystems) and transformed into .tiff format. To assess recombination in the hippocampus DG (molecular layer), CA1, visual cortex and SNpc regions, cells expressing the reporter gene (tdTomato$^+$ cells) were counted using ImageJ; the ROI was identified using the free hand selection tool and the cell counter plugin was used for manual counting. The final cell density is expressed as cells per mm$^2$. tdTomato$^+$ cells double labelled with GS/s100β, NeuN, OLIG2 or IBA1 markers were also counted and expressed as cells per mm$^2$. A minimum of 140

tdTomato$^+$ cells for each category was counted. In all cases, 2–4 images of 620 × 500 μm from 2–4 slices from 2–3 animals per group were analysed. Images in the figures are confocal image maximum projections with contrast adjusted for display purposes.

### RNAscope HiPlex assay

Male *GFAP$^{creERT2}$tdTomato$^{lsl/lsl}$* mice aged 2 months were treated with TAM (7 days) to induce tdTomato fluorescence expression in GFAP-expressing cells. Then, 21 days after the first injection, mice were perfused, and the brains dissected out and post-fixed overnight at 4 °C in 4% paraformaldehyde. After dehydration with a sucrose gradient (10% and 30%), the brains were embedded in OCT and cryopreserved by snap-freezing in dry-ice-cooled isopentane. The brains were horizontally sliced at 16 μm, using the cryostat (Leica CM3050s), and the slices were mounted onto Superfrost Plus slides, left to dry for 3 h at 37 °C and overnight at room temperature. Before starting the RNAscope HiPlex Assay, the sections were counter-stained with DAPI for 30 s, then coverslipped with ProLong Gold Antifade Mountant. Images of DAPI and tdTomato signals were acquired with a ×40 air objective on a Nikon Ti2 | CrEST Optics X-Light V3 microscope, the same used for the acquisition of RNAscope HiPlex Assay. Once the sections were imaged, the coverslips were removed in 4× SSC buffer. The RNAscope HiPlex Assay was performed according to the manufacturer's standard protocol using the RNAscope HiPlex Kit v2. Tissue sections were baked for 1 h at 60 °C and dehydrated in an ethanol series, followed by antigen retrieval (5 min at 100 °C) and protease treatment (protease III for 30 min at 40 °C). Probes were hybridized for 2 h at 40 °C, washed and hybridized with target-binding amplifiers allowing for signal amplification of single RNA transcripts. The final step of the first round of hybridization attached fluorophores to the first target genes. Once the fluorophores were hybridized, the sections were counterstained with DAPI for 30 s, then mounted for image acquisition. Signal detection was performed in three rounds. In each round, the target genes were labelled with cleavable fluorophores and imaged using a ×40 air objective on the Nikon Ti2 | CrEST Optics X-Light V3 microscope. For each section, the gain and laser power were qualitatively optimized by the experimenter for each channel. After the sections were imaged, the coverslips were removed in 4× SSC buffer and the fluorophores were cleaved using the cleaving solution provided in the kit. A new set of fluorophores targeting the next genes was hybridized onto the tissue sections, another round of DAPI counterstaining was performed and the sections were reimaged as described above. This was repeated until all target genes were imaged. Here the list of the targeted transcripts: T3, *Slc17a7*; T6, *Snap25*; T8, *Syt1*; T9, *Slc17a6*. Five other transcripts were targeted together with the above ones for a different experimental purpose. To identify neuron and astrocyte subpopulations, immunofluorescence labelling was performed in the same tissue sections after the cleavage of the fluorophores from the last round of the HiPlex Assay. The sections were briefly washed in 1× PBS before incubation for 60 min in blocking solution containing 0.25% Triton X-100 and 5% BSA in 1× PBS, and then incubated overnight at 4 °C with antibodies diluted in the blocking solution as follows: goat anti-tdTomato (1:500); mouse anti-S100β (1:500); mouse anti-GS (1:500). The sections were washed in 1× PBS and then incubated for 1 h at room temperature with Alexa Fluor 647 or 568-conjugated secondary antibodies (1:500, details in Supplementary Table 3) diluted in blocking solution (1:500). After three washes in 1× PBS, the sections were counterstained with DAPI and coverslipped using ProLong Gold Antifade Mountant. One final round of imaging was performed as described above to capture the mentioned antibodies and DAPI signals (Fig. 1g,h and Extended Data Fig. 2j).

### RNAscope HiPlex assay analysis

**Image registration.** Images for each set (RNA, rounds 1–3; Proteins, round 4) were registered in the DAPI channel. We treated round 1 (R:1)

as the reference image and placed manual landmarks between each pair of reference (R:1) and moving image (R:$i$) where $i$ = {2, 3, 4}. We then performed an affine registration using the scikit-image[54] library, followed by intensity-based nonlinear registration using the SyN[55] algorithm from the DIPY[56] library. Registration results were assessed visually for correctness.

**Blob detection.** Blob detection was performed according to the standard pipeline as described online (https://spacetx-starfish.readthedocs.io/en/latest/index.html). We first applied a white top hat filter, followed by blob detection using the Laplacian of Gaussian function; parameters were determined individually for each image by assessing the results of the blob-detection step manually. The blob-detection steps were implemented using the scikit-image library[54].

**Cell detection.** Cell detection was done automatically on images in the DAPI channel in each round using the pretrained 2D_versatile_fluo model from Stardist[57].

**RNA counting.** All of the blobs within a distance of 1.5× the radius of a cell from the cell centroid were assigned to that cell. As DAPI stains the nucleus, we consider 1.5× the radius as a conservative estimate of the true cell size. We generated a cell $x$ gene count matrix by counting the transcripts of each probe assigned to individual cells to identify glutamatergic astrocytes in the molecular region of the DG across the dorso-ventral axis. The region was chosen for its optimal isolation between DAPI nuclei, resulting in more accurate identification and quantification of individual cells.

**Protein fluorescence intensity.** The same approach used for the RNA counting was used to measure the fluorescence signal intensity for each protein (tdTomato and the combination of astrocyte markers GS/S100β). To improve the detection of positive cells, we computed the background signal for each cell measurement for each channel. We considered an annular region of 30 pixels (8.5 µm) around the cell mask and measured the fluorescence intensity in this background region. We assigned for each cell a background intensity by computing the minimum background intensity over its three nearest neighbours. This respective background signal was then removed in all protein measurements for each cell.

**Glutamatergic astrocyte identification.** The spatial count matrix for RNA (*Slc17a6*, *Slc17a7*, *Syt1*, *Snap25*) and protein (tdTomato, GS/S100β) was normalized using the CLR method from the Seurat package. UMAP visualization was performed by scaling and reducing the dimensionality of the data using the Seurat standard function. Clustering was processed using the FindClusters function with a resolution of 0.4, and astrocyte clusters were identified on the basis of tdTomato and/or GS/S100β fluorescence expression. This type of cluster was represented by azur ROIs in Fig. 1h. A second round of clustering was performed on the astrocytic cluster using only RNA counts for *Slc17a6*, *Slc17a7*, *Syt1*, *Snap25* transcripts and clusters expressing these transcripts were identified as the glutamatergic astrocyte population. This population is represented by yellow ROIs in Fig. 1h. The different hippocampal regions (DG, CA1, CA2, CA3 and their further subdivisions into DG molecular layer and hilus, or CA1, CA2, CA3 stratum oriens and stratum radiatum) were identified using the Allen brain atlas as reference (https://connectivity.brain-map.org/3d-viewer?v=1).

## Acute brain slice preparations

Acute hippocampal or midbrain slices from transgenic mouse lines or WT mice were prepared and used in patch-seq, two-photon imaging and synaptic electrophysiology experiments. Details of each preparation are provided under the related experimental description.

## Patch-seq analysis of astrocytes from mouse hippocampal DG

Patch-seq procedure (Extended Data Fig. 8a–h) was conducted according to published protocols[43,44,58,59] with minor modifications. In some experiments, the procedure was preceded by glutamate imaging in the same astrocyte (see below). In all other cases, male *GFAP*[creERT2] *tdTomato*[lsl/lsl] mice were treated with TAM (2 i.p. injections per day for 3–5 days), to induce tdTomato fluorescence expression in GFAP-expressing cells[53]. Hippocampal slices from TAM-injected *GFAP*[creERT2]*tdTomato*[lsl/lsl] mice were prepared according to standard procedures. In brief, mice (aged 32–56 days) were anaesthetized with isoflurane and decapitated. The brain was rapidly removed from the skull and immersed in ice-cold oxygenated sucrose-containing ACSF (sucrose-ACSF) with the following composition: 62.5 mM NaCl, 2.5 mM KCl, 7 mM MgCl$_2$, 0.5 mM CaCl$_2$, 25 mM NaHCO$_3$, 1.5 mM NaH$_2$PO$_4$, 10 mM glucose and 105 mM sucrose, saturated with 95% O$_2$–5% CO$_2$ (pH 7.4). Hippocampal horizontal slices (250 µm) were cut with a vibratome (HM 650 V Microm) and then kept in oxygenated standard ACSF: 125 mM NaCl, 25 mM NaHCO$_3$, 1.25 mM NaH$_2$PO$_4$, 3.5 mM KCl, 2 mM CaCl$_2$, 1 mM MgCl$_2$ and 10 mM glucose (osmolality, 295 ± 5 mOsm; pH 7.3–7.4) at 34 °C for at least 30 min. A single slice was then transferred in a recording chamber (perfused with ACSF at 3 ml min⁻¹, 34 °C) placed on the stage of an upright fixed-stage microscope (Olympus BX51WI), equipped for infrared differential interference contrast and epifluorescence video microscopy (Polychrome II, TILL Photonics). To minimize contamination with RNase and RNA degradation, instruments (microscope, manipulators, set-up, computer, puller), benches and all used materials were cleaned daily with RNase-ExitusPlus (PanReac AppliChem, A7153), the intracellular solutions were made in RNase free conditions (UltraPure DNase/RNase-free distilled water (Invitrogen), new powders and decontaminated benches and instruments) and the entire experimental procedure was performed with gloves. Putative astrocytes in the molecular layer of the hippocampal DG (DGML) were selected based on cellular size, morphology and red tdTomato fluorescence in the epifluorescence illumination, and confirmed by electrophysiological measures of resting membrane potential ($V_{rest}$), current–voltage ($I/V$) relationship and input resistance ($R_i$), made with a Multiclamp 700B amplifier using Clampex software and an A/D converter Digidata 1440A (all three from Molecular Devices) connected to a computer. Collection of DGML astrocytes was performed in two consecutive slices (500 µm total thickness), placed at 1,200–1,700 µm in the septo-temporal axis[60]. Before patching, broad-field images of the slices were acquired using a CDD camera controlled by TILL Vision Imago software at ×10 magnification. Patch-clamp recordings of DGML astrocytes were performed using borosilicate glass pipettes (World Precision Instruments, TW120F6) pulled with a Zeitz DMZ Puller (Zeitz-Instruments Vertriebs). Patch pipettes (3–5.5 MΩ) were filled with 1 µl of K-gluconate-based solution supplemented with 1 U per µl of recombinant RNase inhibitor (Takara, 2314A). The K-gluconate-based solution was composed of 130 mM K-gluconate, 4 mM NaCl, 5 mM EGTA, 10 mM HEPES, 1 mM CaCl$_2$, 1 mM MgCl$_2$, 0.2 mM Na-GTP and 2 mM Mg-ATP (pH 7.3). Current signals were filtered at 3 kHz and digitized at 10 kHz. Astrocyte $V_{rest}$, expressed as mV, was measured immediately after the whole-cell configuration using the amplified inbuilt voltmeter in current-clamp mode at 0 pA current. The $I/V$ curve was obtained as the relationship between current amplitudes and hyperpolarizing/depolarizing voltage steps (from −120 mV to +100 mV, 20 mV increment, 1 s). $R_i$, expressed as MΩ, was measured as the slope of a linear regression fit to the $I/V$ curve (Clampit, Molecular devices). After acquisition of the electrophysiological parameters, the intracellular content of the astrocyte was slowly aspirated into the micropipette by applying mild negative pressure. Such procedure was performed under cell visualization at higher magnification (Olympus BX51WI, ×60) while monitoring the integrity of patch pipette–cell seal and cell stability in voltage-clamp mode. The complete

extraction of cell content was detectable as retraction of the cytoplasm and total aspiration of the nucleus, with the patched cell visibly shrunken (Extended Data Fig. 8c). The sample collection procedure was documented in several cases by images of patched astrocytes before and after intracellular content aspiration, and by representative real-time videos (Supplementary Video 2) of the entire sample extraction acquired using an Ultima two-photon laser scanning microscope (Bruker Nano Surfaces Division) (details are provided in the 'Two-photon astrocyte glutamate and Ca$^{2+}$ imaging' section) with a ×60 water-immersion objective lens (Olympus Optical LUMPlan FI/IR). After complete extraction of the intracellular content, the patch pipette was slowly retracted and the pipette content was immediately ejected into a 0.2 ml PCR RNase-free tube (Corning, PCR-02-L-C) containing 9.5 µl of lysis buffer, by applying a small positive pressure, and then gently breaking the tip on the bottom of the tube. Lysis buffer was daily prepared from lysis buffer 10× stock (Clontech SMART-Seq v4 3′ DE Kit, 635040) by dilution with nuclease-free water and the addition of recombinant RNase inhibitor. PCR tubes with cell samples were then stored at −80 °C until further processing and RNA-seq analysis. In experiments in which the patch-seq procedure was preceded by glutamate imaging of the same astrocyte, the protocol was modified as follows: mice were injected at 2.5 months of age with a mixture of *AAV5-hGFAP-SF.iGluSnFR(A184S)* and *AAV5-hGFAP-hM3D(Gq)-mCherry* viruses (see the 'Stereotaxic viral injections' section) and used for the experiment at 5–6 months. Hippocampal slices were prepared as described in the 'Two-photon astrocyte glutamate and Ca$^{2+}$ imaging' section and kept in oxygenated ACSF at 34 °C, containing 118 mM NaCl, 10 mM glucose, 2 mM KCl, 2 mM MgCl$_2$, 1.5 mM CaCl$_2$, 25 mM NaHCO$_3$, 1.2 mM NaH$_2$PO$_4$ and 0.001 mM tetrodotoxin (TTX; Alomone). A single slice was positioned on the stage of the Ultima two-photon laser-scanning microscope with a 20× water immersion objective lens and perfused with ACSF containing a synaptic inhibitor cocktail (details are provided in the 'Two-photon astrocyte glutamate and Ca$^{2+}$ imaging' section). Astrocytes displaying good mCherry fluorescence in the soma and arbour, visualized through a Retiga ELECTRO CCD camera interfaced with PrairieView software, were annotated on-line using the mark stage function in PrairieView and selected for sequential imaging and patch-seq. After switching to two-photon imaging mode at 920 nm for visualizing SF-iGluSnFR signal dynamics, CNO and L-Glut puff protocols were performed as described in the 'Two-photon astrocyte glutamate and Ca$^{2+}$ imaging' section. Once the glutamate imaging protocol (~40 min) was completed, we switched back to the bright-field imaging mode, added a third pipette for the patch-seq (see above) and targeted the cell using the position of the puff pipettes as reference. Procedures for patch-clamp analysis of the astrocyte were performed as reported above, except that protocols for acquisition of electrophysiological parameters and *I*/*V* curve were not performed, starting immediately the cell-content extraction procedure to minimize RNA degradation. Subsequent sample collection, ejection in the PCR tubes and storage were performed as described above. Imaging data analysis was conducted as described in the 'Glutamate image analysis' section. Astrocytes were classified as experimentally validated CNO-responders or non-responders by setting the border between the two groups at the mean − s.e.m. of the response to CNO previously determined in pure imaging experiments (Fig. 2q). Transcriptomic analysis was performed as described in the 'Single-cell RNA analysis' section.

## scRNA-seq

cDNA synthesis and preamplification were performed on cell lysates from patch-seq or combined glutamate imaging/patch-seq experiments using the SMART-Seq v4 3′ DE Kit according to the manufacturer's instructions (Takara). scRNA-seq libraries of the cDNA were prepared using the Nextera XT DNA library prep kit (Illumina). Libraries were multiplexed and sequenced according to the manufacturer's recommendations with paired-end reads using the HiSeq 2500 platform (Illumina) with a high sequencing coverage and an expected depth of 500,000 reads per cell. Each pool contained cells from different collection days and conditions. All scRNA-seq experiments were performed at the Genomics Core Facility of the University of Geneva. The sequenced reads were aligned to the mouse genome (GRCm38) using Star mapper[61]. The number of reads per transcript was calculated using the R function summarize overlaps from the genomic alignment packages[62].

## Single-cell RNA analysis

**Mouse hippocampus database.** We generated a single integrated database for hippocampal cells on the basis of the selection of eight existing databases acquired under different experimental conditions (Extended Data Fig. 1a). We obtained read count matrix through the Gene Expression Omnibus database (GEO), Sequence Read Archive (SRA) or specific web-platforms. We submitted each individual dataset to initial quality control and then used canonical correlation analysis (CCA) to generate the integrated database (Fig. 1a; see below), observing clear overlap between the different datasets. More precisely, we obtained 1,448 cells from GSE106447 corresponding to all hippocampal cells collected in the Artegiani dataset[28]; 2,031 cells from GSE114000 corresponding to all hippocampal cells collected in the Batiuk dataset[19]; 19,710 cells from GSE143758 corresponding to WT hippocampal cells collected in the Habib dataset[23]; 12,686 cells from GSE95753 corresponding to P18, P19, P23, P120 and P132 hippocampal cells collected in the Hochgerner dataset[24]; 23,362 cells from SRP135960 corresponding to P12, P16, P24 and P35 hippocampal cells collected in the Zeisel-1 dataset[25]; 53,204 cells from http://dropviz.org corresponding to all hippocampal cells collected in the "Saunders" dataset[27]; 3,005 cells from GSE60361 corresponding to all hippocampal cells collected in the Zeisel-2 dataset[26]; 89,099 cells from https://portal.brain-map.org/atlases-and-data/rnaseq/mouse-whole-cortex-and-hippocampus-10x corresponding to all hippocampal cells collected in the Yao dataset[29] and finally 91 cells collected from patch-seq experiments and 37 cells collected from combined glutamate imaging/patch-seq experiments. Note that the above individual datasets present a certain variability because each of them contains only a fraction of the total cells present in the native tissue and also because each was obtained under non-equivalent biological and/or methodological conditions (Extended Data Fig. 1a). Thus, their integration, while not abolishing variability, enhances sensitivity in detecting cell populations.

**Human hippocampus database.** We used 9,031 cells from https://www.gtexportal.org/home/datasets corresponding to all hippocampus cells collected in the Habib human dataset[32]; 131,325 cells from GSE160189 corresponding to all hippocampal cells collected in the Ayhan dataset[33]; and 10,268 cells from https://github.com/LieberInstitute/10xPilot_snRNAseq-human corresponding to all hippocampal cells collected in the Tran dataset[34].

**Mouse, macaque and human visual cortex database.** We used 54,242 cells corresponding to P28 and P38 cells collected from mouse visual cortex in the Zipursky dataset under GEO accession number GSE190940 (ref. 63), 133,454 cells from the EMBL-EBI repository under accession number E-MTAB-10459 corresponding to all macaque visual cortex cells collected in the Liu dataset[64]; and 41,541 cells from GSE97930 corresponding to all of the human visual cortex cells collected in the Zhang dataset[65].

**Mouse and human substantia nigra database.** We used 19,975 cells from http://dropviz.org corresponding to all of the substantia nigra mouse cells collected in the Saunders dataset[27]; 6,105 cells from GSM4157078 corresponding to all of the human substantia nigra cells collected in the Agarwal dataset[66] and 40,453 cells from GSE126836 corresponding to all of the substantia nigra cells collected in the Welch dataset[67].

**Cell filtering and quality controls.** To filter only high-quality cells using similar quality control criteria among the different databases, we applied filters on unique molecular identifier (UMI), mitochondrial and genes expressed counts per cell. We first filtered cells on the basis of the percentage of UMIs associated with a maximum of 12% mitochondrial transcripts expression. We further excluded cells with UMI and gene numbers above 3 median absolute deviations (MADs) of the population median with a minimum threshold defined at 200 genes detected. Potential doublets were removed using Scrublet[68] except for patch-seq cells that correspond already to singlets. Finally, we also excluded genes detected in less than five cells. After applying these filters, the cells from mouse hippocampus that were retained for further analysis were: 1,086 from the Artegiani dataset; 1,536 from the Batiuk dataset; 18,693 from the Habib dataset; 9,101 from the Hochgerner dataset; 21,064 from the Zeisel-1 dataset; 49,859 from the Saunders hippocampus dataset; 2,626 from the Zeisel-2 dataset; 78,064 from the Yao dataset; 65 from our patch-seq dataset; and 20 from our combined glutamate imaging/patch-seq dataset. For the human hippocampus, cells retained were: 8,370 from the Habib human dataset; 120,842 from the Ayhan dataset; and 8,907 from the Tran dataset. For the mouse, macaque and human visual cortex, cells retained were: 39,684 from the Zipurski dataset; 123,312 from the Liu dataset and 18,079 from the Zhang dataset. For the mouse and human substantia nigra, cells retained were: 16,526 from the Saunders substantia nigra dataset; 38,498 from the Welch dataset; and 5,257 from the Agarwal dataset.

**Astrocyte predictions.** A deep neural network was used to build a multiclass prediction model. This algorithm, tested on a fraction of pre-annotated data that were not used for training, showed a high accuracy (>98.4%; Extended Data Fig. 1e) and was used to predict the identity of each single cell in the integrated UMAP and to name each cluster, according to the main predicted class (>60% of the cells by cluster; Extended Data Fig. 1d). More precisely, the network was implemented in torch and trained on the Yao hippocampal dataset[29] using subclass_label as the target class names to predict. Classes with <50 cells were removed. The Yao dataset was used as reference as it contained high-quality cells deeply annotated with good sequencing depth covering largely all hippocampal cell types. This dataset was split into a training dataset used for modelling (80%) and a test dataset used for validation (20%). Specifically, we defined a four-layer network architecture using the torch library with 1,024, 512, 256 and 15 num_labels nodes, respectively. Hardtanh was used as activation function of layer-1 and ReLU for layer-2 and layer-3. During training, a 50% dropout rate was introduced into layer-1 input and a 30% dropout rate into the other layers inputs. Furthermore, linear weights of layer-1 were constrained so that their norm were 1 for each node. Layer-1 input normalization was also adjusted according to layer-1 weights for each node. The training was performed on 50 epochs, using a random sampling to correct for class imbalances. A second step of training was performed after pruning layer-1 weights to keep 100 genes with the highest weights on the layer-1 node. The pretrained model was used to validate the test dataset and its performance was rigorously assessed through cross-validation (Extended Data Fig. 1e,f). The prediction and clustering results of the dataset were then consolidated to determine the accuracy, specificity and sensitivity of each class. This was done after each round of dataset removal. Our results demonstrated the robust and consistent performance of the model, with no dependence on the dataset used. We next applied this model to subset only the predicted astrocytes from all of the different hippocampal mouse databases. In total, 16,800 astrocytes were predicted: 216 from the Artegiani dataset; 1,368 from the Batiuk dataset; 2,893 from the Habib dataset; 1,054 from the Hochgerner dataset; 3,718 from the Zeisel-1 dataset; 7,002 from the Saunders hippocampus dataset; 176 from the Zeisel-2 dataset; and 373 from the Yao dataset. On the basis of their genetic fate mapping, physiological and morphological properties, all of the patch-seq cells (85) were considered to be astrocytes.

**Data integration and visualization.** For all of the integration that we performed in this Article, we applied the Seurat CCA data integration procedure to identify shared sources of variation between the different astrocyte databases. CCA is well-suited for identifying anchors when cell types are conserved across datasets. CCA-based integration therefore enables integrative analysis when the experimental condition states induce strong expression shifts. More precisely, for data integration, each dataset was normalized and the 2,000 most variable genes were identified and scaled. We next identified common features and used the FindIntegrationAnchors function with the default parameters (normalization.method = "SCT") followed by the IntegrateData function with the default parameters. For UMAP visualization, integrated data were first scaled and dimensionality reduction was performed using a standard function in Seurat (Fig. 1a,b,d,f and Extended Data Figs. 1b–d,g, 2a,d,f,i, 7c, 8e,f,h and 10a). To identify clusters, we adopted a graph-based clustering approach using the FindClusters function from Seurat with a 0.4 resolution (Fig. 1b). The cell cycle score used in Extended Data Fig. 2h was built using the CellCycleScoring function using the default parameters. The astrocytic score used in Fig. 1d,f was built using the AddModuleScore function based on the following gene list: *Slc1a2*, *Gja1* and *Glul*. The glutamate release score used in Fig. 1d,f was determined on the basis of the following gene list: *Snap25*, *Slc17a7* and *Syt1*.

**Differential expression and GO analysis.** Differentially expressed genes between the nine astrocytic clusters (Fig. 1e, Extended Data Fig. 2c,e,g and Supplementary Table 2) were identified on the basis of their weight in the differential pairwise expression analysis using the Seurat FindAllMarkers function with the default parameters (expect only.pos = TRUE, min.pct = 0.1, logfc.threshold = 0.1). The identified gene candidates for each cluster were interrogated for statistically significant gene ontologies using GSEA[69] (http://software.broadinstitute.org/gsea/index.jsp). As a background gene list for the GO term analysis, we used a total of 11,231 genes corresponding to genes detected in at least 5 cells. For GO enrichment, the top 20 biological processes were filtered using a false-discovery-rate-corrected $P < 0.1$ as a cut-off (Extended Data Fig. 3). We then used general terms of enrichment such as ion transport, regulation of metabolic process, mitochondrial respiratory chain complex I, cell development, cilium and synapse to functionally describe each astrocytic cluster and more secretion-related terms such as exocytosis, calcium-ion-regulated exocytosis, regulation of neurotransmitter secretion and regulation of glutamate secretion to describe cluster 7 (Fig. 1c).

**Astrocytic cluster predictions.** To identify and subset astrocyte populations in the human hippocampus (Fig. 1f) in the mouse, macaque and human visual cortex (Extended Data Fig. 7c) and in the mouse and human substantia nigra (Extended Data Fig. 10a), we first used the same computational approach (astrocyte prediction) as previously done for mouse hippocampus datasets. For human hippocampus, 1,084 astrocytes were predicted from the Habib Human dataset; 10,407 from the Ayhan dataset; and 1,183 astrocytes from the Tran dataset. For the mouse, macaque and human visual cortex, 3,617 astrocytes were predicted from the Zipursky dataset (mouse); 29.025 astrocytes were predicted from the Liu dataset (macaque); and 1,105 astrocytes were predicted from the Zhang dataset (human). For the mouse and human substantia nigra, 944 astrocytes were predicted from the Saunders substantia nigra dataset (mouse); 4,752 from the Welch dataset (human); and 389 from the Agarwal dataset (human). Astrocytes subset from patch-seq and combined glutamate imaging/patch-seq (Extended Data Fig. 8e), human hippocampus (Fig. 1f), mouse, macaque and human visual cortex (Extended Data Fig. 7c) and mouse and human substantia nigra (Extended Data Fig. 10a) were then annotated using the Transfer Data function from Seurat to automatically annotate each cluster on the basis of our mouse hippocampus integrated atlas annotation reference.

**Software packages and versions used for analysis.** The following software packages were used: Seurat v.4, R v.4.0.5; HDF5Array v.1.28.1; rhdf5 v.2.44.0; DelayedArray v.0.26.3; S4Arrays v.1.0.4; patchwork v.1.1.2; reticulate v.1.28; Matrix v.1.5-4.1; cowplot v.1.1.1; ggExtra v.0.10.0; ggplot2 v.3.4.2; dplyr v.1.1.2; wesanderson v.0.3.6; RColorBrewer v.1.1-3; Seurat v.4.9.9.9042; SeuratObject v.4.9.9.9084; bmrm v.4.4; SummarizedExperiment v.1.30.1; Biobase v.2.60.0; GenomicRanges v.1.52.0; GenomeInfoDb v.1.36.0; IRanges v.2.34.0; S4Vectors v.0.38.1; BiocGenerics v.0.46.0; MatrixGenerics v.1.12.0; matrixStats v.0.63.0; and torch v.0.10.0.

## Stereotaxic viral injections

For acute hippocampal slice imaging recordings, male or female C57BL/6JRj WT mice (Janvier) $Slc17a7^{fl/fl}$ mice (same genetic background) and $GLAST^{creERT2}P2ry1^{fl/fl}$ mice[39] aged 2–3 months were anaesthetized by isoflurane inhalation (4% induction, 1% maintenance) and positioned within a stereotaxic frame (Stoelting model 51500). Mouse temperature was maintained at 37 °C by a heat pad. All surgeries were performed according to protocols approved by the Cantonal Veterinary Office of Vaud (see above) in accordance with Swiss federal guidelines. Mouse eyes were maintained hydrated by gel artificial tears (Viscotears, Novartis). Fur around the scalp area was removed using depilatory cream. The skin was sterilized with betadine and a mix of lidocaine with epinephrine (6 mg per kg) and carprofen (5 mg per kg) was administered as local analgesic/anaesthesia 5 min before cutting the skin. A single midline anteroposterior scalp incision was made to expose the skull and a burr hole was drilled through the skull above the CA1 region of hippocampus (medial/lateral (ML): ±1.5 mm; anterior/posterior (AP): −2.3 mm; dorsal/ventral (DV): −2.3/−1.8 mm). In experiments performed in the visual cortex, the following procedure and coordinates were used (ML: ±1.5 mm; AP: −2.7 mm; DV: −2.2/−2.5 mm with a 50° angle). Injection of a single AAV virus or a mixture of AAV viruses (800 nl) for each spot was made at 150 nl min⁻¹ using a pulled glass pipette (tip diameter of approximately 50 µm) left in place for 5–10 min after completion of viral infusion to allow viral spreading. The skin was then sutured using prolene suture monofilament (ethicon). For astrocyte glutamate-release imaging experiments, we injected the following viral cocktails containing: (1) a mixture (1:1) of adeno-associated viruses (AAVs) ssAAV5-hGFAP-SF_iGluSnFR(A184S)-WPRE-bGHpA (*AAV5-hGFAP-SF.iGluSnFR(A184S)*, 7.3 × 10¹² vg per ml) and AAV5-hGFAP-hM3D(Gq)-mCherry-WPRE-hGHpA (*AAV5-hGFAP-hM3D(Gq)-mCherry*, 6.0 × 10¹² vg per ml) in Fig. 2b–d,v–x and Extended Data Figs. 4a,g–i, 5k, 7a,b and 8a,g,h; (2) a mixture of *AAV5-hGFAP-SF-iGluSnFR(A184S)* and ssAAV5-hGFAP-mCherry-WPRE-hGHpA (*AAV5-hGFAP-mCherry*, 6.1 × 10¹² vg per ml) as a control in Extended Data Fig. 5l; (3) a mixture (1:1:1) of *AAV5-hGFAP-SF.iGluSnFR(A184S)*, *AAV5-hGFAP-hM3D(Gq)-mCherry* and ssAAV5-hGFAP-eBFP2_iCre-WPRE-hGHpA (*AAV5-hGFAP-eBFP2-iCre*, 8.0 × 10¹² vg per ml) in Fig. 2j–l and Extended Data Fig. 4d,m–o, (4) a mixture (1:1) of *AAV5-hGFAP-SF.iGluSnFR(A184S)* and *AAV5-hGFAP-mCherry-iCre* virus in Fig. 2m–o and Extended Data Fig. 4e; (5) AAV ssAAV5-hGFAP-SF_iGluSnFR(A184S)-WPRE-bGHpA (*AAV5-hGFAP-SF.iGluSnFR(A184S)* alone in Fig. 2f–h,s–u and Extended Data Figs. 4b,j–l and 7d–g; and (6) AAV ssAAV5-hGFAP-eBFP2_iCre-WPRE-hGHpA (*AAV5-hGFAP-eBFP2-iCre*, 8.0 × 10¹² vg per ml) alone in Extended Data Fig. 4c. For astrocyte Ca²⁺ imaging experiments, we injected *AAV2/5 pZac2.1 gfaABC1D-cyto-GCaMP6f (*short name *AAV5-hGFAP::cytoGCaMP6f*, 4.1 × 10¹³ vg per ml) together with ssAAV5-hGFAP-mCherry-WPRE-hGHpA (*AAV5-hGFAP-mCherry*, 6.1 × 10¹² vg per ml) at a 2:1 mixture or with ssAAV-5/2-hGFAP-mCherry_iCre-WPRE-hGHp(A) (*AAV5-hGFAP-mCherry-iCre*, nuclear, 6.6 × 10¹² vg per ml) at a 2:1 mixture in Extended Data Fig. 5i,j. All viral constructs were provided by the Viral Vector Facility, central technology platform of ETH-Zürich.

## Two-photon astrocyte glutamate and Ca²⁺ imaging

**In situ experiments.** Astrocyte glutamate imaging experiments were performed in the medial DGML of 4–5-month-old WT (C57BL/6JRj), $Slc17a7^{fl/fl}$ and $GLAST^{creERT2}P2ry1^{fl/fl}$ mice 6–8 weeks after viral injection (see above). Mice were anaesthetized with isoflurane and decapitated. The brain was removed and quickly placed in ice-cold slicing solution containing 204.5 mM sucrose, 10 mM glucose, 2 mM KCl, 1.2 mM NaH₂PO₄, 25 mM NaHCO₃, 0.5 mM CaCl₂ and 7 mM MgCl₂; the pH was equilibrated with a 5%/95% CO₂/O₂ (Carbogen) gas mix. Horizontal hemibrain slices (thickness, 300 µm) were sectioned and placed into a 34 °C ACSF solution, containing 118 mM NaCl, 10 mM glucose, 2 mM KCl, 2 mM MgCl₂, 1.5 mM CaCl₂, 25 mM NaHCO₃, 1.2 mM NaH₂PO₄ and 0.001 mM TTX (Alomone). After 30 min recovery at 34 °C, slices were maintained at room temperature[70] and used for two-photon imaging for the next 3 h. Slices were placed into a recording chamber perfused (2 ml min⁻¹) with Carbogen-bubbled ACSF containing 120 mM NaCl, 10 mM glucose, 2 mM KCl, 2 mM MgCl₂, 2 mM CaCl₂, 25 mM NaHCO₃ and 1.2 mM NaH₂PO₄. To minimize neuronal glutamate release arising from action potential-evoked or spontaneous/miniature activity, a pharmacological synaptic blocker cocktail consisting of TTX (1 µM), the P/Q-type (ω-Agatoxin IVA; 150 nM), N-type (ω-Conotoxin GVIA; 500 nM) and R-type (SNX-482; 100 nm) voltage-gated calcium channel antagonists was added to ACSF together with antagonists for AMPA (NBQX; 100 µM) and NMDA (MK801; 100 µM) receptors to further suppress neuronal excitability and presynaptic modulation of glutamate release[71,72]. In parallel with this pharmacologic synaptic inhibition, in chemogenetic stimulation experiments (Fig. 2a,c,k), we used *AAV5-hGFAP-hM3D(Gq)-mCherry* virus-mediated astrocyte expression of $G_q$-DREADD to selectively stimulate astrocytes and putatively induce their glutamate release after local delivery of the designer drug CNO[73,74] (100 µM). In experiments with endogenous $G_q$-GPCR stimulation, we used WT mice and locally delivered the selective agonist of purinergic P2Y1 receptors, 2-methyl-thio-adenosine-5′-diphosphate trisodium salt (2MeSADP; 10 µM; Tocris). For detection of extracellular glutamate, we used *AAV5-hGFAP-SF.iGluSnFR(A184S)* virus-mediated expression of next-generation superfolder GFP (SF-iGluSnFR)[22] at the astrocyte surface (Fig. 2a,b). This glutamate sniffer version has an improved signal-to-noise ratio (SNR) and an alanine 184 substitution to serine (A184S) that provides greater glutamate affinity ($EC_{50}$ = 0.6 µM) and slower off-rates (450 ms) compared with versions used in previous astrocyte studies[3,17]. Two-photon imaging was performed as described in our previous studies[36,37] using the Ultima two-photon laser scanning microscope (Bruker Nano Surfaces Division) consisting of an Olympus BX61WI-equipped resonant scanning system, two highly sensitive GaAsP detectors and a multi-alkali detector, and a high-numerical-aperture (NA = 1.0) long-working-distance ×20 water-immersion objective lens (Olympus N20X-PFH XLUMPLFLN). The light source was a Chameleon Vision II Ti:Sa laser, with 140 fs pulse duration, tuned to 920 nm for SF-iGluSnFR imaging. The laser power was modulated electro-optically using a Pockels cell (Conoptics 302 RM). The two-photon imaging system was run using the Prairie View software. Fluorescent emission from the sample was passed through a 660LP dichroic mirror and directed to a 495 long-pass filter. This long-pass filter reflected wavelengths shorter than 495 nm to a 450/50 band-pass filter before a GaAsP detector, which allowed blue visualization. Wavelengths higher than 495 nm were split by a filter-cube set containing a 560LPXR beam splitter with a 520/540 band-pass filter attached to a GaAsP detector (to visualize green), and a 610/675 band-pass filter attached to a multi-alkali detector (to visualize red). Laser-power was measured using a power meter (Melles Griot, 13PEM001) and determined to be 6–8 mW at the sample level. All recordings were acquired at >50 µm below the brain-slice surface. For SF-iGluSnFR signal detection, we used conditions based on previous reports[75–77], but refined in terms of acquisition-speed (33 Hz), optical-zoom (×16), frame-average (4×), pixel-size (0.293 µm) and size of the FOV (37.3 µm × 37.3 µm), constituting around one-third of the total area of a typical DG astrocyte[37], thereby aiming to be capable of detecting small/fast SF-iGluSnFR signals that could be otherwise missed by too

slow acquisition speeds, too large FOVs or because of large contribution of unbound iGluSnFR to $F_0$ (refs. 70,78). To select the imaging FOV, we first visualized at low optical zoom (×2) and at 720 nm medial DGML sites containing multiple astrocytes with hM3D(Gq)-mCherry red signal visible throughout their structure. We then switched to 920 nm to confirm the presence in the same cells of the SF-iGluSNFR green signal. To ensure consistency of the FOV location across acquisitions, we used a common spatial orientation, in which the astrocyte occupies most of the FOV, positioning the soma around the centre/side of the lower half of the FOV and its arbor mostly above (Fig. 2a (middle)). In experiments using slices from *Slc17a7*[fl/fl] mice injected with *AAV5-hGFAP-eBFP2-iCre* virus, or from *GLAST*[creERT2]*P2ry1*[fl/fl] mice injected with *AAV5-hGFAP-mCherry-iCre* virus, to induce *cre* recombination selectively in astrocytes, reported by nuclear blue eBFP2 or red mCherry, we first targeted astrocytes with small blue punctate structures visible at 750 nm (ref. 79) in the nucleus (or with nuclear mCherry expression at 720 nm) and then checked that the same cells co-expressed in their soma and arbor SF-iGluSnFR at 920 nm (and hM3D(Gq)-mCherry, in the case of *Slc17a7*[fl/fl] mice). As controls for the latter experiments, we also checked in WT mice with triple virus injection for a lack of any fluorescent signal alterations and an unchanged ability to respond to CNO stimulations compared with uninjected mice (responders, 2 out of 5 tested cells; CNO responding area, 16 ± 3.2% of the L-Glut-responsive FOV; compare with Fig. 2p,q). Delivery of CNO, 2MeSADP and other drugs to astrocytes was performed locally in the FOV through time-controlled puffs from patch pipettes. For this, pipettes were pulled to a resistance of ~5 MΩ using the DMZ-Universal-Electrode-Puller. All pipettes were filled with a vehicle solution containing the red-fluorescent dye Alexa-594 (dissolved in ACSF and DMSO) to visualize the temporal-spatial features of drug delivery in the FOV[6]. A vehicle solution was then added of either CNO (100 µM) or 2MeSADP (10 µM) or L-glutamic acid (L-glut; 1 mM). In some experiments, the vehicle solution was used alone as a control to exclude potential pressure-dependent effects. For imaging experiments, we used a dual-pipette set-up in which micro-injection occurred through a Pneumatic Pico-Pump (PV820; WPI) that was controlled by voltage commands issued through a pulse generator (A.M.P.I. MASTER-8). Before positioning both pipettes at the tissue level, they were visualized ~2.5 mm above at 720 nm using Dodt gradient contrast, and red multi-alkali PMTs to adjust the positive pressure required to prevent back-flow, or non-controlled outflow of drug solution. Pipettes were then lowered to the tissue level and positioned at the left and right edges of the recording FOV just before acquisition (Fig. 2a (middle)). Each experimental recording session for an astrocyte's FOV consisted of a first 2 min (4,000 *t*-frames) acquisition period in which the CNO (or vehicle solution as control) was pressure-ejected from the pipette (~100 mbar; 10 ms) onto the astrocyte six times at an interevent interval of 20 s to stimulate astrocyte signalling (Fig. 2a (right) and Extended Data Fig. 5e). Next, a second 2 min acquisition was undertaken, in which L-glut was pressure ejected also six times at an interevent interval of 20 s, to identify functional SF-iGluSnFr-expressing sites within the same FOV. This repeated applications protocol was devised to verify the spatial-temporal reliability of the iGluSnFR responses time-locked to drug stimulation, and to evaluate the endogenous signal during baseline periods. In experiments in which imaging was followed by patch-seq, the protocol was slightly modified (see the 'Patch-seq analysis of astrocytes from mouse hippocampal DG' section). In a few experiments, we visualized CNO-evoked Ca[2+] responses in *GFAP*[creERT2]*GCaMP6f* mice treated with TAM as previously described[37], and virally injected with *AAV5-hGFAP-hM3D(Gq)-mCherry*. In this case, FOVs comprised whole individual astrocytes, and we used a single CNO stimulation protocol through local puff as above. In a few other experiments, we compared Ca[2+] responses in the medial DGML of *Slc17a7*[fl/fl] mice injected with *AAV5-hGFAP-GCaMP6f* Ca[2+] indicator and either *AAV5-hGFAP-mCherry* (controls) or *AAV5-hGFAP-mCherry-iCre* (VGLUT1[GFAP-KO]) viruses during

medial perforant pathway (MPP) electrical stimulation periods (either single stimuli or Θ-LTP protocols; for details see the 'LTP of excitatory synapses in the hippocampal DG' section below. Mice were virally injected at 2 months of age and used experimentally at 4 months of age. Two-photon imaging was performed in the same system as above, but using the galvo mode at 0.3 Hz frame-rate and a low optical zoom (×1.5) to simultaneously monitor responses in multiple astrocytes. The laser was tuned to 920 nm for GCaMP6f imaging and at 1,000 nm for mCherry imaging.

**In vivo experiments.** WT mice (aged 2–3 months) were injected in the primary visual cortex with *AAV5-hGFAP-SF.iGluSnFR(A184S)* virus or a mixture of *AAV5-hGFAP-SF.iGluSnFR(A184S)* and *AAV5-hGFAP-hM3D(Gq)-mCherry* viruses (see the 'Stereotaxic viral injections' section) and, after 4 weeks, prepared for awake in vivo two-photon imaging, including attachment to a metal bar allowing for head fixation. The mice were next habituated to the set-up and, during three sessions, were trained for being head-fixated in a tube below the microscope objective. On the day of the experiment, an acute cranial window allowing local puff of drug microvolumes was surgically opened above the primary visual cortex. This was done under 1.5% isoflurane anaesthesia supplemented with a carprofen injection (5 mg per kg, subcutaneous) and local anaesthesia in the form of lidocain (0.2%, subcutaneous) under the scalp. During the surgery, the animals were kept warm on a temperature-controlled heat blanket and their eyes were protected against dehydration with visco-tears. A circular hole (diameter, 3 mm) was drilled into the bone centred over the visual cortex (2 mm laterally, 3.5 mm behind bregma) and the dura was carefully removed under ice-cold, freshly made, ACSF. 2% agarose was then added in a thin layer beneath a glass coverslip (thickness #1), which was fastened with additional agarose to form a soft-walled well. In this well, ACSF was placed to keep the agarose moist throughout the experiment. With the exception of experiments in which the hM3D(Gq)-mCherry construct was used, astrocytes were labelled by a tail-vein injection of sulforhodamine 101 (SR101, 10 mg ml[−1] in 0.9% NaCl sterile solution, 100 µl bolus i.v.) 1 h before imaging. The anaesthetized mouse was brought to the microscope, placed in the tube in which it had been previously habituated to stay and was allowed to wake up after head fixation. The exposed visual cortex was then imaged during the next 2–3 h. The FOVs were either small (37.3 µm × 37.3 µm; acquisition-speed, 33 Hz; optical-zoom, ×16; pixel-size, 0.293 µm) containing just part of one astrocyte, similar to the experiments in situ (see above), or large (151 µm × 151 µm; acquisition-speed, 33 Hz; frame-average, 4×; pixel-size, 1.18 µm) with up to nine astrocytes in focus. These FOVs were 100–200 µm below the surface and were imaged for 1 min periods with 2–4 min breaks in between acquisitions depending on the mouse behaviour (Fig. 2r–y and Extended Data Fig. 7d–g). Two-photon imaging was performed using the Bruker in vivo Investigator system (Bruker Nano Surfaces Division) equipped with an 8 kHz resonant galvanometer scanner, coupled to a MaiTai eHP DS laser (Spectra-physics, Milpitas) with a 70 fs pulse duration, tuned to 920 nm. Negative dispersion was optimized for each wavelength, and the laser power was rapidly modulated by Pockels cells. A ×20 LUMPFL60X W/IR-2 NA 0.9 Olympus objective was used. Emission was separated by a dichroic beam splitter (t560lpxr) and passed through either an et520/540m-2p (for red) or an et610/675m-2p (for green) emission filter, before reaching the GaAsP detectors. Their negative dispersion allowed for minimal laser dose applied to the tissue. The laser power varied during experiments depending on the depth of the focus but was kept below 7 mW and measured continuously with a power meter. Experiments were generally in two parts. The first part started with imaging spontaneous SF-iGluSnFR activity in small or large FOVs, containing one or multiple astrocytes respectively, followed by incubation (40 min) with a synaptic blocker mixture that is known to eliminate neuronal activity in acute cranial windows[41] (topical solution in ACSF of 200 µM NBQX, 300 µM MK801, 20 µM TTX, 1.5 µM

Ω-agatoxin IVA, 5 μM Ω-conotoxin GVIA, 1 μM SNX-482). During this period, the mouse was put under light anaesthesia (<1% isoflurane) and, at the end, was allowed to wake up completely before a second imaging round was performed in the same FOV as before synaptic blockers and during the same variations in physical activity. The effect of the blocker cocktail was evident from direct visual examination and confirmed by post hoc quantification (Extended Data Fig. 7d–f). In the following part of the experiments, the effect of local applications of specific agents (CNO (100 μM–1 mM) or Ach (10–50 mM), both in ACSF containing 25 nM Alexa Fluor 594) during imaging was tested in small FOVs. An electrode with 0.1 Ω resistance was filled with the agent solution, carefully inserted into the cranial window and moved to the FOV under low magnification. Release of microvolumes of solution in the glass pipette was then performed in a timed manner using air pressure controlled by a pneumatic PicoPump (PV820; WPI). The protocol involved 5 puffs (10–50 ms, 15–50 psi) spaced by 10 s intervals to allow tissue diffusion of the agent and avoid build-up of tissue pressure. Effective delivery was confirmed by the appearance of Alexa Fluor 594 red fluorescence in the FOV. In some cases, pipette clogging during the experiment required a temporary change of the picopump settings until the red fluorescence appeared in the tissue around the pipette tip. The start time of exposure to the agent in these cases was assigned to the puff on which Alexa Fluor 594 fluorescence first appeared in the tissue.

### Glutamate image analysis

**In situ experiments.** We first developed an analytical pipeline called AstroGlu as an application program interface within a Python v.3.7.6 virtual environment (venv) running Jupyter Lab/Notebook (Anaconda; Jupyterhub v.1.0.0) on an Ubuntu v.18.04.4 server (CPU, 48 cores; RAM, 1 TB; storage, 2 TB solid-state driver; GPU, NVIDIA Quadro P5000). The application program interface allowed remote access of the analytical pipeline through the browser (in a Jupyter notebook v.6.4.12) using simplified Python code to customize parameters and sequence of the pipeline's software library modules for (1) file loading/export; (2) visualization (as time series, projections or videos); (3) pre-processing; (4) signal extraction; (5) peak detection and analysis. In the process of preparing the AstroGlu pipeline for release, we also tested its functionality using a Python venv assigned to a node with 24 cores and 128 GB RAM. The pipeline runtime was 10–15 min. To simplify installation and accessibility for users, the AstroGlu source code was converted to a Dockerized image, which provided the same Jupyter front-end used on the server/cluster, to run the analytical pipeline. The Docker image of AstroGlu was tested on the above cluster with varying node configurations for CPU cores and RAM. We determined that the minimum resource allocation capable of reliably completing all 20 steps of the pipeline without kernel crash was a node with 6 cores and 16 GB RAM (1 h runtime). However, conventional Windows or Mac systems with these specs were unable to reliably run the pipeline. We therefore suggest a minimal requirement of 64 GB RAM and 8–12 CPU cores to reliably run the full AstroGlu pipeline, preferably in conditions in which all or most resources can be allocated to running the pipeline (that is, cluster node). Raw acquisitions were imported into the AstroGlu pipeline as 3D (2D + $t$) NumPy v.1.19.5 arrays. Raw acquisitions were explored as interactive time-series plots that displayed $xy$ values corresponding to the location of a hovering mouse cursor. Acquisitions were also explored as videos using a viewing tool with slider that allowed manual advance of $xy$ frames in time; this tool also allowed side-by-side comparison of the output of preprocessing steps in a frame-by-frame manner. Both interactive tools were adapted from the Bokeh v.1.3.4 library. Raw acquisitions were de-noised frame by frame using the Scikit (v.1.2.0; https://scikit-image.org/docs/dev/auto_examples/filters/plot_nonlocal_means.html) implementation of a feature-preserving non-local means (NLM) filter[80,81]. NLM filter parameters—patch size ($s = 2$), patch distance ($W = 4$) and smoothing factor ($h = 0.8$)—were selected on the basis of ref. 82 and were further adjusted to values that

consistently provided the greatest improvement to the calculated SNR. We selected NLM over Jupyter implementations of noise-2-void[83] or anisotropic diffusion filters[84] because it provided the best result between SNR enhancement and raw signal dynamics preservation. Denoised recordings were then convolved in space and time using a Gaussian filter with a small kernel ($\sigma X = 0.5$, $\sigma Y = 0.5$, $\sigma T = 0.5$) to stabilize frame-to-frame pixel fluctuations, thereby improving the SNR without affecting signal dynamics. The impact of the Gaussian filter on signal dynamics was assessed using the interactive time-series plot and viewing tool. The denoised recordings were used to calculate $\Delta F/F_0$. The relative change in fluorescence was computed frame-by-frame using the mean of the entire acquisition in $xy$ as the baseline and in turn expressed as $z$-scores to normalize variance across time-frames[85,86]. The same preprocessing pipeline was applied to the Alexa-594 red fluorescence signal associated with the drug puff. For extraction and quantification of SF-iGluSnFR signals in response to drug applications (CNO, 2MeSADP and L-Glut), we had to initially consider the lack of any spatial information about the origin of the signals and the underlying structure in astrocytes. We decided to use an agnostic grid-based analysis[87] and sample the SF-iGluSnFR signal throughout the FOV. For this, we developed a Python-based interactive grid using HoloViews (v.1.15.4; https://holoviews.org/getting_started/Gridded_Datasets.html), and the Bokeh (https://docs.bokeh.org/en/latest/index.html) library, which allowed visualization of overlaid time-locked epochs within a given FOV. We used a 1.13 μm × 1.13 μm grid size (1,024 grid spaces/FOV), with a spatial resolution as in our previous work with GCaMP6f [37] and an automated detection strategy using the open-source Scipy.Signal (v.1.10.0; https://docs.scipy.org/doc/scipy/reference/signal.html) analysis package and Neurokit2 (v.0.1.6)[88]. We conducted our analysis at each grid location, focusing on a set of six epochs in which the $z$-scored SF-iGluSnFR signal in the frames preceding the stimulus onset (puff of a drug) was used as a local baseline and compared to the $z$-scored signal in the frames after stimulus onset for peak detection. To demarcate the onset time for drug delivery to the FOV, and define this as the epoch onset time, we used a $1z$ threshold change in Alexa-594 fluorescence intensity. Considering that the average rise time of CNO- and 2MeSADP-evoked SF-iGluSnFR events was ~100 ms, we applied an analytical window of 240 ms (8 frames at 33 Hz) before the stimulus onset (baseline) and a peak-detection window of 240 ms after the stimulus onset. In the case of responses to L-Glut, the epoch window was extended to 2 s to ensure full detection of the L-Glut-related SF-iGluSnFR signal, which often showed more than one peak. Each individual grid location in which the iGluSnFR signal reached ≥$2z$ within 240 ms from stimulus onset was considered to be a responding location and scored. The $2z$ threshold was selected as it was the lowest value across all acquisitions that did not pick up noise and corresponded to a statistically significant threshold[89]. The ensemble of the responses in individual grid locations was converted to an array, representing a functional map of the locations responding to the stimulus in the FOV. For optimized visual display (Fig. 2 and Extended Data Figs. 4 and 5), we used a non-ROI-based analysis of the SF-iGluSnFR signal at the highest $xy$ resolution available (0.293 μm per pixel) instead of the grid-based analysis and generated projections using the final step of the preprocessing pipeline ($z$-score) as an input. The $z$-scored projections corresponding to CNO or 2MeSADP applications were generated as mean projections of the 240 ms before and after the drug puff for each of the 6 epochs. For L-Glut applications, mean projections represented the 2 s before and after drug puffs. The turbo colormap from matplotlib was used to assign amplitude values. A single s.d. projection was used to represent fluorescence intensity variance across CNO and 2MeSADP applications, which enabled the visualization of regions within the FOV capable of repetitive SF-iGluSnFR responses. The s.d. projection was generated by using as input a masked acquisition excluding all $xyt$ frames except the $xy$ frames corresponding to $t - 240$ ms and $t + 240$ ms ($t - 2$ s and $t + 2$ s for L-Glut) for all 6 epochs. Mean and s.d. projections were also used to validate the quality of

the grid-based analysis, by comparing the shape of SF-iGluSnFR responses at the highest $xy$ resolution to their shape after spatial downsampling used in grid-based analysis. The reliability of the responding locations was determined by counting the number of times that each grid location showed a response to the stimulus during the six-application protocol and by creating a colour-coded map of the FOV with a scale going from 0/6 (never responding) to 6/6 (always responding) for each grid location (Extended Data Fig. 5e). To better discriminate real biological responses from possible artifacts, we adopted restrictive criteria and, in each recording, we selected for further analysis only grid locations that (1) reliably received CNO (or 2MeSADP) and L-Glut puffs across all six stimulations (selection on the basis of ≥2$z$ peak Alexa-594 fluorescence responses); (2) exhibited repeated SF-iGluSnFR responses to CNO (or 2MeSADP) and L-Glut (selection on the basis of ≥4 suprathreshold responses to 6 drug puffs); and (3) did not display evident local motion artifacts. The latter, rarely induced by the puff, had characteristic features (displacement of several pixels in $x$ and y time-locked and anti-correlated to the increase of the Alexa signal) enabling us to recognize them and exclude them from the analysis. We implemented these criteria using matrix analysis through NumPy and by generating 2D arrays (maps) of Alexa-594 fluorescence responses for each series of six CNO (or 2MeSADP) and L-Glut applications to an individual astrocyte, and 2D arrays of the corresponding SF-iGluSnFR fluorescence responses to CNO (or 2MeSADP) and L-Glut. From these 2D arrays, we generated binarized arrays, that is, binarized functional maps of the responses, containing (1) only those grid locations with ≥4 responses to L-Glut and positive for Alexa signal; (2) only those grid locations with ≥4 responses to CNO (or 2MeSADP) and positive for Alexa signal; and (3) an integrated binary array composed only of the selected (reliable) CNO (or 2MeSADP) response locations that localized to reliable L-Glut response locations (Extended Data Figs. 5f–h,k,l, 6a,c,e,g and 8g). The number of grid locations with suprathreshold CNO-evoked (or 2MeSADP-evoked) SF-iGluSnFR responses was counted and normalized to the number of L-Glut responsive grid locations and pooled into group data. To identify discrete clusters of suprathreshold recurrently active grid locations in response to CNO or 2MeSADP (hotspots) and calculate their area, we used skimage.morphology.label (https://scikit-image.org/docs/dev/api/skimage.morphology.html#skimage.morphology.label) in conjunction with numpy.unique (https://numpy.org/doc/stable/reference/generated/numpy.unique.html). We set to 4 the minimal number of connected active grid locations defining a cluster, as this is the range of $xy$ spatial spread of the SF-iGluSnFR signal reporting glutamate release from a single synaptic bouton[90,91]. We classified astrocytes as responders, that is, releasing glutamate in response to $G_q$-DREADD or endogenous P2Y1R activation, by setting a theoretical threshold to 5% of their FOV exhibiting enhanced SF-iGluSnFR signal to CNO or 2MeSADP, respectively. In a few experiments on $Ca^{2+}$ responses to CNO in $G_q$-DREADD-expressing astrocytes, we analysed GCaMP6f $Ca^{2+}$ signals in FOVs representing entire astrocytes as done previously[37]. In experiments evaluating astrocyte $Ca^{2+}$ dynamics during MPP stimulations, we analysed FOVs containing several astrocytes in different stimulation conditions. For this, we generated mean time projections of the GCaMP6f signal over 70–80 frames (21–24 s) representing periods comprising either (1) spontaneous astrocyte $Ca^{2+}$ activity with responses to a single stimulus or (2) peak astrocyte $Ca^{2+}$ responses to $\Theta$-LTP stimulation. Corresponding $Ca^{2+}$ traces were extracted from representative single-astrocyte ROIs set in ImageJ and compared between astrocytes from control mice and from VGLUT1[GFAP-KO] mice.

**In vivo experiments.** Two-photon images from awake mice were analysed using a custom MATLAB-based code (2019b version) and run through two separate scripts in consecutive order. The first one (Plotting_SnFrActivity_2D) was used to handle the images and define ROIs; the second (SnFrActivity_PeaksinTraces) was used to evaluate average activities from ROIs. MATLAB was installed on a PC with 16 GB RAM and a 2.71 GHz processor and a 64 bit system. In these conditions, the analysis would take 5 min in total from loading the image stack to saving the final results from an acquisition. Individual acquisition periods (33 Hz frame rate) lasted 60 s. Movements during each imaging period were detected to enable image stabilization. Periods contaminated by large movements in which the imaged FOV could not be contained despite the image stabilization procedure (see below) were discarded. To identify movements, the morphological signal in the red channel was used. This was given by SR101 loaded through tail vein injections before imaging, or by mCherry in $G_q$-DREADD-expressing mice. Each red channel image was normalized, smoothed using a Gaussian filter ($\sigma = 3$) (imgaussfilt.mat) and a $1 - 1.5\ F/F_{mean}$ threshold was imposed to set low-intensity regions to zero. Thereby, a stack of high-contrast images was obtained and was used to stabilize the original images. The latter were compared to the max projection of the average of the entire stack (normxcorr2.mat). In this way, the amount of movement in pixels along the $x$ and y dimensions of each original image could be quantified and the image positions regulated relative to the average position. Images in which the structure moved >5 pixels (large FOVs, 1.18 μm per pixel; small FOVs, 0.29 μm per pixel) from the average position of the stack were disregarded, excluding even tiny movements, therefore limiting the possibility that mouse movements influenced the results. The stable SF-iGluSnFR images were then averaged and normalized to the s.d. around the mean ($z$-score) per pixel over the entire duration of the acquisition. This approach enabled us to next identify in an unbiased manner potentially active ROIs in the FOV. These ROIs were formed by grouping the sets of contiguous pixels that displayed the largest increases in fluorescence amplitude over the acquisition. A minimum of 5 pixels per ROI was imposed based on biological considerations (see the 'In situ experiments' section) and susceptibility to movement artifacts. A fixed number of 50 ROIs per acquisition, representing those with highest fluorescence changes, were evaluated for each FOV. The average SF-iGluSnFR fluorescence signal within each ROI was then measured, normalized to $z$-score values and filtered using a pass filter excluding fluctuations lasting >2 s or <150 ms (idealfilter.mat), based on the expected kinetics of astrocyte SF-iGluSnFR signals from our in situ studies (Fig. 2e,i and Extended Data Fig. 6b,d,f). SF-iGluSnFR fluorescence signals >2$z$ of the average fluorescence and lasting >250 ms (FWHM) were detected (findpeaks.mat) and quantified in terms of SF-iGluSnFR peak frequency. In the 60 s imaging acquisitions involving application of Ach, CNO or ACSF, the first 10 s was the baseline period. This was followed by the stimulation period (always 5 puffs at 10 s interval, but of which the official start was set each time when the first puff successfully released the studied agent, as confirmed by visibility of the Alexa Fluor 594 dye in the FOV). Within each ROI, the peak frequency was compared between the baseline period and the stimulation period. The ROI was considered to be responsive to the stimulus when its peak frequency increased by >25% after stimulus administration. An astrocyte was considered to be responsive to the stimulus when (1) the total area of its responsive ROIs covered ≥3% of the FOV; and (2) the mean frequency of all of its ROIs (responsive and not) was statistically higher in the stimulation period compared with the baseline period (two-sided Wilcoxon rank-sum test, $P < 0.05$; signrank.mat). The final statistical value expressing the difference between the before and after stimulation periods was calculated using the same Wilcoxon test by comparing the 10 s period of maximal stimulus effect with the 10 s baseline period. In some experiments, repetition of the drug administration protocol after a washout period (≥2 min) allowed us to compare the pattern of stimulus-responsive ROIs in sequential acquisitions. Tentative identification of hotspots of astrocyte glutamate release was done considering the responsive ROIs in each acquisition. These were slightly enlarged by a weak gaussian blur ($\sigma = 1$) and overlapping pixels between them in the two acquisitions identified the hotspot responding region. As the number of

responding ROIs fluctuated between acquisitions, the lowest number of responding ROIs in either of the two acquisitions was used to estimate the percentage of hotspots/responding ROIs. In experiments in which the effect of a synaptic blocker mixture was assessed on spontaneous SF-iGluSnFR signals in the awake mouse, the average SF-iGluSnFR peak frequency from all ROIs in a FOV (small or large) was compared during 60 s periods before and after application of the mixture and statistical difference tested with the two-sided Wilcoxon rank-sum test, $P < 0.05$ (signrank.mat). In this case, the relevant temporal window for peak detection was set between 500 ms and 10 s owing to the presence in the pre-blocker condition of slower and longer lasting signals reflecting inherent coordinated cortical activity in the awake mouse[92] that were abolished in the post-blocker period, when the drug stimulations were performed.

### Fibre photometry measures of astrocyte SF-iGluSnFR signal

Adult mice (C57BL/6, aged 2 months) were virally injected with a mixture (1:1) of *AAV5-hGFAP-SF.iGluSnFR(A184S)* and *AAV5-hGFAP-hM3D(Gq)-mCherry* viruses in the hippocampus (ML: ±1.5 mm; AP: −2.3 mm; DV: −2.3/−1.8 mm) as described in the 'Stereotaxic viral injections' section. Then, 2 weeks after the viral injection, a single fibre probe coupled with an injection cannula (200 µm, Doric Lenses) was inserted (at a constant speed of 7 µm s⁻¹) 150 µm above the viral injection site (ML: ±1.5 mm; AP: −2.3 mm; DV: −1.7 mm;) and secured with C&B Metabond (Parkell). A head bar was concomitantly fixed behind the implant. In detail, mice were anaesthetized with isoflurane (Univentor; induction, 2%; maintenance, 1–1.5%) and placed into the stereotaxic apparatus (Kopf). The ocular protector Viscotear was used to prevent eye damage. The surgery was performed on a heating pad to keep a stable body temperature. Mice were given 3 weeks for recovery and viral expression. Fibre photometry measurements were carried out using the ChiSquare X2-200 system (ChiSquare Biomaging; Extended Data Fig. 7a,b). In brief, blue light from a 473 nm ps-pulsed laser (at 50 MHz; pulse width, 80 ps FWHM) was delivered through a single mode fibre. Fluorescence emission from the tissue was collected by a multimode fibre with a sample frequency of 100 Hz. The single mode and multimode fibres were arranged side by side in a ferrule that is connected to a detachable multimode fibre implant. The emitted photons collected through the multimode fibre pass through a band-pass filter (FF01-550/88, Semrock) to a single-photon detector. Photons were recorded by the time-correlated single-photon counting (TCSPC) module (SPC-130EM, Becker and Hickl) in the ChiSquare X2-200 system. Before fibre photometry recordings, mice were habituated to the head-fixation system for 5 days for around 20 min. On the first experimental day, the mice were head-fixed, connected to the recording apparatus and left waiting 5–10 min to allow the photon recording traces to stabilize. A stable baseline of 5 min was recorded and, subsequently, a Hamilton syringe (500 nl) was carefully plugged in the head-implant cannula. Soon after, 150 nl of vehicle (1× PBS), in which synaptic blockers were dissolved (5 mM NBQX, 5 mM MK801, 0.1 mM TTX, 0.015 mM Ω-agatoxin, 0.05 mM Ω-conotoxin, 0.010 mM SNX-482) was administered at a constant flow rate of 1.6 nl s⁻¹. The recording was stopped 10 min after the end of the infusion. Mice were given 3 days for recovery from the first injection and all of the procedures were repeated administering CNO (2.5 mM) in the same synaptic blockers mixture. Raw fibre photometry data were processed using the Spike2 v.8 software (Cambridge Electronic Design). Data were smoothed by a factor of 0.1 and downsampled to reach a final frequency of 2 Hz. Data were finally normalized using the $\Delta F/F_0$ formula where $F_0$ corresponds to the average photometry value during the 5 min recording before the cannula was plugged in.

### Synaptic electrophysiology experiments and analysis

**LTP of excitatory synapses in the hippocampal DG.** Male *GFAP*[CreERT2] *Slc17a7*[fl/fl]*tdTomato*[lsl/lsl] mice (P21–25) were injected with TAM (a single i.p. injection per day for 2–3 days, short protocol) to induce cell-specific *Slc17a7* gene deletion coupled to tdTomato red fluorescent protein expression in subpopulations of GFAP-expressing cells. Mice were used for electrophysiology experiments 14–19 days after the start of the TAM treatment (that is, when they were 35–40 days old). We previously showed that, after this interval, the short TAM protocol induces *cre* recombination in about one-third of the GFAP-expressing cell population in the DGML (>99% astrocytes) and that these red fluorescent astrocytes have patchy distribution[6,37]. At this stage, TAM-injected mice, were anaesthetized with isoflurane and decapitated. The brain was rapidly removed from the skull and immersed in ice-cold oxygenated sucrose-ACSF with the following composition: 62.5 mM NaCl, 2.5 mM KCl, 7 mM MgCl₂, 0.5 mM CaCl₂, 25 mM NaHCO₃, 1.5 mM NaH₂PO₄, 10 mM glucose and 105 mM sucrose, saturated with 95% O₂–5% CO₂ (pH 7.4). Horizontal hippocampal slices (350 µm) were cut with a vibratome (HM 650 V Microm) and kept in oxygenated standard ACSF: 125 mM NaCl, 25 mM NaHCO₃, 1.25 mM NaH₂PO₄, 3.5 mM KCl, 2 mM CaCl₂, 1 mM MgCl₂ and 10 mM glucose (osmolality, 295 ± 5 mOsm; pH 7.3–7.4) at 34 °C for at least 45 min. A single slice was then transferred into a recording chamber perfused at 3 ml min⁻¹ and 34 °C with ACSF containing 100 µM of the GABA_A antagonist, picrotoxin. The chamber was placed onto the stage of an upright fixed-stage microscope (Olympus BX51WI), equipped for infrared differential interference contrast and epifluorescence video microscopy (TILL Photonics). Extracellular recordings of field excitatory postsynaptic potentials (fEPSPs) were made from the medial DGML during MPP electrical stimulation using an Axopatch 200 B amplifier (Molecular Devices) with the Clampex software (Molecular Devices) and the A/D converter Digidata 1440A (Molecular Devices) connected to a computer. Stimulation was delivered by using a glass pipette filled with ACSF and connected with a current-constant stimulator (A.M.P.I., Isoflex). From each hippocampal DG slice, paired fEPSP recordings[93] were made in response to MPP stimulation by placing two recording electrodes (pipettes of 3–5 MΩ impedance filled with ACSF) along the bundle of MPP fibres, spatially aligned to the stimulating electrode (the closest at more than 200 µm from it) and spaced among them by around 200 µm (Extended Data Fig. 9d,e). One recording electrode was placed in the domain of a red fluorescent astrocyte (Astro-tdTom⁺, putatively VGLUT1[GFAP-KO]) and the other one in the domain of a non-fluorescent astrocyte (Astro), alternating in different experiments which one of the two was positioned closest to the recording electrode. In control experiments, an identical arrangement of the paired fEPSP recordings was used in slices from WT mice, but using three electrodes with around a 100 µm interdistance between them. The stimulation intensity was set to elicit around 40% of the maximal response based on input/output. A Θ-burst stimulation protocol—consisting of 10 trains of 5 pulses at 200 Hz with an intertrain interval of 100 ms, and repeated 5 times with an interval of 20 s—was used to induce long-term potentiation (Θ-LTP) of fEPSPs. The magnitude of Θ-LTP was evaluated by measuring fEPSP slopes in the period 20–30 min after Θ-burst delivery, and data were normalized to the baseline (that is, fEPSP slopes recorded in the 10 min preceding LTP stimulation; Fig. 3d–f). For each paired fEPSP recording, the magnitude of Θ-LTP (expressed as percentage increase above the baseline) in the field containing Astro-tdTom⁺ was compared to the magnitude in the field containing Astro ~200 µm away[93].

**Excitatory synaptic transmission in midbrain DA neurons.** Male *GFAP*[CreERT2]*Slc17a6*[fl/fl]*tdTomato*[lsl/lsl] mice (P21–25) were treated with TAM (2 i.p. injections per day for 5 days, long protocol) to induce cell-specific *Slc17a6* gene deletion coupled to tdTomato fluorescence expression in a large population of GFAP-expressing cells (VGLUT2[GFAP-KO]). As a control, littermate male *GFAP*[creERT2]*Slc17a6*[fl/fl]*tdTomato*[lsl/lsl] mice of the same age were treated with vehicle (corn oil, same protocol as TAM, VGLUT2[GFAP-WT]). As a further control, in this case of the TAM treatment in the absence of *cre* recombination, littermate male *Slc17a6*[fl/fl]

$tdTomato^{lsl/lsl}$ mice of the same age were treated with TAM (same as above, VGLUT2$^{WT\text{-}TAM}$). Electrophysiological experiments were conducted in horizontal midbrain slices containing the SNpc, prepared according to published procedures[94]. In brief, TAM or vehicle-treated mice (P35–40, 14 days after the first TAM or vehicle injection) were anaesthetized with isoflurane and decapitated. The brain was rapidly removed from the skull and a tissue block containing the midbrain was isolated and immersed in cold ACSF at 8 °C. The ACSF contained 126 mM NaCl, 2.5 mM KCl, 1.2 mM MgCl$_2$, 2.4 mM CaCl$_2$, 1.2 mM NaH$_2$PO$_4$, 24 mM NaHCO$_3$, 10 mM glucose, saturated with 95% O$_2$–5% CO$_2$ (pH 7.4). Horizontal midbrain slices (250 μm) were cut with a vibratome (Leica VT1000S, Leica Microsystems). Slices were maintained in ACSF at 33.0 ± 0.5 °C for 30 min before electrophysiological recordings. Whole-cell patch-clamp recordings of SNpc DA neurons were performed at 33.0 ± 0.5 °C in a recording chamber placed on the stage of an upright microscope (Nikon Eclipse FN1) equipped for infrared and epifluorescence video microscopy (CoolSnap EZ Photometrics). Slices were continuously perfused at 2.5–3.0 ml min$^{-1}$ with ACSF. SNpc DA neurons were visually selected by their localization, morphology and proximity to astrocytes, and further identified on the basis of the presence of regular spontaneous firing at 1.5–3 Hz (in cell-attached mode). Patch-clamp recordings were performed with glass borosilicate pipettes (6–8 MΩ) (WPI, TW150F-4) pulled with a PP-83 Narishige puller and filled with a solution containing 115 mM Cs-methanesulfonate, 10 mM CsCl, 0.45 mM CaCl$_2$, 10 mM HEPES, 1 mM EGTA, 4 mM MgATP, 0.3 mM NaGTP (pH 7.3 with CsOH). Recordings were made with a Multiclamp 700B amplifier (Molecular Devices) using Clampex software (Molecular Devices) and the A/D converter Digidata 1440A (Molecular Devices) connected to a computer. sEPSCs in SNpc DA neurons ($V_h = -60$ mV) were recorded in ACSF supplemented with the GABA$_A$ antagonist picrotoxin (100 μM) and the GABA$_B$ antagonist CGP55845 (1 μM). Recordings showing changes >100 pA in the holding current (at −60 mV) were discarded. Current signals were low-pass filtered at 3 kHz and digitized at 10 kHz. sEPSC amplitude and frequency were analysed from 3 min traces using Clampfit software v.10.3 (Molecular Devices). Event amplitudes and frequencies were first averaged within each experiment and regrouped by condition and the resulting means were averaged between experiments. In a set of experiments, a glass pipette electrode was placed in the STN and paired pulses at a 50 ms interval (20 Hz) were delivered every 30 s through a constant current isolated stimulator (Digitimer) to evoke excitatory postsynaptic currents (eEPSCs) in SNpc DA neurons in the presence of picrotoxin and CGP55845. The amplitude and duration of stimulation pulses were set to obtain eEPSCs of about 60–200 pA. A 2 mV hyperpolarizing step was continuously applied before each eEPSC to monitor changes in access resistance ($R_a$). Recordings were discarded if $R_a$ changed >20% during experiments or holding currents (at −70 mV) changed >100 pA during recordings. The PPR was obtained as peak 2 amplitude/peak 1 amplitude, by averaging 10 sweeps (5 min of recording) for each experiment (Fig. 4b–f and Extended Data Fig. 10e). The resulting means were averaged between experiments. To analyse the contribution of group III mGluRs in the astrocyte VGLUT2-dependent regulation of glutamatergic synaptic inputs to SNpc DA neurons, we evaluated the effects of the group III mGluR agonist L-SOP (10 μM) or the group III mGluR antagonist MSOP (10 μM) on sEPSCs and STN-stimulation-evoked EPSCs in nigral DA neurons of VGLUT2$^{GFAP\text{-}KO}$ and VGLUT2$^{GFAP\text{-}WT}$ control mice (Fig. 4d,f). In a set of experiments to evaluate the role of astrocyte VGLUT1-dependent signalling on spontaneous excitatory synaptic inputs to SNpc DA neurons, we analysed sEPSCs in nigral DA neurons of VGLUT1$^{GFAP\text{-}KO}$ and VGLUT1$^{GFAP\text{-}WT}$ mice (Extended Data Fig. 10f,g). To this aim, male $GFAP^{CreERT2}Slc17a7^{fl/fl}tdTomato^{lsl/lsl}$ mice (P21–25) were treated with TAM (2 i.p. injections per day for 5 days, long protocol) to induce cell-specific $Slc17a7$ gene deletion coupled to tdTomato fluorescence expression in GFAP-expressing cells (VGLUT1$^{GFAP\text{-}KO}$). As a control, littermate male $GFAP^{creERT2}Slc17a7^{fl/fl}tdTomato^{lsl/lsl}$ mice of the same age were treated with vehicle (corn oil, same protocol as TAM, VGLUT1$^{GFAP\text{-}WT}$). The procedures for midbrain slice preparation and patch-clamp recordings of sEPSCs in SNpc DA neurons were performed as described above.

## Behavioural experiments

Behavioural experiments consisted of an open-field test followed (1–3 days) by a contextual fear-conditioning test. They were conducted in 3–5-month-old male $GFAP^{creERT2}Slc17a7^{fl/fl}tdTomato^{lsl/lsl}$ mice injected with TAM (1 i.p. per day for 8 days, long protocol, VGLUT1$^{GFAP\text{-}KO}$) or vehicle (corn oil) as a control (VGLUT1$^{GFAP\text{-}WT}$) and in TAM-injected $Slc17a7^{fl/fl}$ $tdTomato^{lsl/lsl}$ mice, as a further control of the TAM treatment in the absence of $cre$ recombination (VGLUT1$^{WT\text{-}TAM}$). Experiments started within 19 days from the first TAM or vehicle injection as in our previous work[6] to maximize the number of recombined astrocytes while avoiding potential effects of recombined SGZ neural precursors on contextual memory[95]. The open-field test was conducted in a home-made squared open-field arena (45 cm × 45 cm, with a height of 40 cm) made of grey plastic. Illumination was set at 70–80 lux. Mice were gently placed in the middle of the open field and allowed to freely explore the arena for 30 min. They were continuously recorded using an infrared video camera placed above the arena and the images were analysed using Ethovision XT 11 software (Noldus). The mean speed, total distance travelled and time spent immobile were measured to assess locomotor activity and exploration. The time spent in a virtual inner zone (15 × 15 cm area in the middle of the arena) was measured as an inverse index of anxiety (Extended Data Fig. 9m). To study contextual learning and memory, mice were then tested with contextual fear-conditioning for acquisition of context-conditioned fear (learning) and for its expression 24 and 48 h later (memory) using a fear conditioning arena (context) with a grid floor that could be electrified, placed within an isolation chamber (Ugo Basile) and controlled by Ethovision XT software as above. Mice were first given a 5 min activity test under video recording without electroshocks to assess locomotor activity, baseline freezing and rearing. Next, the conditioning session comprised six inescapable electroshocks (0.3–0.6 mA × 2 s each), delivered at intervals of 2 min. After the session, the mice were returned to their home cages. The next day (for the 24 h test) or after 2 days (for the 48 h test), the mice were placed back into the same arena in the absence of electroshocks for the mnemonic fear expression test lasting 21 min. In each of the above sessions, the main measure was the percentage of time spent by the mice freezing during each time interval, with freezing defined as an episode during which no movement of the mouse was detected for at least 2 s. For the conditioning session, the mean percentage of time freezing was calculated for each 2 min interval between electroshocks and the cumulative percentage of freezing of 2 consecutive inter-shock-intervals (ISI) was summed and displayed (ISI 1–2 min, ISI 3–4 min, ISI 5–6 min); for the fear expression at 24 h and 48 h, sessions lasted 11 min and the mean percentage of time freezing was calculated during 10 min, excluding the first minute of recording. Furthermore, the distance moved by the mice during the electroshock delivery was measured as an indicator of pain sensitivity, assuming that such distance is proportional to the sensed pain. All of the behavioural apparati were carefully washed with 70% ethanol solution between tests (Fig. 3g,h and Extended Data Fig. 9m).

## Video EEG recording and analysis of acute seizures

Surgeries for the EEG headmount implant were performed in 2–5-month-old male $GFAP^{creERT2}Slc17a7^{fl/fl}tdTomato^{lsl/lsl}$ mice 7 days after the last TAM injection (once i.p. per day for 8 days, VGLUT1$^{GFAP\text{-}KO}$) or vehicle (corn oil) injection as control (VGLUT1$^{GFAP\text{-}WT}$), as well as in $Slc17a7^{fl/fl}tdTomato^{lsl/lsl}$ mice injected with TAM (same as above), as a further control of the TAM treatment in the absence of $cre$ recombination (VGLUT1$^{WT\text{-}TAM}$; Fig. 3j–p). The initial surgery preparation was performed as described in the "Stereotaxic viral injections"

section. After scalp exposure, six holes to host the EEG electrodes were drilled into the skull. Two electrodes were placed between the dura and the skull above the frontal cortex (right and left), two at intrahippocampal locations in the CA1 (right and left; ML: ±1.8 mm; AP: −1.8 mm; DV: −1.9 mm), one as a ground and another as a reference between the dura and the skull above the cerebellum. All electrodes were soldered to an EEG headmount (Pinnacle Technology) and fixed onto the skull by acrylic cement (Paladur-Dentonet). Transmitter-implanted mice were single-housed in individual cages and allowed to recover for 7 days before pharmacological induction of acute seizures by subcutaneous injection of a single dose (10 mg per kg) of kainic acid[96] (KA, Abcam 120100, in sterile NaCl 0.9% at 0.5 mg ml$^{-1}$). EEGs were continuously recorded at a sampling rate of 250 Hz in each experimental cage using an EEG-telemetry system with associated video monitoring of the movement of the animals (Pinnacle Technology) from 24 h before administration of KA to 48 h after its injection. For detecting and analysing KA-induced seizures, EEG traces and video recordings were evaluated by the experimenter through manual scoring with Sirenia seizure software v.1.7 (Pinnacle Technology). Artifacts in the raw EEG traces (electrical noise, exploratory behaviour and grooming) were manually identified and excluded from analyses. Quantitative analysis of seizures was performed in the first 4 h after KA injection, when most seizures occurred. Seizures were defined as EEG segments starting with low-amplitude high-frequency activity (tonic phase) and evolving into higher-amplitude and lower-frequency bursts (clonic phase) with a minimal duration of 30 s. Parameters analysed and expressed as the mean value for each individual mouse were: (1) seizure latency (mean time to first seizure): the time between KA injection and the start of the first seizure; (2) the total number of seizures during the 4 h post-KA period. Furthermore, for each individual seizure, we measured (3) the seizure length, defined as the time between the onset and the end of the seizure episode, with the end defined as the time when the EEG returned to the mean baseline or (in the case of postictal depression) to a value lower than the mean baseline; (4) the time first-to-last seizure, which is the time interval between the appearance of the first seizure and of the last seizure; (5) the inter-ictal activity duration, which is the time between seizures.

## In vivo striatal DA levels measured by microdialysis

Male *GFAP$^{creERT2}$Slc17a6$^{fl/fl}$tdTomato$^{lsl/lsl}$* mice (aged 21–25 days) were treated with TAM (VGLUT2$^{GFAP-KO}$) or vehicle (corn oil, VGLUT2$^{GFAP-WT}$) (2 i.p. injections per day for 5 days). Four weeks after the TAM or vehicle treatment, mice (aged 50–55 days) underwent surgeries for the implant of microdialysis probes in the nucleus striatum, according to authorized procedures (375/2018 PR, Animal Care Committee of Italian Ministry of Health). Mice, anaesthetized with Zoletil 100 (tiletamine HCl 50 mg ml$^{-1}$ + zolazepam HCl 50 mg ml$^{-1}$) and rompun (xylazine 20 mg ml$^{-1}$), were mounted on a stereotaxic frame (David Kopf Instruments) and implanted unilaterally with a microdialysis probe (length, 4.5 mm; dialysing portion, 2 mm; AN69 fibres, Hospal Dasco) at the level of the dST (AP +1.0 mm, ML +1.8 mm). Probes were fixed onto the mouse skull with epoxy glue and dental cement, and the skin was sutured. At the end of surgery, animals were housed individually into a new home cage to avoid breaking the implantation and were allowed to recover for 24–36 h. On the day of the experiment, the mice were introduced into individual testing cages and the microdialysis probe was connected to a CMA/100 pump (Carnegie Medicine) through PE-20 tubing and an ultralow torque multi-channel power assisted swivel (Model MCS5, Instech Laboratories) to allow free movement. AaCSF (140.0 mM NaCl, 4.0 mM KCl, 1.2 mM CaCl$_2$, 1.0 mM MgCl$_2$) was pumped through the dialysis probe at a constant flow rate of 2.1 µl min$^{-1}$. After the start of the dialysis perfusion, the mice were left undisturbed for at least 1 h, then the dialysates were collected every 20 min. The mean DA concentration of the three samples collected before treatment was

taken as the baseline concentration. After collection of three baseline samples, the mice were injected intraperitoneally with amphetamine (2.0 mg per kg, Sigma-Aldrich) to amplify the extracellular DA signal[97] or its vehicle (0.9 % NaCl) and dialysates were collected for an additional 120 min after drug treatment. Each dialysate sample (20 µl) was analysed through an ultraperformance liquid chromatography apparatus (ACQUITY, Waters) coupled to an amperometric detector (Decade II, Antec Leyden) containing an in situ Ag/AgCl reference electrode and an electrochemical flow-cell (VT-03, Antec Leyden) with a 0.7 mm glassy carbon electrode, mounted with a 25 mm spacer. The electrochemical flow-cell, set at a potential of 400 mV, was positioned immediately after a BEH C18 column (2.1 × 50 mm, 1.7 mm particle size; Waters) kept at 37 °C (0.07 ml min$^{-1}$ flow rate). The composition of the mobile phase was as follows: 50 mM phosphoric acid, 8 mM KCl, 0.1 mM EDTA, 2.5 mM 1-octanesulfonic acid sodium salt, 12% methanol and pH 6.0 adjusted with NaOH. The peak height obtained by oxidation of DA was compared with that produced by a standard. The detection limit was 0.1 pg. DA concentration was expressed as pg per 20 µl dialysate sample (Fig. 4h,i). The correctness of probe placement in the dST was confirmed after each experiment by brain dissection and slicing, and through observation under a microscope.

## Statistics and reproducibility

Data are presented as mean ± s.e.m. and normalized data are calculated on the baseline, unless otherwise indicated. Data were considered to be significantly different when $P < 0.05$. $P$ values less than 0.05, 0.01 and 0.001 are indicated by 1, 2 and 3 asterisks, respectively. The normality of datasets was evaluated using Shapiro–Wilk or Kolmogorov–Smirnov tests. For normally distributed data, two-tailed unpaired $t$-tests were used when comparing the means of two independent experimental populations. For cases in which the populations were not independent, paired $t$-tests were used. ANOVA was used when comparing the means of more than two populations. Repeated measures ANOVA with or without independent treatment groups was performed on time-course experiments. For cases in which ANOVA tests yielded significant effects, appropriate post hoc comparisons were used to identify significant pairwise differences (Fisher's LSD test or Tukey's test). For analysis of the predicted cell-type contingency table, we used two-tailed Fisher's exact tests. When normality was violated, the data were analysed using nonparametric Kolmogorov–Smirnov tests, Kruskal–Wallis tests with Dunn's multiple-comparison test, two-sided Wilcoxon rank-sum tests or Friedman ANOVA followed by post hoc Wilcoxon signed-rank test or Kolmogorov–Smirnov test. For statistical analyses, GraphPad Prism 9 (GraphPad), OriginPro 2018b (OriginLab), MATLAB 2019b and Excel were used. For GO statistical analysis, GSEA software was used (http://software.broadinstitute.org/gsea/index.jsp). When possible, experimenters were blinded to the genotypes of the animals and genotypes were decoded after data had been processed and analysed. Full statistical details for each main figure panel are included in Supplementary Table 4. Data shown from representative experiments were repeated with similar results in at least two independent biological replicates, unless otherwise noted. Sample sizes were estimated empirically on the basis of previous studies.

## Reporting summary

Further information on research design is available in the Nature Portfolio Reporting Summary linked to this article.

## Data availability

The scRNA-seq datasets generated and/or analysed during this study are available at Zenodo (https://doi.org/10.5281/zenodo.7704838). The available datasets analysed during this study and their public domain

resources are indicated in Extended Data Fig. 1a and in the Methods. All other original datasets presented in this study are provided as Source Data. Source data are provided with this paper. Correspondence for A.V. may be addressed to andrea.volterra@unil.ch or andrea.volterra@wysscenter.ch.

## Code availability

Algorithms and codes used to analyse scRNA-seq datasets are described in Methods and Extended Data Fig. 1e. Codes used to analyse glutamate two-photon imaging experiments are described in Methods and public domain resources are listed in Supplementary Table 3. Folders containing scripts and data sufficient to visualize the code use and to reproduce analysis of a demo dataset are available for download at Zenodo (https://doi.org/10.5281/zenodo.7704838). The folder 'transcriptomic_analysis.tar.xz' is an archive file containing an R project including all of the scripts and data to reproduce the scRNA-seq analysis. The folder 'iGluSnFRanalysis.7z' is a zip file containing two distinct subfolders called in-vivo and ex-vivo data. They contain analysis tools to reproduce the iGluSnFR datasets presented in Fig. 2. Each folder contains a specific ReadMe file to explain step by step how to perform the analysis.

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

**Acknowledgements** We thank H. Stubbe for technical support with animal breeding and genotyping through a large part of the project; D. Sahlender for initial characterization of *GFAP*[creERT2]*Slc17a6*[fl/fl]*tdTomato*[lsl/lsl] mice; C. Vivar Rios for developing the pipeline for glutamate imaging analysis in situ in the first part of the study; C. Ruiz and F. Calvo from the CSR-UNIL team for developing an open source version of the same imaging pipeline; J. Prados for sharing the torch prediction algorithm and for his general advice; E. Magrinelli for performing initial bioinformatic analysis of patch-seq experiments and integration of the existing databases; B. Xiong for contributing to image acquisition for the RNAscope HiPlex experiment; R. Edwards and F. Kirchhoff for sharing the *Slc17a6*[fl/fl] and *GLAST*[creERT2]*P2ry1*[fl/fl] mouse lines, respectively; M. Holt, M. Batiuk and A. Martirosyan for sharing unpublished data and methods, advice and discussions; C. Pryce for advice on protocols and analysis of behavioural experiments; N. Liaudet for advice on imaging experiments and analysis, and for reading parts of the manuscript; and D. Jabaudon for advice on single-cell transcriptomics and for reading the manuscript. Research in the Volterra laboratory was supported by Swiss National Science Foundation (SNSF) NCCR TransCure grant/award number 51NF40-160620; SNSF grants/award numbers 31003A-173124 and 31003B-201276; European Research Council (ERC) Advanced Grant 340368 "Astromnesis"; Stiftung Synapsis—Demenz Forschung Schweiz DFS, grant/award number 2018-PI-01; and the Wyss Center for Bio and Neuroengineering, Geneva. Research in the Telley laboratory was supported by ERC starting grant CERDEV_759112 and a SNSF grant 31003A_182676/1.

**Author contributions** R.D.C. supervised most of the experimental and analytical part of the project, designed, performed and analysed viral, molecular, morphological, EEG and behavioural experiments, designed and supervised breeding schemes of transgenic lines, participated in single-cell transcriptomics, RNAscope HiPlex and imaging experimental design, experiments and analysis, including fibre photometry, wrote parts of the manuscript, designed most of the figures, and supervised some of the co-authors during experiments, analyses and writing of the paper. A.L. performed patch-seq experiments, performed and analysed electrophysiological experiments in hippocampus, designed experiments on the nigrostriatal DA circuit, performed and analysed electrophysiological recordings in nigral DA neurons, contributed to design, execution and analysis of in vivo microdialysis experiments, and contributed to writing the corresponding sections of the manuscript. D.G.L. designed, performed and analysed SF-iGluSnFR imaging experiments in situ, participated in the design and preparation of the ad hoc pipeline for SF-iGluSnFR image analysis in situ, performed combined imaging and patch-seq experiments with A.L. and wrote the corresponding sections of the manuscript. B.L.L. designed, performed and analysed SF-iGluSnFR imaging experiments in vivo, designed and wrote the code for SF-iGluSnFR image analysis in vivo and wrote the corresponding sections of the manuscript. G.C. developed the acute seizure and video EEG approach and performed the initial studies and performed viral injections. E.C.L. designed, performed and analysed in vivo microdialysis experiments. E.B. performed two-photon $Ca^{2+}$ imaging experiments and patch-seq videos with A.L. M.A.D.C. performed electrophysiology experiments combined with two-photon $Ca^{2+}$ imaging. I.S. designed dual-recording LFP Θ-LTP hippocampal experiments and performed the initial studies. I.V. performed the RNAscope HiPlex assay. A.R. developed analysis for RNAscope HiPlex experiments. M.C. performed fibre photometry experiments. T.C. performed

immunohistochemistry, image analysis and cell counting experiments, and collaborated in the behavioural experiments. W.W. and K.H. developed the *Slc17a7*$^{fl/fl}$ transgenic line and gave advice on its use in this project. M.M. designed the fibre photometry experiments, contributed to their analysis and to writing that part of the manuscript. N.M. contributed to designing the substantia nigra electrophysiological experiments and supervised that part of the project. L.T. designed and supervised the single-cell and spatial transcriptomics part of the project, developed and used the bioinformatics approaches for generating all of the integrated databases and for their analysis as well as for analysis of patch-seq and RNAscope HiPlex experiment, wrote that part of the manuscript, and participated in the overall writing and strategy of the project. A.V. supervised the entire project, defined its strategy and, together with the other authors, designed its different components and wrote the manuscript.

**Competing interests** The authors declare no competing interests.

**Additional information**
**Correspondence and requests for materials** should be addressed to Ludovic Telley or Andrea Volterra.

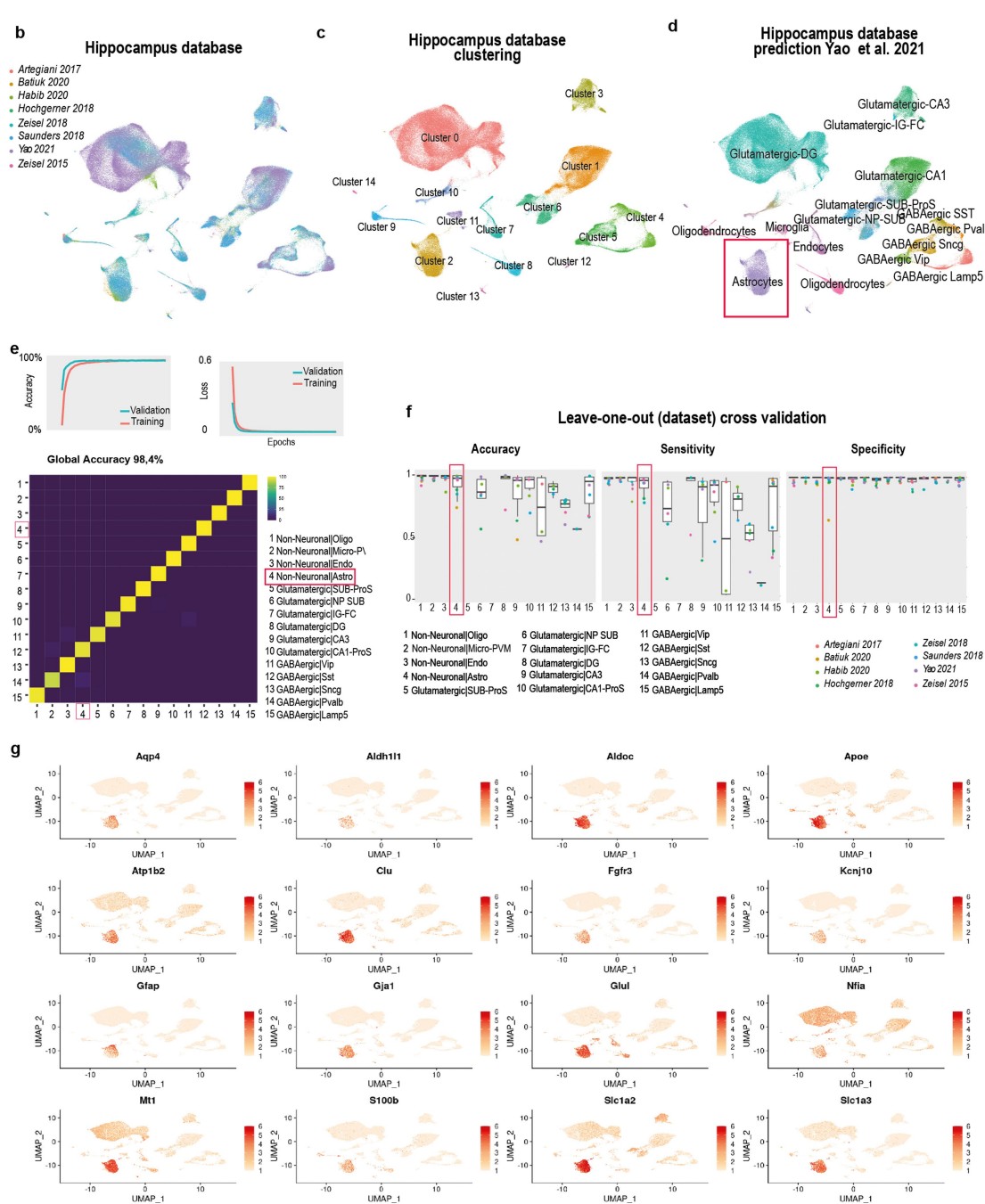

**a** Single cell Hippocampus database

| dataset | mouse age | Region | sc/snRNA-seq technology | dataset code | Reference |
|---|---|---|---|---|---|
| Artegiani 2017 | P56 | DG | scRNAseq/Sort-seq | GSE106447 | 10.1016/j.celrep.2017.11.050 |
| Batiuk 2020 | P56 | HIP | scRNAseq/Sort-seq | GSE114000 | 10.1038/s41467-019-14198-8 |
| Habib 2020 | P210 | HIP | sNuc-seq/10X Genomics | GSE143758 | 10.1038/s41593-020-0624-8 |
| Hochgerner 2018 | P18,P19,P23, P120,P132 | DG | scRNAseq /10X Genomics | GSE95753 | 10.1038/s41593017-0056-2 |
| Saunders 2018 | P60 | HIP | scRNAseq/Drop-seq | http://dropviz.org | 10.1016/j.cell.2018.07.028 |
| Yao 2021 | P56 | HIP | scRNAseq/Sort-seq & 10X | https://assets.nemoarchive.org/dat-jb2f34y | 10.1016/j.cell.2021.04.021 |
| Zeisel 2015 | P25 | CA1 | scRNAseq/C1-Fluidigm | GSE60361 | 10.1126/science.aaa1934 |
| Zeisel 2018 | P23,P27,P28, P60 | DG | scRNAseq/C1-Fluidigm | SRP135960 | 10.1016/j.cell.2018.06.021 |

**Extended Data Fig. 1** | See next page for caption.

**Extended Data Fig. 1 | Single-cell mouse hippocampus integrated database: clusters and cell types prediction. a**, Single-cell mouse hippocampus datasets used to create the integrated database. Notice that studies generating individual datasets present relevant biological (age of mice, hippocampal region) and/or methodological (dissociation methods, chemistry, platform, enrichment strategies) differences among them. **b**, UMAP representation of 8 integrated hippocampus scRNA-seq datasets labelled by dataset. **c**, Cluster analysis revealed 15 transcriptionally distinct clusters (clustering resolution = 0.1). **d**, Integrated hippocampus database coloured by cell type prediction. **e**, Deep neural network multiclass model trained on a comprehensive database[29] at "subclass level". *Top*, Accuracy and loss value after each epoch for training and validation data; *Bottom*, confusion matrix showing cell prediction for validation data. **f**, We performed cross-validation by removing each individual dataset one at a time and running the prediction and clustering using the others, then calculated the overall prediction efficiency. This showed the accuracy, sensitivity, and specificity of our model. Data are shown as box plot (25–75 percentile with median) with min to max whiskers excluding outliers. **g**, Expression levels for canonical astrocytic markers in the integrated hippocampus database.

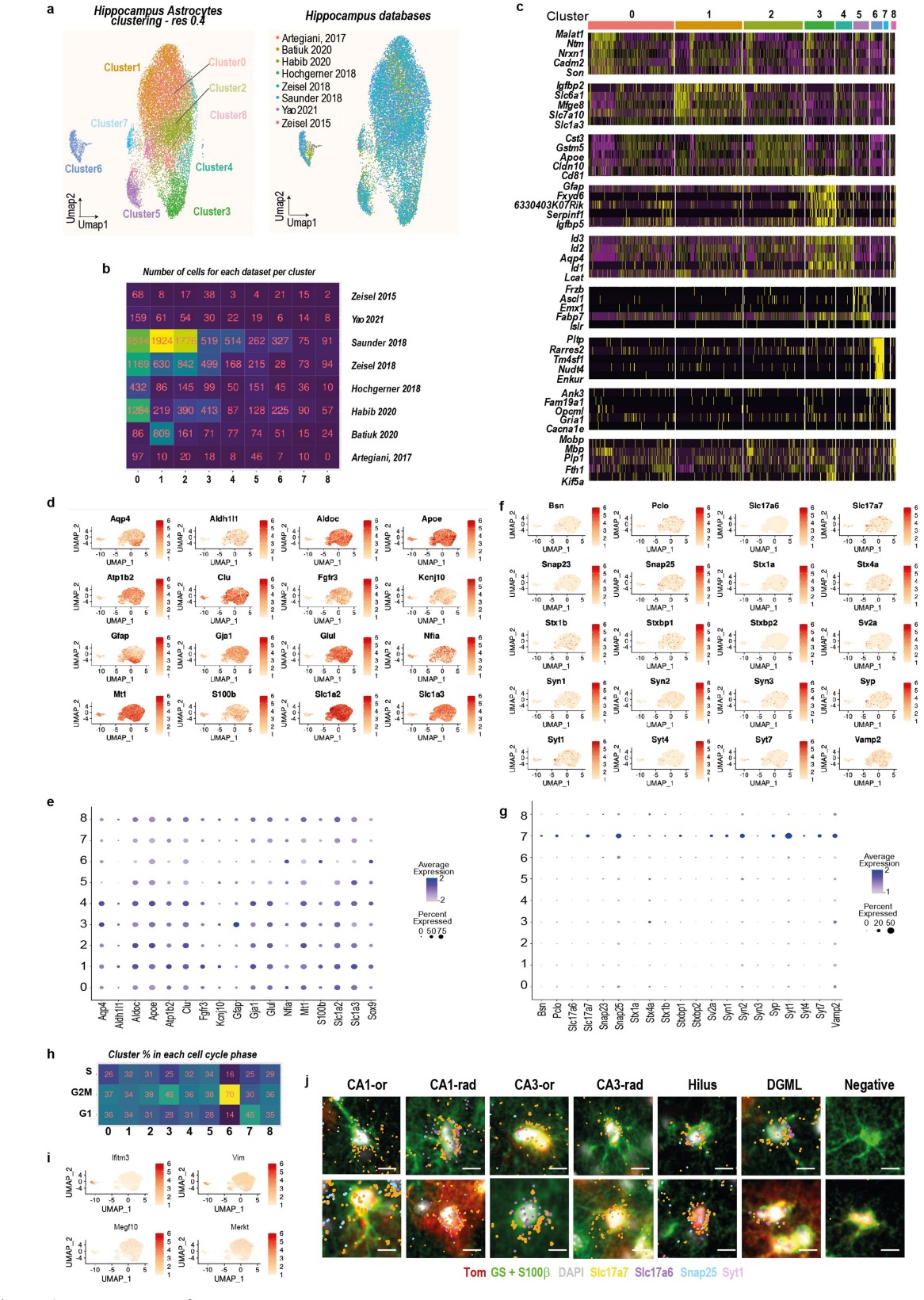

**Extended Data Fig. 2** | See next page for caption.

**Extended Data Fig. 2 | Characteristics of the distinct astrocyte transcriptional clusters and presence of cluster 7 glutamatergic astrocytes in all the hippocampal regions. a**, UMAP representation of subset astrocytes labelled by transcriptionally distinct clusters (clustering resolution = 0.4, *left*) or datasets (*right*). **b**, Heatmap showing the total number of astrocytes for each dataset and each cluster. Noteworthy, cluster 7 was found in all interrogated hippocampal databases. **c**, Expression of the top 5 enriched genes per astrocyte cluster (see also Supplementary Table 2). **d**–**g**, Expression intensity per astrocyte cluster and corresponding dot plot for selected canonical astrocytic (**d**,**e**), vesicular trafficking, regulated exocytosis and glutamatergic pre-synaptic function markers (**f**,**g**). **h**, Heatmap showing the percent of cells in each cell cycle phase for each astrocytic cluster. **i**, Expression level per astrocyte cluster for *Ifitm3*, *Vim*, *Megf10* and *Merkt*. **j**, High-magnification images of examples of glutamatergic astrocytes in various regions of the hippocampus: DGML: molecular layer of the dentate gyrus; Hilus: Hilus region of the dentate gyrus, CA3-RAD: stratum radiatum of the CA3 region, CA3-OR: stratum oriens of the CA3 region; CA1-RAD: stratum radiatum of the CA1 region, CA1-OR: stratum oriens of the CA1 region. The visualization was achieved using a combination of immunohistochemistry for tdTomato (Tom, red), GS/S100β (green), and DAPI (white) and fluorescent *in situ* hybridization for *Slc17a7* (yellow), *Slc17a6* (violet), *Snap25* (blue), and *Syt1* (pink). *n* = 12 slices, 2 mice. Scale bar: 10 μm.

## Top 10 Gene ontology Biological Processes

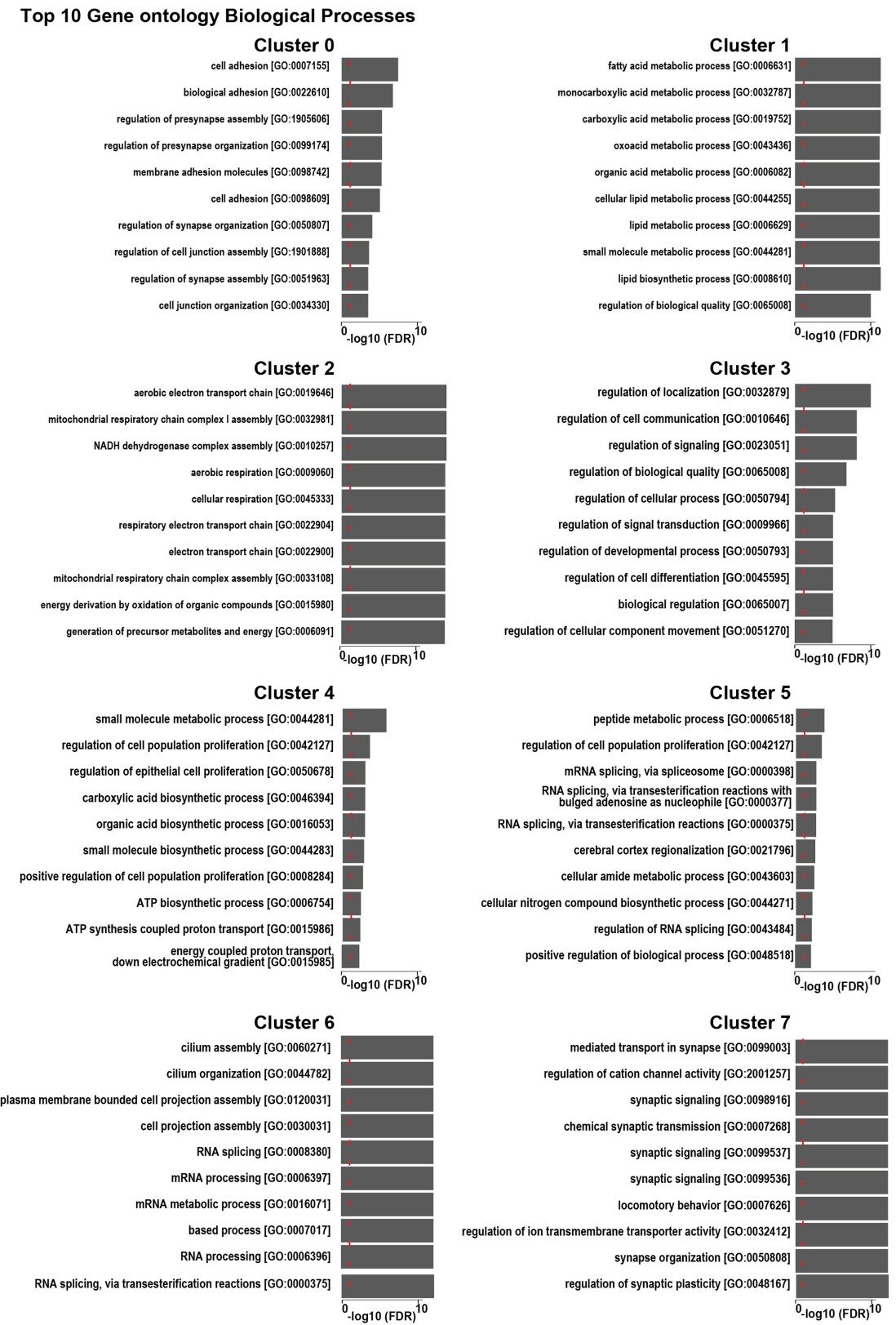

**Extended Data Fig. 3 | Top 10 biological processes ontology enrichment for each astrocyte cluster.** The red dashed line indicates the threshold for significant enrichment in these gene ontologies.

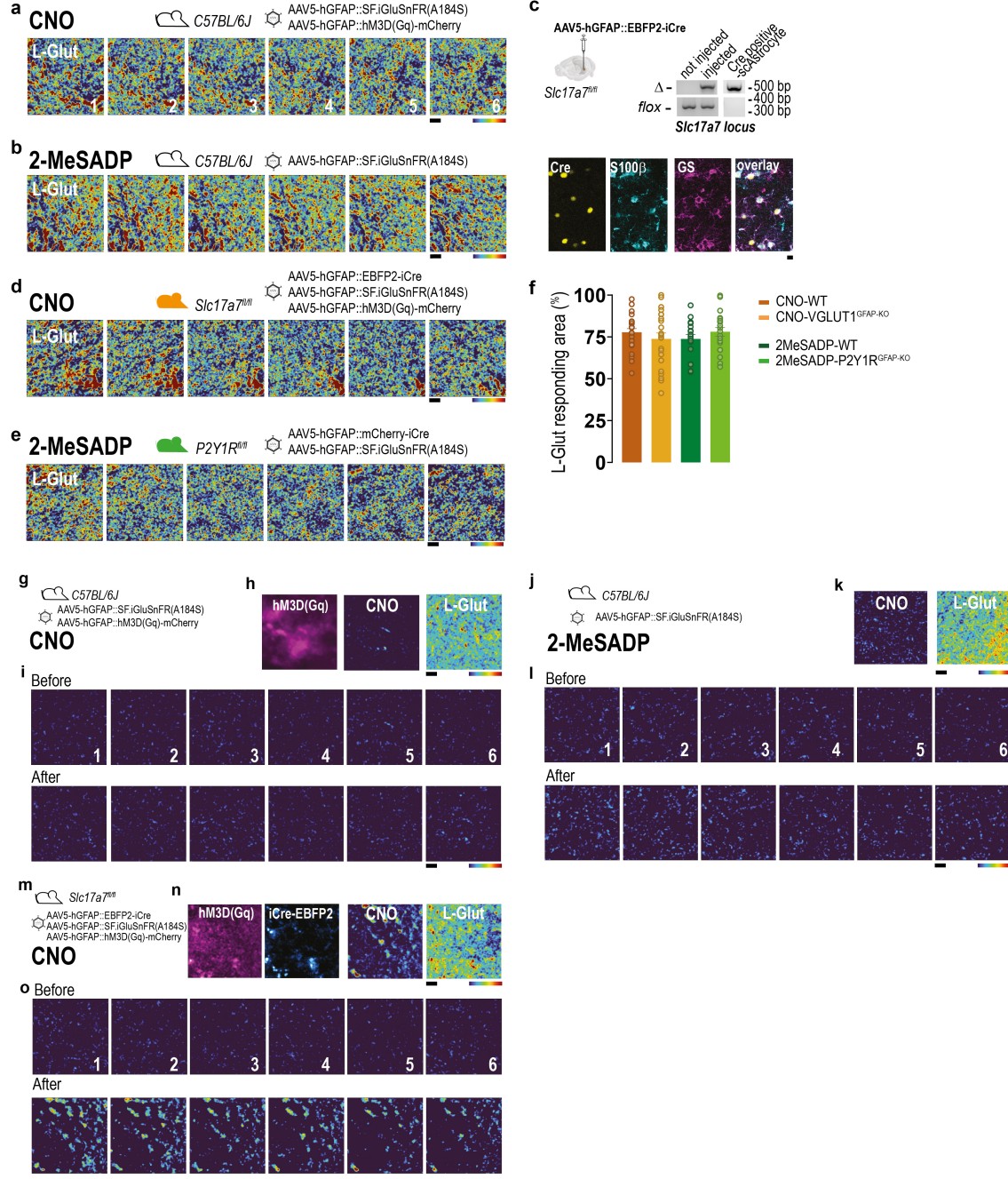

**Extended Data Fig. 4** | See next page for caption.

**Extended Data Fig. 4 | Additional data related to the glutamate imaging studies *in situ* presented in Fig. 2. a**, 6 mean projections showing the individual SF-iGluSnFR responses to 6 L-Glut puffs onto the same FOV as the wild-type CNO-responder shown in Fig. 2c–e. Each image is a pixel-by-pixel map of the SF-iGluSnFR signal after the L-Glut application. Amplitude of the responses in individual pixels is expressed in z-scores scale and colour coded, going from 0 (dark blue) to 6 (red). Note the responsiveness of most of the FOV. **b**, 6 mean projections showing the individual SF-iGluSnFR responses to 6 L-Glut puffs onto the same FOV as the wild-type 2MeSADP-responder shown in Fig. 2g–i. **c**, Validation of VGLUT1 deletion in astrocytes (VGLUT1^GFAP-KO); *Top left*, strategy to obtain *cre*-mediated VGLUT1 deletion in astrocytes through injection of *AAV5-hGFAP::EBFP2iCre* into the hippocampus of *Slc17a7 ^fl/fl* mice. The whole-brain image is from the Allen Mouse Brain Connectivity Atlas (https://mouse.brain-map.org/). *Right*, *top*, validation by genomic PCR revealing *Slc17a7* locus deletion (Δ band in all gels) in brains of virally-injected and not injected *Slc17a7^fl/fl* mice (*n* = 2 independent biological brain samples per group), or in *cre*-positive, blue-fluorescent DGML astrocytes individually collected with a patch pipette (*n* = 2 cells, 1 mouse). *Bottom*, co-labelling of *cre*-expressing cells (yellow, false colour for EBFP2-expressing, blue-fluorescent cells) with S100β (cyan) and GS (magenta) astrocytic markers. Scale bar: 5 μm. **d**, 6 mean projections showing the individual SF-iGluSnFR responses to 6 L-Glut puffs onto the same FOV as the VGLUT1^GFAP-KO not responding to CNO shown in Fig. 2k,l. **e**, 6 mean projections showing the individual SF-iGluSnFR responses to 6 L-Glut puffs onto the same FOV as the P2Y1R^GFAP-KO not responding to 2MeSADP shown in Fig. 2n,o. **f**, Proportion of the imaged FOV responding to sequential L-Glut administrations after either CNO or 2MeSADP administrations for all the tested cells in the mouse groups as in Fig. 2p. Responses to L-Glut in VGLUT1^GFAP-KO or P2Y1R^GFAP-KO mice do not differ from those in wild-type mice.

Data presented as mean ± s.e.m. WT, n = 23 cells, 5 mice; VGLUT1^GFAP-KO mice, n = 24 cells, 5 mice. 2MeSADP-stimulated astrocytes: WT, n = 18 cells, 2 mice; P2Y1R^GFAP-KO, n = 20 cells, 2 mice. **g–i**, Example of a wild-type CNO non-responder; **g**, viral injections as in Fig. 2b. **h**, *Left*, mean projection showing Gq-DREADD-mCherry expression pattern, resembling the one seen in CNO-responders (Fig. 2c). *Middle*, Standard deviation (s.d.) projection displaying the signal variance across 6 CNO stimulations, showing no CNO-evoked SF-iGluSnFR response. *Right*, S.d. projection showing reliable responses to L-Glut applications for the same FOV. **i**, 6 mean projections showing the individual SF-iGluSnFR responses to 6 CNO puffs in the same cell as in h. Each image is a pixel-by-pixel colour-coded z-score map of the SF-iGluSnFR signal in the FOV during the 240s before and after the CNO application. **j-l**, Example of a wild-type 2MeSADP non-responder; **j**, viral injections as in Fig. 2f; **k**, *Left*, S.d. projection displaying the signal variance across 6 2MeSADP stimulations, showing no 2MeSADP-evoked SF-iGluSnFR response. *Right*, S.d. projection showing reliable responses to L-Glut application for the same FOV. **l**, 6 mean projections showing the individual SF-iGluSnFR responses to 6 2MeSADP puffs in the same cell as in k. Details as in panel i. **m-o**, Example of a VGLUT1^GFAP-KO astrocyte classified as CNO responder (n = 3/24 cells); **m**, mouse line and viral treatments as in Fig. 2j; **n**, Images from *left* to *right*: (i) mean projection showing Gq-DREADD-mCherry expression pattern (magenta); (ii) mean projection showing EBFP2-iCre expression (blue) within the same FOV; (iii) S.d. projection displaying the signal variance across 6 CNO stimulations, showing some SF-iGluSnFR response to CNO; (iv) S.d. projection showing reliable responses to L-Glut application for the same FOV. **o**, 6 mean projections showing the individual SF-iGluSnFR responses to 6 CNO puffs in the same cell as in n. Details as in panel i. Responses are smaller than in CNO responder cells from wild-type mice: quantitative comparison in Fig. 2q.

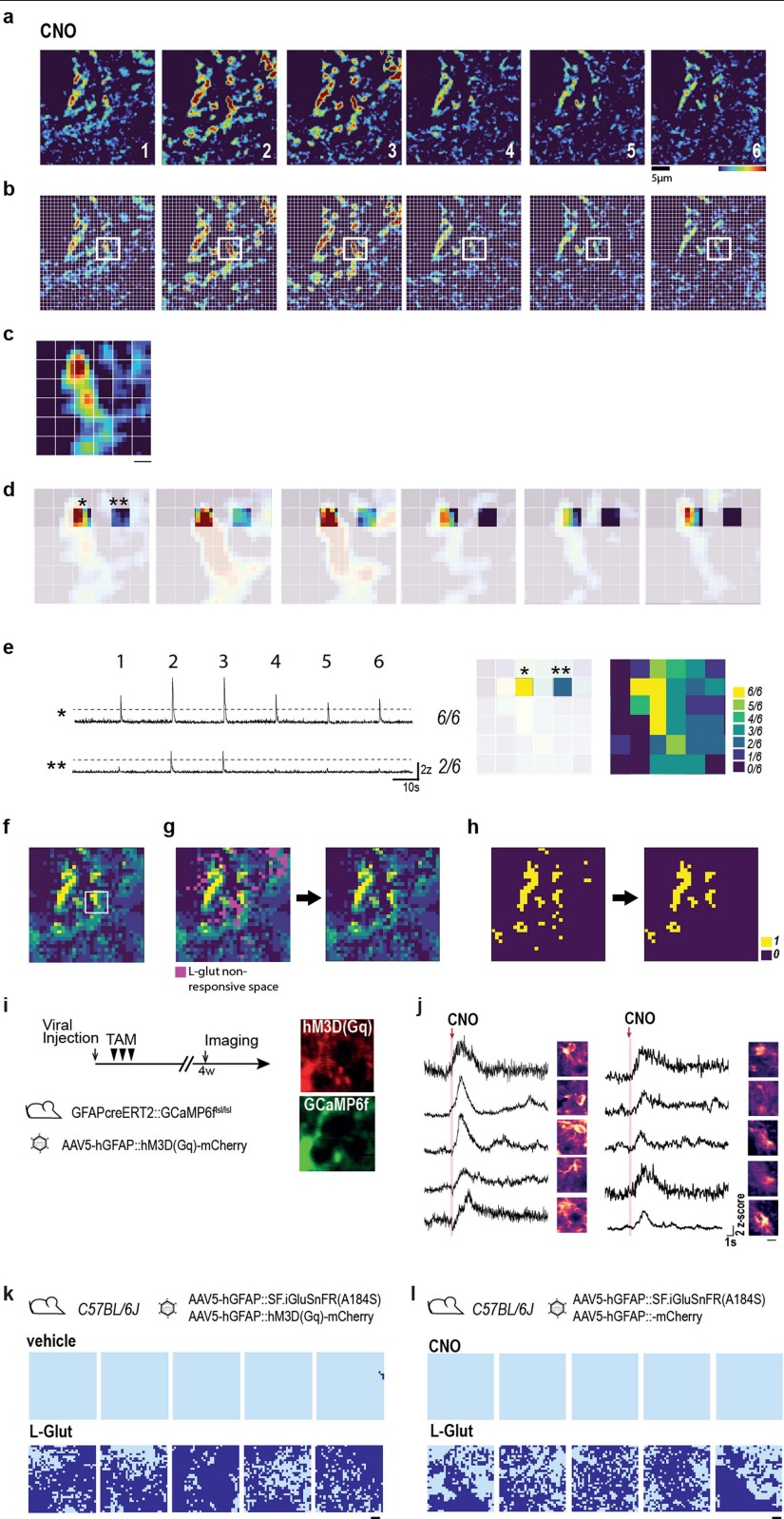

**Extended Data Fig. 5** | See next page for caption.

**Extended Data Fig. 5 | Generation of binarized functional maps of stimulus-evoked SF-iGluSnFR responses and additional experiments related to chemogenetic activation of astrocytes *in situ*. a**–**h**, Description of the analytical pipeline used to quantify SF-iGluSnFR responses. **a**, 6 epochs corresponding to short periods before or after drug applications (240 ms after CNO in this example) were used as input for the analytical pipeline. For each epoch, we generated an image representing the pixel-by-pixel colour-coded z-score mean projection map of the SF-iGluSnFR signal in the FOV for the period. **b**, For each of the epochs, we segmented the FOV by a 32 x 32 grid, in which each of the 1024 spaces represented a 1.13 µm x 1.13 µm ROI. **c**, As an example, we show at higher magnification the z-scored SF-iGluSnFR signal for epoch 1 in the region of 36 ROIs framed in **b**. **d**, To continue the example, we then focus on two nearby individual ROIs (* and **) within this framed region, and perform peak detection across the 6 rounds of CNO application. **e**, *Left*, Traces show 6/6 suprathreshold (>2 z-scores) responses to CNO in ROI (*) and only 2/6 in ROI (**). Peak detection is similarly performed in all 1024 ROIs of the 32x32 grid, counting the number of responses to CNO application (maximum of 6) within each ROI to generate a colour-coded map of the entire 37.3 x 37.3 µm FOV, going from yellow ROIs (6/6 suprathreshold responses like in ROI *) to dark blue ROIs (0/6 responses). *Right*, example of the colour-coded map in the magnified region of 36 ROIs. **f**, The low-magnification view of the colour-coded map for the entire 37.3 x 37.3 µm FOV, with magnified region in the white square, allows to visually appreciate the ROIs most consistently responding to CNO. **g**, The same analytical steps used for segmentation and peak detection of the SF-iGluSnFR responses to CNO were applied to the responses evoked by 6 applications of L-Glut in the same FOV. *Left*: while most ROIs reliably responded to L-Glut application (>4 suprathreshold peaks; not shown), a few of them did not (here depicted as magenta ROIs) and were subtracted from the CNO map to generate a new grid map (*Right*) containing only CNO responsive ROIs also reliably responsive to L-Glut application. This step helped eliminating false positive, ensuring that the CNO-evoked SF-iGluSnFR response came from a location capable of reliably detecting L-glutamate. **h**, *Left*, binarized map of the grid map from panel g *Right*. ROIs with ≥4 CNO-evoked SF-iGluSnFR responses were assigned a value of 1 (yellow) and those with ≤3 responses were assigned a value of 0 (purple). *Right*, we grouped clusters of suprathreshold recurrently active ROIs (yellow) based on 8-neighbour connectivity (all edges and corners) and excluded active clusters containing <4 ROIs by spatial filtering (see Methods). The final binarized functional map, containing only active clusters ("hotspots") with ≥4 neighbours, was used to calculate hotspots number and areas. **i-j**, CNO-dependent Gq-DREADD stimulation evokes Ca²⁺ elevations in all the tested astrocytes. **i**, *Left*, timeline of the experiments: TAM-inducible *GFAP^creERT2^GCaMP6f^fl/fl^* mice were unilaterally injected with *AAV5-hGFAP::hM3D(Gq)-mCherry* virus. After 3 days mice received TAM administration for 3 days and after 4 weeks two-photon Ca²⁺ imaging was performed. *Right*, representative fluorescence image of an astrocyte FOV (red: hM3D(Gq); green: GCaMP6f). (*n* = 2 mice). **j**, Traces of cytosolic GCaMP6f Ca²⁺ responses in the ROI (whole astrocyte) for each tested astrocyte (*n* = 10 cells) in response to a single puff of CNO (100 µM) expressed in z-scores of the raw GCaMP6f signal. Note large Ca2+ elevation in all CNO-stimulated astrocytes. Each trace is accompanied by ROI display as perceptually uniform 'magma' colormap. Scale bar: 5 µm. **k**, Stimulation with vehicle does not reproduce the glutamate-releasing effect of CNO in Gq-DREADD-expressing astrocytes: *top*, wild-type mice (n = 2) were unilaterally injected in hippocampus with *AAV5-hGFAP::SF.iGluSnFR(A184S)* and *AAV5-hGFAP::hM3D(Gq)-mCherry* viruses. *Bottom*, Binarized functional maps of vehicle- and L-Glut-evoked SF-iGluSnFR responses of 5 individual astrocyte FOVs. In none of them, vehicle induced a significant response, while in all of them L-glut elicited the usual large response. Scale bar: 5 µm. **l**, CNO does not evoke glutamate release in astrocytes expressing an mCherry scrambled virus instead of Gq-DREADD. *Top*, wild-type mice (n = 2) were unilaterally injected in hippocampus with *AAV5-hGFAP::SF.iGluSNFR(A184S)* and *AAV5-hGFAP::mCherry* viruses. *Bottom*, binarized functional maps of CNO- and L-Glut-evoked SF-iGluSnFR responses of 5 individual astrocyte FOVs. CNO never evoked a significant response, whereas L-Glut always did. Scale bar: 5 µm.

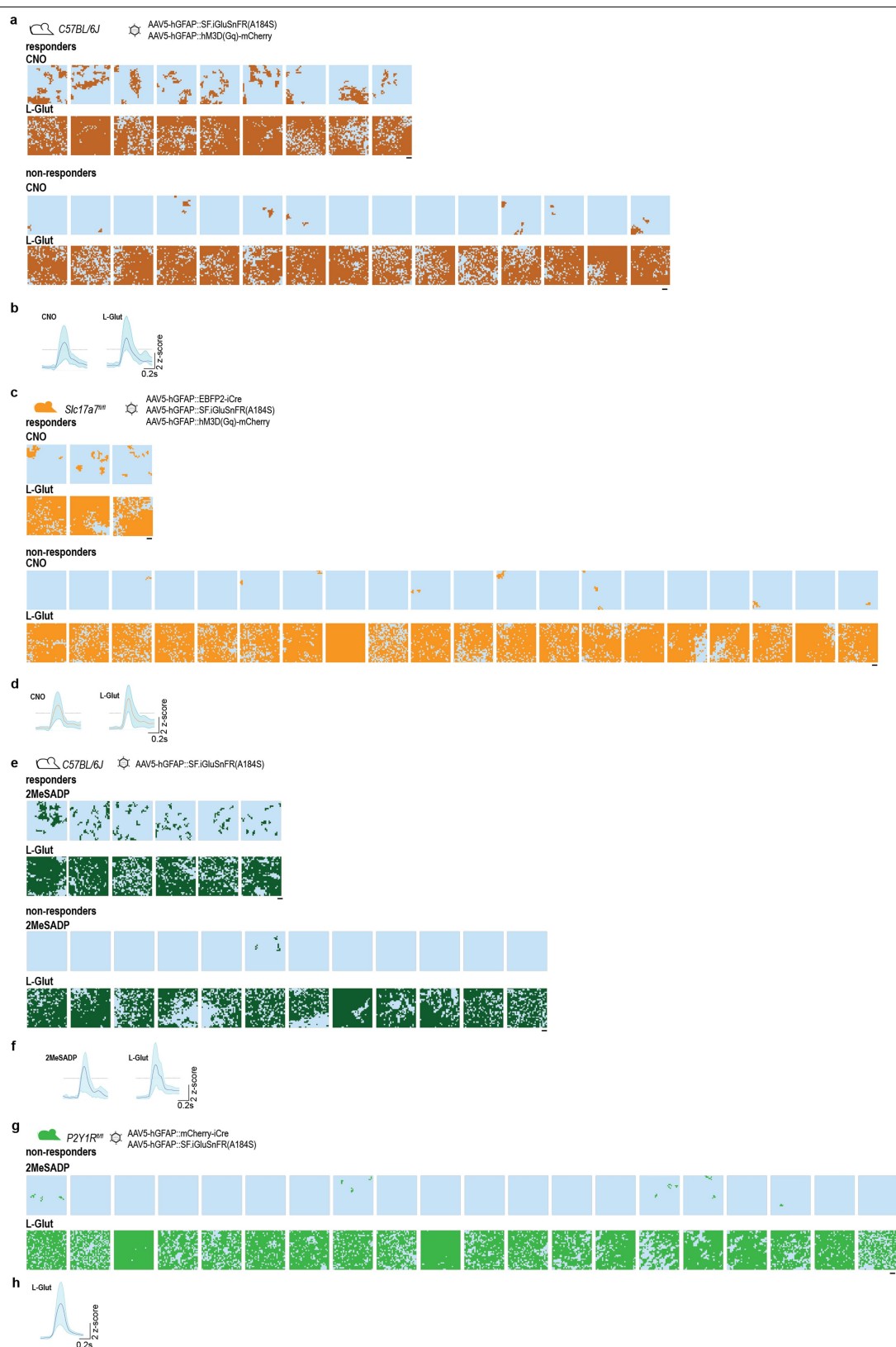

**Extended Data Fig. 6** | See next page for caption.

**Extended Data Fig. 6 | Astrocyte SF-iGluSnFR responses evoked by chemogenetic or endogenous Gq-GPCR activation and by L-Glutamate in all tested astrocytes from wild-type, VGLUT1$^{GFAP-KO}$ and P2Y1R$^{GFAP-KO}$ mice.** **a**–**d**, Chemogenetic astrocyte Gq-GPCR activation with CNO in wild-type (**a**,**b**) and VGLUT1$^{GFAP-KO}$ (**c**,**d**) mice. **a**, *Top*, schematic of the viral treatments in wild-type mice; *Middle and Bottom*, matched CNO- and L-Glut-evoked SF-iGluSnFR fluorescence responses (brown) in wild-type mice expressed as binarized functional maps for each individual astrocyte FOV (24 FOVs, n = 5 mice). *Middle*, individual FOVs with ≥5% CNO-responsive area within the L-Glut-responsive area were classified as responders (Methods; mean response: 15.12 ± 2.35%, n = 9). *Bottom*, FOVs with subthreshold responses or without response at all were collectively classified as non-responders (mean response: 0.84 ± 0.37%, n = 15). Responses to L-Glut were analogous in CNO-responding and non-responding astrocytes (77.77 ± 2.2% and 77.63 ± 4.7% of the total FOV, respectively). Scale bars: 5 µm. **b**, Mean kinetics ± s.e.m. (azure halo) of CNO- and L-Glut-evoked SF-iGluSnFR responses in wild-type mice. For CNO: rise-time$_{10-90}$: 93.08 ± 9.56 ms; full-width half-maximum (FWHM): 445.20 ± 57.34 ms; decay time: 352.10 ± 51.21 ms; ≥29 traces from 12 ± 2 responding grid locations from 9 FOVs; For L-Glut: >100 traces from 9 FOVs. **c**, *Top*, schematic of the viral treatments in *Slc17a7$^{fl/fl}$* mice; *Middle* and *Bottom*, matched CNO- and L-Glutamate (L-Glut-)-evoked SF-iGluSnFR fluorescence responses (orange) as in **a** but in VGLUT1$^{GFAP-KO}$ mice (23 FOVs, n = 5 mice); **d**, Mean kinetics ± s.e.m. of CNO- and L-Glut-evoked SF-iGluSnFR responses as in **b** but in VGLUT1$^{GFAP-KO}$ mice. For CNO: rise-time$_{10-90}$: 84.42 ± 16.04 ms; FWHM: 350.90 ± 45.66 ms; decay time: 266.5 ± 29.96 m; 11 ± 1 grid locations from 3 FOVs for both CNO and L-Glut responses. **e**–**h**, Activation of the endogenous Gq-GPCR P2Y1R with 2MeSADP in wild-type (**e**,**f**) and P2Y1R$^{GFAP-KO}$ (**g**,**h**) mice. **e**, *Top*, schematic of the viral treatments in wild-type mice; *Middle* and *Bottom*, matched 2MeSADP- and L-Glut-evoked SF-iGluSnFR fluorescence responses (dark green) in wild-type mice expressed as binarized functional maps for each individual astrocyte FOV (18 FOVs, n = 2 mice). 2MeSADP-responder (*Middle*) and non-responder (*Bottom*) FOVs classified as in **a**. For responses to 2MeSADP see Fig. 2p,k; and to L-Glut, Extended Data Fig. 4f. Scale bars: 5 µm. **f**, Mean kinetics ± s.e.m. (azure halo) of 2MeSADP- and L-Glut-evoked SF-iGluSnFR responses in wild-type mice. For 2MeSADP: rise-time$_{10-90}$: 96.78 ± 17.53 ms; FWHM: 400.39 ± 42.98 ms; decay time: 303.61 ± 30.56 ms; ≥32 traces from 7 ± 1 grid locations from 6 FOVs). For L-Glut: >100 traces from 6 FOVs. **g**, *Top*, schematic of the viral treatments in *GLAST$^{creERT2}$P2y1$^{fl/fl}$* mice; *Middle* and *Bottom*: matched 2MeSADP- and L-Glut-evoked SF-iGluSnFR fluorescence responses (light green) as in **e** but in P2Y1R$^{GFAP-KO}$ mice (20 FOVs, n = 2 mice). **h**, Mean kinetics ± s.e.m. of L-Glut-evoked SF-iGluSnFR responses as in **f** but in P2Y1R$^{GFAP-KO}$ mice (>100 traces from 10 FOVs). For 2MeSADP no kinetics of evoked responses are shown because no 2MeSADP-responder FOV was observed in P2Y1R$^{GFAP-KO}$ mice.

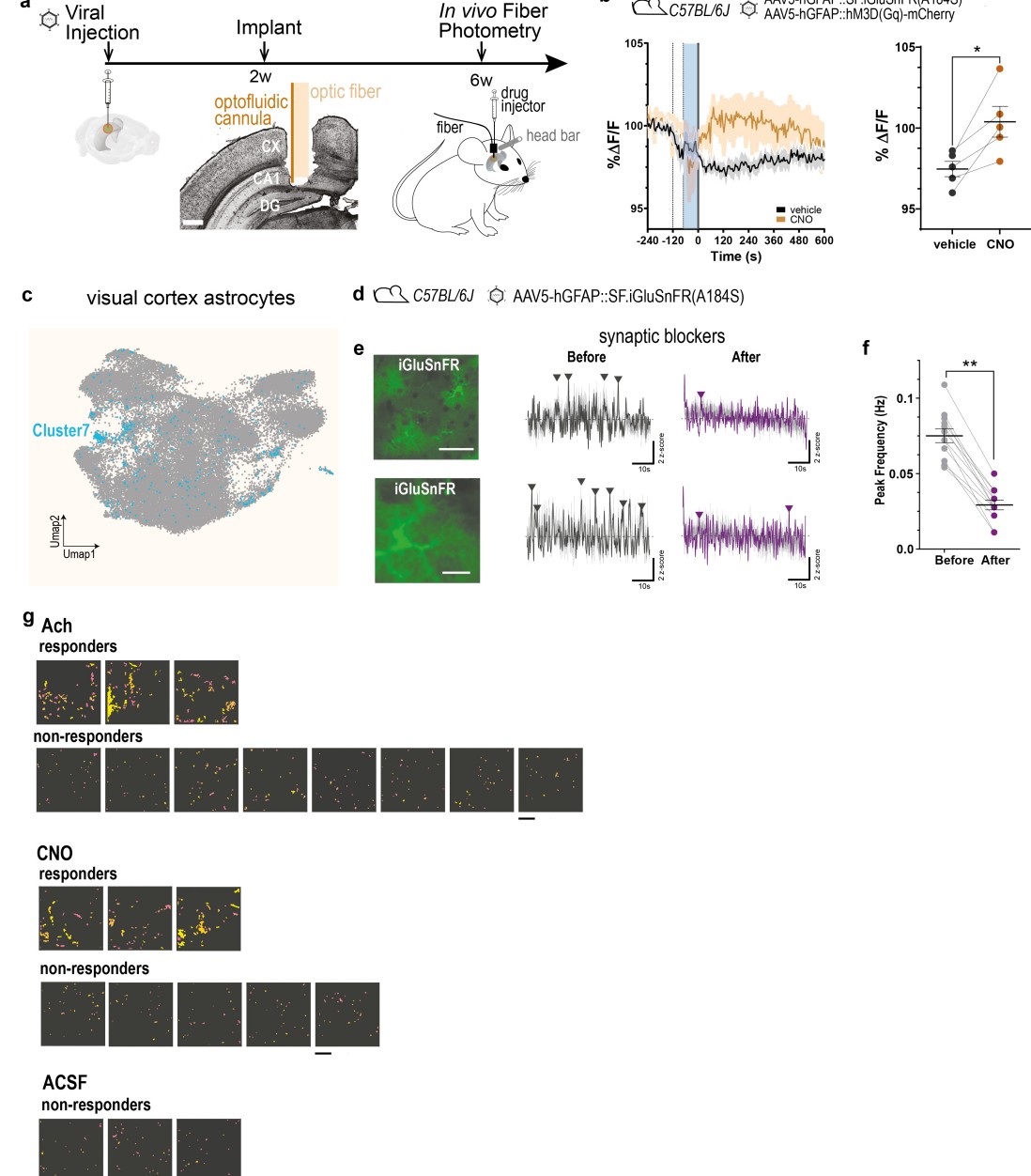

**Extended Data Fig. 7** | See next page for caption.

**Extended Data Fig. 7 | Astrocyte SF-iGluSnFR responses in awake mice recorded with fibre photometry in hippocampus and two-photon imaging in visual cortex, with additional information to Fig. 2. a**, Experimental paradigm for *in vivo* fibre photometry SF-iGluSnFR fluorescence measurements and local drug delivery through optofluid cannula positioned in the dorsal hippocampus, above DG. The whole-brain image is from the Allen Mouse Brain Connectivity Atlas (https://mouse.brain-map.org/). **b**, *Top*, viral injections for astrocyte expression of SF-iGluSnFR and Gq-DREADD in wild-type mice. *Bottom*, *left*: time course of averaged SF-iGluSnFR fluorescence responses to vehicle (black) and CNO (2.5 mM, brown), both in the presence of synaptic blockers mixture (Methods, n = 5 mice). Traces are aligned to the cannula plug (dotted line) and drug injection time (blue bar). Data are normalized to baseline and presented as mean ± s.e.m. *Right*, Normalized SF-iGluSnFR maximal fluorescence values in individual pairs at 3 minutes after application of vehicle and CNO. Lines represent mean ± s.e.m. (vehicle, black, mean: 97.5 ± 0.47; CNO, brown, mean: 100.4 ± 0.94). (*$P$ = 0.02, two-tailed paired t test). **c**, Visual cortex UMAP representation of 1 mouse, 1 macaque and 1 human integrated visual cortex scRNA-seq datasets (Methods) annotated with a neural network classifier trained on a comprehensive database[29] and subset for astrocyte population. Blue cells show the distribution of predicted cluster 7 according to the astrocyte reference annotation from the integrated astrocytic database in Fig. 1b. **d**–**f**, Two-photon imaging *in vivo* of the spontaneous SF-iGluSnFR activity in the visual cortex of the awake mouse before and after topical infusion of a synaptic blockers mixture (Methods). (n = 12 FOVs, 6 mice). **d**, Experiments performed in wild-type mice injected with *AAV5-GFAP::SF.iGluSNFR(A184S)* in the visual cortex **e**, *Top*, *left to right:* (*i*) mean projection of the SF-iGluSnFR

fluorescence signal in a representative large FOV (151 μm x 151 μm) containing multiple astrocytes (137 μm below surface); scale bar: 50 μm. (*ii*) Effect of the synaptic blockers: *left:* traces (grey, original; black, filtered) of mean SF-iGluSnFR activity from all ROIs in the FOV during the 60s acquisition before incubation with synaptic blockers (before); arrowheads point to identified SF-iGluSnFR activity peaks based on peak duration and z-score (Methods). *Right*, traces (grey, original; violet, filtered) and SF-iGluSnFR activity peaks detected after incubation with synaptic blockers (after). *Bottom*, from *left* to *right:* (*i*) mean projection of the SF-iGluSnFR fluorescence signal in a representative small FOV (37.5 x 37.5 μm) containing in this case a single astrocyte (137 μm below surface); scale bar: 10 μm. (*ii*) Effect of the synaptic blockers: descriptive as in *top* part of the panel. n = 12 FOV, 6 mice. **f**, Summary reporting SF-iGluSnFR mean peak frequency before (grey,: mean: 0.75 ± 0.04 Hz) and after infusion of synaptic blockers (violet: 0.29 ± 0.03 Hz) for each tested FOV. The effect of synaptic blockers was significant in all FOVs (Wilcoxon rank sum test, two sided, **$P$ = 0.0025 n = 12 FOV from 6 mice). Blockers mainly suppressed synchronized activity between cells and between ROIs within a cell, likely representing coordinate neuronal glutamate release responses to inherent patterns of cortical activity and inputs from other regions[42]. **g**, SF-iGluSnFR signal responses to Ach, CNO or ACSF in all astrocytes investigated *in vivo* in the visual cortex in 37.5 x 37.5 μm FOVs in the presence of synaptic blockers. Astrocytes are regrouped as responders or non-responders to the stimulus (Methods). For each astrocyte and for each stimulus is presented a colour-coded spatial map of the ROIs in the FOV displaying increased peak frequency upon stimulus application. The colour scale represents intensity of frequency increase above baseline, from 0 (pink) to 0.25 Hz (yellow); scale bar: 10 μm.

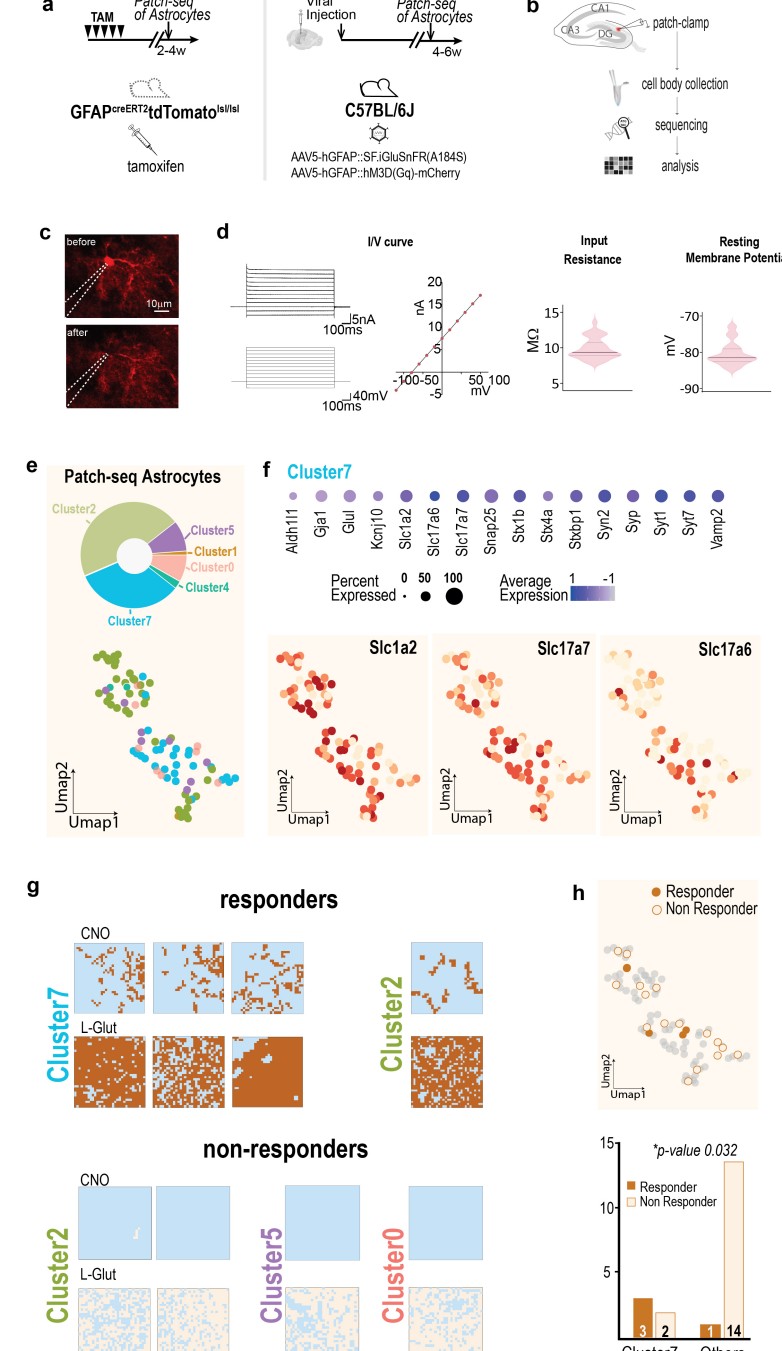

**Extended Data Fig. 8** | See next page for caption.

**Extended Data Fig. 8 | Patch-seq experiments on individual DGML astrocytes: astrocyte clusters prediction from scRNAseq and from combined glutamate imaging and transcriptomic information. a,b,** Patch-seq experiment on DGML astrocytes: workflow of the experimental procedure. **a,** *Left,* patch-seq in red-fluorescent astrocytes expressing tdTomato from *GFAP^creERT2^tdTom^lsl/lsl^* mice. *Right,* patch-seq preceded by SF-iGluSnFR imaging in astrocytes virally injected to express GqDREADD-mCherry and SF-iGluSnFR. Stimulations with CNO and L-Glut are like in Fig. 2a–e. The whole-brain image is from the Allen Mouse Brain Connectivity Atlas (https://mouse.brain-map.org/). **b,** schematic representation of the patch-seq procedure. **c,** Representative tdTomato-positive astrocyte before and after cell body collection by gentle aspiration (n = 65 cells), here imaged with two-photon microscope (n = 2 cells; see also Supplementary Video 2). **d,** Electrophysiological properties of individual patch-seq astrocytes recorded before collection: all cells showed linear current/voltage (I/V) curve, low input resistance and very negative membrane potential typical of astrocytes. **e,** UMAP representation of 85 patch-seq astrocytes predicted according to astrocyte reference annotation (cluster 0 to cluster 8 from integrated astrocytic database) and pie-chart distribution of the patch-seq astrocytes among each predicted cluster. Number of cells predicted per cluster were: cluster 0: 7; cluster 1: 1; cluster 2: 39; cluster 3: 0; cluster 4: 2; cluster 5: 8; cluster 6: 0; cluster 7: 28; cluster 8: 0. **f,** *Top,* Dot plot of selected marker genes related to astrocyte identity, vesicular trafficking, and glutamate regulated exocytosis for predicted cluster 7; *Bottom,* expression level for *Slc1a2, Slc17a7* and *Slc17a6* in the predicted astrocyte clusters. Note *Slc17a7* and *Slc17a6* enrichment in cluster 7. Noteworthy, cells assigned to cluster 7 had electrophysiological properties within the average of the whole patch-seq population (resting membrane potential: −79.4 ± 0.87 mV; input resistance: 9.9 ± 0.32 MΩ; linear I/V curve). **g,** Binarized functional maps of the SF-iGluSNFr signal response to CNO and L-Glut applications for the four astrocytes functionally identified as "responder" (brown), and for four representative astrocytes identified as "non responder" (sand), associated with the cluster prediction for each individual cell. **h,** *Top,* UMAP representation of the predicted cluster 7 for "responder" and "non responder" astrocytes, according to the astrocyte reference annotation from the integrated astrocytic database in Fig. 1b. *Bottom,* corresponding histogram quantification showing statistical significance (two tails Fisher exact test, $P = 0.0320$) for correct prediction of cluster 7 for "responder" astrocytes and of other clusters for "non responder" astrocytes. Overall, 3 out of 4 responders were correctly attributed to cluster 7, and one to cluster 2. Of the 16 non-responders, 14 were correctly attributed to non-glutamatergic clusters (9 to cluster 2; 3 to cluster 4 and 2 to cluster 5) and two to cluster 7.

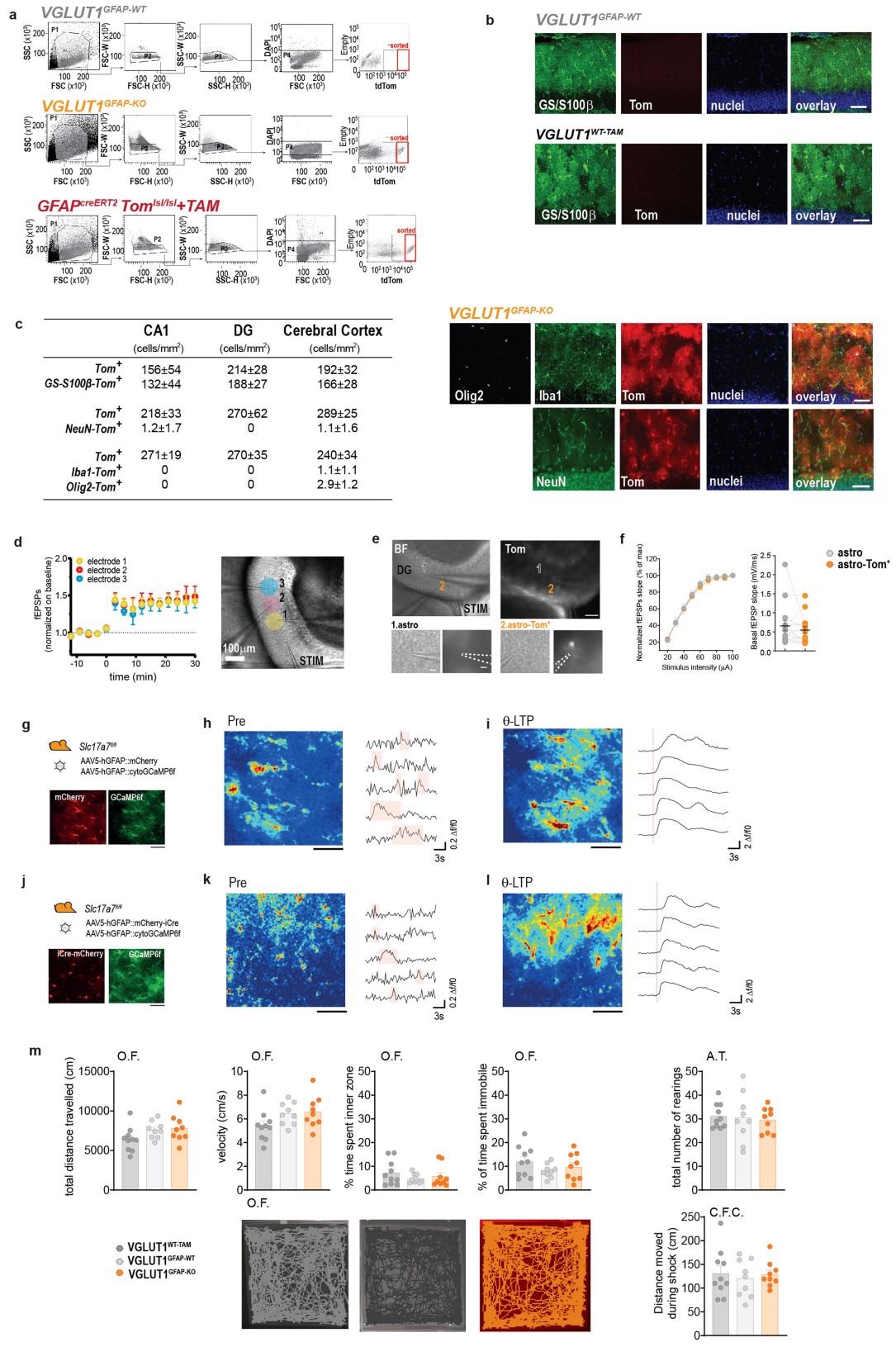

**Extended Data Fig. 9** | See next page for caption.

**Extended Data Fig. 9 | Additional data related to the *GFAP^creERT2^Slc17a7^fl/fl^ tdTom^lsl/lsl^* mouse model, the electrophysiology and the behavioural studies presented in Fig. 3. a**, Representative fluorescence activated cell sorting (FACS) of tdTomato-positive (Tom+) astrocytes in cerebral cortex samples of VGLUT1^GFAP-WT^ mice (sorted ≥2 x 10^5^ Tom+ cells per experiment, $n = 2$ independent experiments, 5 mice per experiment), VGLUT1^GFAP-KO^ mice (sorted ≥2 x 10^5^ Tom+ cells per experiment, $n = 2$ independent experiments, 5 mice per experiment) and *GFAP^CreERT2^tdTom^lsl/lsl^* mice (sorted ≥2 x 10^5^ Tom+ cells per experiment, $n = 2$ independent experiments, 5 mice per experiment). **b**, Representative images, here acquired with confocal microscope (n = 2), confirming no leakage in the absence of TAM-induced *cre* recombination, i.e., lack of any Tom+ cells (red) in the hippocampus of VGLUT1^GFAP-WT^ and VGLUT1^WT-TAM^ control mice also stained with the astrocyte markers GS and S100β (green), and the nuclear marker, DAPI (blue), $n = 8$ images from 4 independent experiments, 2 mice per group. Scale bar: 50 µm.**c**, *Left*, table presenting the total number per mm² of Tom+ cells and the relative numbers of the same Tom+ cells co-labelled with astrocyte (GS+S100β), neuron (NeuN), oligodendrocyte (Olig2) or microglia (Iba1) markers, counted in two hippocampal regions (CA1 and DG) and in the visual cortex of VGLUT1^GFAP-KO^ mice upon TAM-induced *cre* recombination. Data are presented as means ± s.e.m. *Right*, Confocal images confirming lack of any co-labelling of Tom+ cells with microglia (Iba1, green) oligodendrocyte (Olig2, white) or neuronal (NeuN, green) markers in the DG of VGLUT1^GFAP-KO^ mice. $n = 8$ images from 4 independent experiments, 2 mice per group. Scale bar: 50 µm. **d**, Θ-LTP recorded in DGML of wild-type mice by 3 local field potential (LFP) electrodes positioned along the same bundle of PP fibres at an average distance of 200 µm (electrode 1), 300 µm (electrode 2) and 400 µm (electrode 3) from the stimulation pipette (STIM). Θ-LTP magnitude is the same at all tested locations (two-way ANOVA repeated measures ($P = 0.78$, n = 6 slices, 3 mice). Data are means ± s.e.m. **e**, Setting for Θ-LTP induction and measure in *GFAP^creERT2^Slc17a7^fl/fl^tdTom^lsl/lsl^* mice undergone short TAM treatment (Methods). *Top*, bright-field (BF) and fluorescence images (Tom) show positioning of the stimulation pipette (STIM) and of the two LFP recording electrodes in the DGML, with about 200 µm interdistance. Scale bar: 200 µm. *Bottom*, higher zoom images show the position of electrode 1, proximal to a non-fluorescent astrocyte (astro), and of electrode 2, proximal to a fluorescent astrocyte (astro Tom+). $n = 16$ slices, 12 mice. Scale bar: 50 µm. **f**, Basal input-output curves (*left*) and basal fEPSP amplitudes (*right*) recorded in two DGML fields containing, respectively, a VGLUT1^GFAP-WT^ (grey) and a VGLUT1^GFAP-KO^ astrocyte (orange), show no significant differences (data mean ± s.e.m.; paired Student's t test, two tails, $P = 0.337$). Thin lines connect individual LFP electrode pairs ($n = 16$ slices, 12 mice). **g–l**, Astrocyte Ca²⁺ dynamics during low and high-frequency stimulation of MPP in VGLUT^GFAP-WT^ and VGLUT1^GFAP-KO^

mice. **g**, *Top*, in control experiments, *Slc17a7^fl/f^* mice are injected with *AAV5-GFAP-mCherry* virus (control virus) and *AAV5-GFAP-GCaMP6f* virus to report astrocyte Ca²⁺ dynamics. *Bottom*, multiple astrocytes present in the same FOV as in **h** and **i** display both mCherry (red) and GCaMP6f (green) fluorescence. Scale bar: 50 µm. $n = 2$ FOVs, 2 mice. **h**, *Left*, mean time projection over 70 frames (21 s) of the GCaMP6f signal in astrocytes in the period before Θ-LTP induction (Pre). *Right*, representative Ca²⁺ traces from selected single-cell ROIs during the same Pre period. Astrocytes show small asynchronous local Ca²⁺ activity and a few larger responses to single MPP stimulations. **i**, *Left*, mean time projection of the GCaMP6f signal in astrocytes as in **h** but during MMP stimulation inducing Θ-LTP (Θ-LTP). *Right*, representative Ca²⁺ traces from single astrocyte ROIs during Θ-LTP induction. Multiple astrocytes show very large Ca²⁺ elevation (note the scale is 10-fold larger than in the Pre period) almost synchronously at the start of the Θ-LTP protocol (red vertical line). **j**, *Top*, in experiments in VGLUT1^GFAP-KO^ mice, *Slc17a7^fl/fl^* mice are injected with *AAV5-GFAP-mCherry-iCre* virus to delete VGLUT1 selectively in astrocytes, and with *AAV5-GFAP-GCaMP6f* virus to report astrocyte Ca²⁺ dynamics. *Bottom*, multiple astrocytes, present in the same FOV as in **k** and **l**, display both mCherry fluorescence (red) indicating *Cre* recombination and GCaMP6f fluorescence (green). Scale bar: 50 µm. n = 2 FOVs, 2 mice.**k** *Left*, mean time projection over 80 frames (24 s) of the GCaMP6f signal in astrocytes in the Pre period. *Right*, representative Ca²⁺ traces from selected single-cell ROIs in the same Pre period. Astrocyte Ca²⁺ dynamics in VGLUT1^GFAP-KO^ mice in the Pre period are comparable to those in controls mice (**h**). **l**, *Left*: mean time projection of the GCaMP6f signal in astrocytes as in **k** but in the Θ-LTP induction period. *Right*, representative Ca²⁺ traces from selected single-cell ROIs during Θ-LTP induction. The very large synchronous astrocyte Ca²⁺ responses in VGLUT1^GFAP-KO^ mice are comparable to those in controls. **m**, Open field (O.F.) and activity tests (A.T.) performed on VGLUT1^GFAP-KO^ (orange, n = 9 mice), VGLUT1^GFAP-WT^ (light grey, n = 9 mice) and VGLUT1^WT-TAM^ (dark grey, n = 10 mice) mice. *Left, top*, histograms reporting parameters index of locomotor activity (total distance travelled, $P = 0.102$; and velocity, $P = 0.086$) and anxiety (time spent immobile, $P = 0.28$; and time in the inner zone of the arena, $P = 0.28$) do not show group differences. *Bottom*, example traces of locomotor activity in the three mouse groups placed in the O.F. for 20 min. *Right, top*: Histograms reporting parameters index of exploratory activity (total number of rearings measured during the 5 min activity test preceding fear conditioning) do not show group differences ($P = 0.89$). *Right, bottom*, histograms reporting parameters index of pain sensitivity (mean total distance moved during 2s e-shocks repeated 6 times during the fear conditioning test) do not show group differences (P = 0.84). Data presented as mean ± s.e.m. One-way ANOVA with Tukey test.

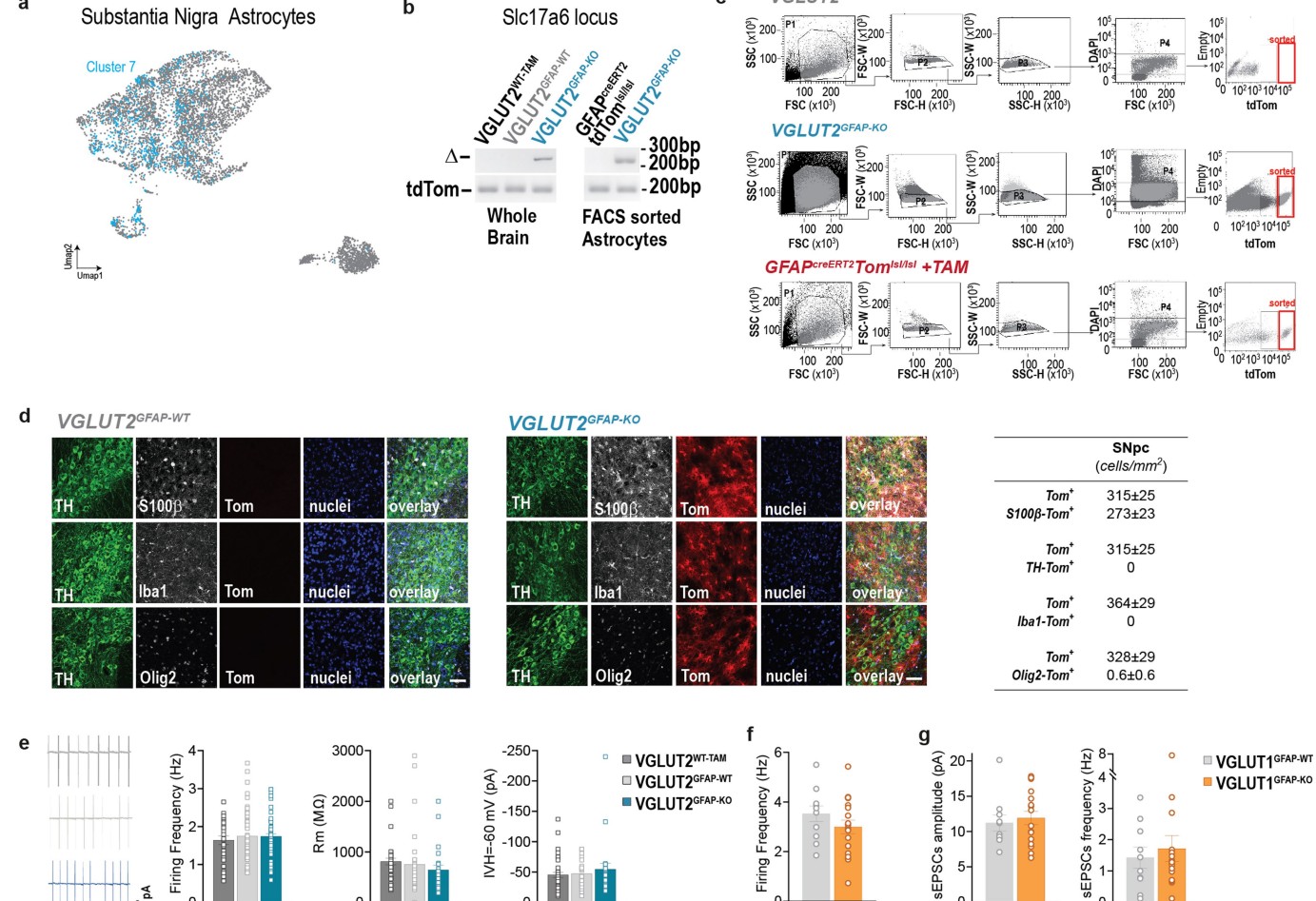

**Extended Data Fig. 10 | Additional data related to the *GFAP^creERT2 Slc17a6^fl/fl tdTom^lsl/lsl* mouse model, and the electrophysiology studies presented in Fig. 4. a**, UMAP representation of 2 integrated human and 1 mouse substantia nigra scRNA-seq datasets (Methods) annotated with a neural network classifier trained on a comprehensive database[29]. The UMAP represents the distribution of predicted cluster 7 (blue) according to astrocyte reference annotation from the integrated astrocytic database in Fig. 1b. **b**, Genomic PCR to validate deletion of the *Slc17a6 locus* (Δ) in VGLUT2^GFAP-KO^ mice. *Left*, validation on whole brain homogenates of VGLUT2^GFAP-KO^, and of VGLUT2^GFAP-WT^ and VGLUT2^WT-TAM^ controls (n = 2 per group). *Right*, validation on FACS-sorted astrocytes from the midbrain region of VGLUT2^GFAP-KO^ and of *GFAP^creERT2 tdTom^lsl/lsl* controls (n = 2 per group). **c**, Representative fluorescence activated cell sorting of tdTomato positive (Tom+) astrocytes in midbrain samples of VGLUT2^GFAP-WT^ mice (sorted ≥2 x 10⁵ Tom+ cells per experiment, *n* = 2 independent experiments, 5 mice per experiment), VGLUT2^GFAP-KO^ mice (sorted ≥2 x 10⁵ Tom+ cells per experiment, *n* = 2 independent experiments, 5 mice per experiment) and *GFAP^CreERT2 tdTom^lsl/lsl* mice (sorted ≥2 x 10⁵ Tom+ cells per experiment, *n* = 2 independent experiments, 5 mice per experiment) See Methods for details. **d**, *Left*, Representative images, here acquired with confocal microscope, confirming no leakage in the absence of TAM-induced *cre* recombination, that is, lack of any Tom+ cells (red) in the SNpc of VGLUT2^GFAP-WT^ and VGLUT2^WT-TAM^ control mice also stained with the neuronal marker TH (green), the astrocyte marker S100β (grey), the oligodendrocyte marker Olig2 (grey) or the microglia marker Iba1 (grey), and the nuclear marker, DAPI (blue). Scale bar: 50 μm. *Middle*, confocal images confirming co-labelling of Tom+ cells (red) with the astrocyte marker S100β (grey) but not with the neuronal (TH, green) microglia

(Iba1, grey) or oligodendrocyte (Olig2, grey) markers in the SNpc of VGLUT2^GFAP-KO^ mice. Scale bar: 50 μm. *Right*, table presenting the total number of Tom+ cells and the relative numbers of the same Tom+ cells co-labelled with astrocyte (S100β), neuron (TH), oligodendrocyte (Olig2) or microglia (Iba1) markers, counted in SNpc of VGLUT2^GFAP-KO^ mice after TAM-induced *cre* recombination. *n* = 12 images, 2 independent experiments, 3 mice. **e**, *Left:* representative cell-attached firing traces and *Right*, histograms of basal electrophysiological properties (firing frequency, membrane resistance (R_m) and holding current at −60 mV) of SNpc DA neurons in midbrain slices from VGLUT2^GFAP-KO^, VGLUT2^GFAP-WT^ and VGLUT2^WT-TAM^ mice. Data are presented as mean ± s.e.m. (firing frequency: VGLUT2^GFAP-KO^, *n* = 59 cells, 9 mice; VGLUT2^GFAP-WT^, *n* = 58 cells, 8 mice; VGLUT2^WT-TAM^, *n* = 26 cells, 8 mice; R_m and I_hold at −60 mV: VGLUT2^GFAP-KO^, *n* = 23 cells, 9 mice; VGLUT2^GFAP-WT^, *n* = 29 cells, 8 mice; VGLUT2^WT-TAM^ mice, *n* = 38 cells, 8 mice). No differences among groups were observed: one-way ANOVA: *P* = 0.7399 for firing frequency; *P* = 0.47 for R_m; *P* = 0.516 for holding current at −60 mV. **f**, Plot of spontaneous firing frequency recorded in cell-attached mode in SNpc DA neurons of VGLUT1^GFAP-KO^ (*n* = 18 cells, 3 mice) and VGLUT1^GFAP-WT^ (*n* = 11 cells, 3 mice). No significant differences were found between the two groups: P = 0.228, unpaired Student's *t* test, two tails. Data are presented as mean ± s.e.m. **g**, Histograms of frequency and amplitude of spontaneous excitatory postsynaptic currents (sEPSCs) recorded in SNpc DA neurons of VGLUT1^GFAP-KO^ (n = 17 cells, 3 mice) and VGLUT1^GFAP-WT^ (n = 10 cells, 3 mice). Data are presented as mean ± s.e.m. and show no differences between the two groups. P = 0.63 for sEPSC frequency and P = 0.64 for sEPSC amplitude, unpaired Student's *t* test, two-tails.

# Reporting Summary

## Statistics

For all statistical analyses, confirm that the following items are present in the figure legend, table legend, main text, or Methods section.

| n/a | Confirmed | |
|---|---|---|
| ☐ | ☒ | The exact sample size (*n*) for each experimental group/condition, given as a discrete number and unit of measurement |
| ☐ | ☒ | A statement on whether measurements were taken from distinct samples or whether the same sample was measured repeatedly |
| ☐ | ☒ | The statistical test(s) used AND whether they are one- or two-sided *Only common tests should be described solely by name; describe more complex techniques in the Methods section.* |
| ☒ | ☐ | A description of all covariates tested |
| ☐ | ☒ | A description of any assumptions or corrections, such as tests of normality and adjustment for multiple comparisons |
| ☐ | ☒ | A full description of the statistical parameters including central tendency (e.g. means) or other basic estimates (e.g. regression coefficient) AND variation (e.g. standard deviation) or associated estimates of uncertainty (e.g. confidence intervals) |
| ☐ | ☒ | For null hypothesis testing, the test statistic (e.g. *F*, *t*, *r*) with confidence intervals, effect sizes, degrees of freedom and *P* value noted *Give P values as exact values whenever suitable.* |
| ☒ | ☐ | For Bayesian analysis, information on the choice of priors and Markov chain Monte Carlo settings |
| ☒ | ☐ | For hierarchical and complex designs, identification of the appropriate level for tests and full reporting of outcomes |
| ☒ | ☐ | Estimates of effect sizes (e.g. Cohen's *d*, Pearson's *r*), indicating how they were calculated |

*Our web collection on statistics for biologists contains articles on many of the points above.*

## Software and code

Policy information about availability of computer code

| Data collection | Data collection, such as image acquisitions, electrophysiological recording etc. were performed with the specific instrument softwares installed on the instruments, as detailed in the methods. |
|---|---|
| Data analysis | Data analysis was performed with the following softwares:<br>Image analysis: ImageJ-Fiji (version 1.53, https://imagej.nih.gov/ ), LAS X (version 3.7.4.23463 LeicaMicrosystems).<br>scRNA seq data: Seurat 4 in R 4.0.5, GSEA (http://software.broadinstitute.org/gsea/index.jsp)<br>Flow Cytometry: BD FACSDiva 8.0.1, FlowingSoftware 2.5.1.<br>Behaviour: Ethovision XT 11<br>Electrophysiology: in vivo , Sirenia seizure (v1.7, Pinnacle), ex vitro, Clampex and Clampfit (v10.3)<br>statistic, analysis and graphs: Origin Pro (2022, Origin Lab), Imaris v9.1.1 (Bitplane), Python (Python.org), GraphPad Prism 9 (GraphPad), Adobe Illustrator and Adobe  photoshop (Adobe 2023), Microsoft excel, Matlab 2019b, Spike2 version 8<br>Resources used for custom Python v3.7.6 virtual environment code:<br>Jupyter Notebook http://jupyter.org (6.4.12)<br>NumPy http://www.numpy.org (NumPy v1.19.5)<br>Scikit https://scikit-image.org (v1.2.0)<br>Holoviews https://holoviews.org/ (1.15.4)<br>Neurokit2 https://neurokit2.readthedocs.io/en/latest (0.1.6)<br>Scipy.Signal (SciPy v1.10.0) Virtanen et al., 2020 Nature Methods; https://docs.scipy.org/doc/scipy/index.html<br><br>Details of the software codes used are described in the method section and their availability is as stated in the "Code availability" section in the manuscript<br>HDF5Array_1.28.1 ; rhdf5_2.44.0 ; DelayedArray_0.26.3 ; S4Arrays_1.0.4; patchwork_1.1.2; reticulate_1.28 ; Matrix_1.5-4.1; cowplot_1.1.1; ggExtra_0.10.0; ggplot2_3.4.2; dplyr_1.1.2; wesanderson_0.3.6 ; RColorBrewer_1.1-3; Seurat_4.9.9.9042; SeuratObject_4.9.9.9084; bmrm_4.4; SummarizedExperiment_1.30.1; Biobase_2.60.0; GenomicRanges_1.52.0; GenomeInfoDb_1.36.0; IRanges_2.34.0 ; |

S4Vectors_0.38.1 ; BiocGenerics_0.46.0 ; MatrixGenerics_1.12.0 ; matrixStats_0.63.0 ; torch_0.10.0
Details of the software codes used are described in the method section and their availability is as stated in the "Code availability" section in the manuscript

For manuscripts utilizing custom algorithms or software that are central to the research but not yet described in published literature, software must be made available to editors and reviewers. We strongly encourage code deposition in a community repository (e.g. GitHub). See the Nature Portfolio guidelines for submitting code & software for further information.

## Data

Policy information about availability of data

All manuscripts must include a data availability statement. This statement should provide the following information, where applicable:

- Accession codes, unique identifiers, or web links for publicly available datasets
- A description of any restrictions on data availability
- For clinical datasets or third party data, please ensure that the statement adheres to our policy

We have included a data availability statement in the manuscript. Notably, the new single-cell RNAseq datasets generated during and/or analyzed during the current study are made available for download on Zenodo repository (10.5281/zenodo.7704838). The already published and available datasets analyzed during the current study and their public domain resources are indicated in Extended Figure 1a and in Methods:
GSE106447 "Artegiani" – https://doi.org/10.1016/j.celrep.2017.11.050;
GSE114000 "Batiuk" - https://doi.org/10.1038/s41467-019-14198-8;
GSE143758 "Habib" - https://doi.org/10.1038/s41593-020-0624-8;
 GSE95753 "Hochgerner" - https://doi.org/10.1038/s41593-017-0056-2;
SRP135960 "Zeisel-1" https://doi.org:https://doi.org/10.1016/j.cell.2018.06.021;
 "Saunders" - http://dropviz.org;
GSE60361 "Zeisel-2" - https://doi.org:doi.org/10.1126/science.aaa1934;
"Yao" - https://portal.brain-map.org/atlases-and-data/rnaseq/mouse-whole-cortex-and-hippocampus-10x
"Habib Human" - https://www.gtexportal.org/home/datasets
GSE160189 "Ayhan" - https://doi.org/10.1016/j.neuron.2021.05.003;
"Tran" - https://github.com/LieberInstitute/10xPilot_snRNAseq-human
GSE190940 "Zipursky" - https://doi.org/10.1016/j.cell.2021.12.022;
EMBL-EBI repository E-MTAB-10459 "Liu" - https://www.ebi.ac.uk/biostudies/arrayexpress/studies/E-MTAB-10459;
GSE97930 "Zhang" - https://doi.org:10.1038/nbt.4038;
GSM4157078 "Agarwal" - https://doi.org/10.1038/s41467-020-17876-0;
GSE126836 "Welch" - https://doi.org:https://doi.org/10.1016/j.cell.2019.05.006.

# Field-specific reporting

Please select the one below that is the best fit for your research. If you are not sure, read the appropriate sections before making your selection.

☒ Life sciences　　☐ Behavioural & social sciences　　☐ Ecological, evolutionary & environmental sciences

For a reference copy of the document with all sections, see nature.com/documents/nr-reporting-summary-flat.pdf

# Life sciences study design

All studies must disclose on these points even when the disclosure is negative.

| Sample size | The nature of the n is described for each experiment in the corresponding figure legends. Sample size determinations are based on previous experience (Habbas et al., Cell, 2015, XDi Castro et al., Nat. Neurosic., 2011) and standards in the field (Rusina et al.,eNeuro,2021; D'Amour etal. Exp. Neurol.,2015). The low variability between the same type of samples, as indicated by the SEM, confirms that the sampling was sufficient to observe statistically significant differences between groups. Where variability was expected to be higher (such as for instance in in vivo experiments), up to 13 biological replicates were included. |
|---|---|
| Data exclusions | Patchseq: low quality RNAseq cells were eliminated as described in the method section. EEG recording: mice that detached from the recording system during the experiments or died after KA administration were excluded. Behavior: mice that in the Inter E-shock Interval 5-6 did not reach 40% of freezing or went over 70% were excluded as used in Contextual fear conditioning experiments. This range represents the optimal window for being in the position to observe either increments or decrements in performance potentially induced by genetic or pharmacological interference. Glutamate imaging analysis: FOV that did not respond to glutamate as positive control were excluded. in vivo fiber photometry fluorescence measurements: excluded mice that have no stable signal during the baseline experiment in vivo two-photon experiments: FOV with artifacts due to mouse movements were excluded. for details see in vivo imaging "Methods" section |
| Replication | Experiments were repeated as indicated in detail for each figure panel and as described in the methods. Reported findings were reproduced across animals in EEG recordings, behavioural and microdialysis experiments as well as across cells/animals in patchseq, imaging and electrophysiology experiments. In patchseq experiments, each single-cell RNA sequencing pool contained cells from different collection days and conditions to minimize batch effect. The total number of animals and cells/FOV is reported for all experiments. The replications are shown as individual dots together with the |

| | calculated means and variability (mean +/- sem). |
|---|---|
| Randomization | In all experiments using littermate mice with different pharmacological treatments, animals were randomized in the various groups to avoid cage, litter and batch effects. |
| Blinding | Data acquisition and analysis were done blind when possible. Cell clusters, based on gene expression patterns, were identified computationally without input from a trained neuroscientist. Independent researchers were 'blinded' to each others work. |

# Reporting for specific materials, systems and methods

We require information from authors about some types of materials, experimental systems and methods used in many studies. Here, indicate whether each material, system or method listed is relevant to your study. If you are not sure if a list item applies to your research, read the appropriate section before selecting a response.

## Materials & experimental systems

| n/a | Involved in the study |
|---|---|
| ☐ | ☒ Antibodies |
| ☒ | ☐ Eukaryotic cell lines |
| ☒ | ☐ Palaeontology and archaeology |
| ☐ | ☒ Animals and other organisms |
| ☒ | ☐ Human research participants |
| ☒ | ☐ Clinical data |
| ☒ | ☐ Dual use research of concern |

## Methods

| n/a | Involved in the study |
|---|---|
| ☒ | ☐ ChIP-seq |
| ☐ | ☒ Flow cytometry |
| ☒ | ☐ MRI-based neuroimaging |

## Antibodies

| Antibodies used | All antibodies (provider, ordering numbers and dilution) used in this study are listed in supplementary Table 3.<br><br>guine pig anti-Iba1 Synaptic System 234004  2-28  1:500<br>mouse anti-Cre Recombinase Merk millipore MAB3120 JC1631396  1:500<br>mouse anti-S-100b SIGMA S2532 048m4858v  1:500<br>mouse anti-NeuN Merk millipore MAB377 1991263  1:500<br>mouse anti-glutamine synthetase Merk millipore MAB302 2676275  1:500<br>mouse  anti-Olig2 Merk millipore MABN50 3421971  1:500<br>rabbit anti-glutamine synthetase Abcam ab 73593 gr3200078-1  1:500<br>rabbit anti-S100b Synaptic system S287003  1-5  1:500<br>rabbit  anti-Olig2 NovusBio NBP1-28667 DF1210  1:100<br>rabbit Anti-Tyrosine Hydroxylase Merk millipore AB152 3114503  1:200<br>chicken anti-S100b Synaptic system 287006 287006/1-4  1:500<br>goat anti-TdTomato BioSource cat. n. MBS448092 lot: 0081191218 1:500<br><br>Goat anti-Mouse IgG (H+L) Highly Cross-Adsorbed Secondary Antibody, Alexa Fluor Plus 405 Thermofisher Cat # A48255 51912A 1:500<br>Goat anti-Mouse IgG (H+L) Highly Cross-Adsorbed Secondary Antibody, Alexa Fluor Plus 488 Thermofisher Cat # A-32723 TC252656 1:500<br>Goat anti-Mouse IgG (H+L) Highly Cross-Adsorbed Secondary Antibody, Alexa Fluor 633 Thermofisher Cat # A-21052 1906490  1:500<br>Goat anti-Rabbit IgG (H+L) Highly Cross-Adsorbed Secondary Antibody, Alexa Fluor Plus 488 Thermofisher Cat # A-11034 SH251139 1:500<br>Goat anti-Rabbit IgG (H+L) Highly Cross-Adsorbed Secondary Antibody, Alexa Fluor 633 Thermofisher Cat # A-21071 1932492  1:500<br>Goat anti-Guinea Pig IgG (H+L) Highly Cross-Adsorbed Secondary Antibody, Alexa Fluor 488 Thermofisher Cat # A-11073 1458631 1:500<br>Goat anti-Guinea Pig IgG (H+L) Highly Cross-Adsorbed Secondary Antibody, Alexa Fluor 633 Thermofisher Cat # A-21105 514962 1:500 |
|---|---|
| Validation | All antibodies used in this study are from commercial suppliers (see notes above for each antibody) that have verified the specificity of the antibodies. All the antibodies have been previously used by various laboratories. All secondary antibodies are verified to not give a specific staining without the primary antibody.<br><br>Primary antibodies:<br>mouse anti-Cre Recombinase Merck millipore MAB3120 JC1631396  1:500 - Manufacturer:Proven to reliably detect Cre Recombinase, this mAb is validated for use in ELISA, IC, IF, IH & WB and is backed by multiple publications.<br>mouse anti-S-100b SIGMA S2532 048m4858v  1:500 -Manufacturer:These antibodies have been verified by Relative Expression to confirm specificity to S100B.<br>mouse anti-NeuN Merck millipore MAB377 1991263  1:500 -  Manufacturer: it detects level of NeuN and has been published and validated for use in FC, IC, IF, IH, IH(P), IP and WB. Positive control -Brain Tissue. Negative control - Any non neuronal tissue eg Fibroblasts<br>mouse anti-glutamine synthetase Merck millipore MAB302 2676275  1:500 -Manufacturer: Detect Glutamine Synthetase using this Anti-Glutamine Synthetase Antibody, clone GS-6 validated for use in ELISA, IH, IH(P) & WB with more than 45 product citations.Controlled in Rat brain tissue, rat brain cytosolic fraction extract<br>mouse  anti-Olig2 Merk millipore MABN50 3421971  1:500 - Manufacturer:clone 211F1.1, from mouse.controlled on mouse brain |

samples.
rabbit anti-glutamine synthetase Abcam ab 73593 gr3200078-1  1:500 -Manufacturer:Suitable for: IHC-P, ICC/IF, WB. Species reactivity: Mouse, Rat, Human, Common marmoset.
rabbit anti-S100b Synaptic system S287003  1-5  1:500  -Manufacturer: Reacts with: rat (P04631), mouse (P50114).
rabbit  anti-Olig2 NovusBio NBP1-28667 DF1210  1:100 -Manufacturer: Reacts with: human, rat, mouse. Validated for use in  IH and IH(P) with more than 25 product citations.
rabbit Anti-Tyrosine Hydroxylase Merk millipore AB152 3114503  1:200 -Manufacturer:validated for use in ELISA, IH, IH(P) & WB with more than 35 product citations.Controlled in Human, Rat, Mouse, Drosophila,Cat, Ferret, Squid, Mollusc.
chicken anti-S100b Synaptic system 287006/1-4  1:500 -Manufacturer: Reacts with: rat (P04631), mouse (P50114). Validated for use in IHP, ICC,IHC.
goat anti-TdTomato  BioSource cat. n. MBS448092 lot: 0081191218  1:500 -Manufacturer:validated for use in  IF, IHC-P, IHC-F & WB

Secondary antibodies:
Goat anti-Mouse IgG (H+L) Highly Cross-Adsorbed Secondary Antibody, Alexa Fluor Plus 405 Thermofisher Cat # A48255 51912A 1:500
Goat anti-Mouse IgG (H+L) Highly Cross-Adsorbed Secondary Antibody, Alexa Fluor Plus 488 Thermofisher Cat # A-32723 TC252656 1:500
Goat anti-Mouse IgG (H+L) Highly Cross-Adsorbed Secondary Antibody, Alexa Fluor 633 Thermofisher Cat # A-21052 1906490  1:500
Goat anti-Rabbit IgG (H+L) Highly Cross-Adsorbed Secondary Antibody, Alexa Fluor Plus 488 Thermofisher Cat # A-11034 SH251139 1:500
Goat anti-Rabbit IgG (H+L) Highly Cross-Adsorbed Secondary Antibody, Alexa Fluor 633 Thermofisher Cat # A-21071 1932492  1:500
Goat anti-Guinea Pig IgG (H+L) Highly Cross-Adsorbed Secondary Antibody, Alexa Fluor 488 Thermofisher Cat # A-11073 1458631 1:500
Goat anti-Guinea Pig IgG (H+L) Highly Cross-Adsorbed Secondary Antibody, Alexa Fluor 633 Thermofisher Cat # A-21105 514962 1:500
Alexa Fluor-568 dk anti-goat IgG H+L Invitrogen cat. n. A11057 2421197  1:500
Alexa Fluor Plus-647 dk anti-mouse IgG H+L Invitrogen cat. n. A32787 TJ271040  1:500
guine pig anti-Iba1 Synaptic System 234004  2-28  1:500 - Manufacturer: Reacts with: mouse (Q9EQW9), rat (P55009), human (P55008).
To minimize cross-reactivity, Highly Cross-adsorbed secondary antibodies where preferred. These antibodies have been highly cross-adsorbed against bovine IgG, goat  IgG, mouse IgG, rat IgG, and human IgG. Cross-adsorption or pre-adsorption is a purification step to increase specificity of the antibody resulting in higher sensitivity and less background staining. The secondary antibody solution is passed through a column matrix containing immobilized serum proteins from potentially cross-reactive species. Only the nonspecific-binding secondary antibodies are captured in the column, and the highly specific secondaries flow through. Further passages through additional columns result in 'highly cross-adsorbed' preparations of secondary antibody. The benefits of these extra steps are apparent in multiplexing/multicolor-staining experiments where there is potential cross-reactivity with other primary antibodies or in tissue/cell fluorescent staining experiments where there may be the presence of endogenous immunoglobulins.
Alexa Fluor dyes are among the most trusted fluorescent dyes available today.

Simultaneous staining for the following sets of antibodies were performed to validate cross-identification of the same protein:
-mouse, rabbit and chicken anti-S100b
-mouse and rabbit Glutamine Synthetase
-mouse and rabbit anti Olig2

# Animals and other organisms

Policy information about studies involving animals; ARRIVE guidelines recommended for reporting animal research

| Laboratory animals | original mouse lines used to generate the mouse lines used in this study:<br>hGFAP-CreERT2, Tg(GFAP-cre/ERT2)1Fki, ref. https://onlinelibrary.wiley.com/doi/10.1002/glia.20342<br>Slc17a6fl/fl,  ref. https://www.ncbi.nlm.nih.gov/pmc/articles/PMC2846457/<br>Slc17a7fl/fl,  https://doi.org:10.1101/2021.06.19.449108 ref.<br>GLASTcreERT2P2ry1fl/fl https://doi.org:https://doi.org/10.1002/glia.22999<br>Rosa26-loxP-stop-loxP-tdTomato reporter mice (Ai14), Jackson Stock No: 007908 - B6;129S6-Gt(ROSA)26Sortm14(CAG-tdTomato)Hze/J<br>B6;-Gt(ROSA)26Sortm95.1(CAG-GCaMP6f)Hze/J, JAX 024105<br>C57/Bl6 mice, Janvier, France<br><br>derived mouse lines were used in this study:<br>33 male 2-4 months C57/Bl6 mice, Janvier, France<br>27 male  1-3 months hGFAP-CreERT2, Rosa26-loxP-stop-loxP-tdTomato<br>105 male 1-5 months hGFAP-CreERT2, Slc17a7fl/fl,Rosa26-loxP-stop-loxP-tdTomato<br>114 male 1-3 months hGFAP-CreERT2, Slc17a6fl/fl,Rosa26-loxP-stop-loxP-tdTomato<br>2 male 4-5  months GLASTcreERT2P2ry1fl/fl<br>2 male 4-5 months hGFAP-CreERT2, Rosa26-loxP-stop-loxP-GCaMP6f<br><br>Housing conditions were as following: dark/light cycle 12/12, ambient temperature around 21-22°C and humidity between 40 and 70% (55% in average). |
|---|---|
| Wild animals | The study did not involve wild animals. |
| Field-collected samples | The study did not involve field-collected samples |
| Ethics oversight | All experiments including animals were carried out in compliance with the Swiss Federal and Cantonal authorities (authorizations: |

| Ethics oversight | VD1873.1, VD2982, VD3053.1, VD3115.1) or by the Council Directive of the European Communities (2010/63/EU), and the Animal Care Committee of Italian Ministry of Health (authorization: 375/2018-PR). |

Note that full information on the approval of the study protocol must also be provided in the manuscript.

# Flow Cytometry

## Plots

Confirm that:

☒ The axis labels state the marker and fluorochrome used (e.g. CD4-FITC).

☒ The axis scales are clearly visible. Include numbers along axes only for bottom left plot of group (a 'group' is an analysis of identical markers).

☒ All plots are contour plots with outliers or pseudocolor plots.

☒ A numerical value for number of cells or percentage (with statistics) is provided.

## Methodology

| Sample preparation | the detailed step-by-step procedure for preparation of a single-cell suspension from mouse brain regions is described in the method section. |

| Instrument | BD FACS Aria III |

| Software | BD FACSDiva software (BD Biosciences) and FlowingSoftware (TreeStar) |

| Cell population abundance | Purity checked of tdtomato sorted cells fraction was 88-98%. |

| Gating strategy | Cells were gated on forward/side scatter, live/dead by DAPI exclusion, and tdTomato (BP 585/42), using tdTomato and DAPI controls to set gates for each experiment. See Extended Figure 8 and 9. |

☒ Tick this box to confirm that a figure exemplifying the gating strategy is provided in the Supplementary Information.

