## [Peer Review File · Nature]

Manuscript Title: Specialized astrocytes mediate glutamatergic gliotransmission in the CNS

Reviewer Comments & Author Rebuttals

Reviewer Reports on the Initial Version:

Referees' comments:

Referee #1 (Remarks to the Author):

De Ceglia and colleagues aim to demonstrate the existence of astrocytes and define their transcriptomic type that mediate glutamatergic gliotransmission. This is a question that is rather controversial in the field and this paper applies a variety of methods to try and address it. Overall, I found the paper interesting, and I was impressed by the variety of methods they used to study glutamate release including in vivo and behavior studies as well as to demonstrate the existence of this cell types across brain areas and species. I have two questions concerning experiments and the analysis.

- 1) It was not clear to me if with the in situ functional imaging studies, they established that the specific astrocytes in the DGLM population that release glutamate are the ones in cluster 7 as defined with their transcriptomic analysis. Demonstrating this for the same cells is critical since this is a major claim of the paper. This result would directly link the physiology (glutamate release) to the specific transcriptomic type of astrocytes. The authors may have done this experiment and I have missed it. One would want to perform guided Patch-seq on cells that show glutamate release (i.e., Figure 2) vs ones that do not and show which cluster number these cells belong to. Given that the group has established Patch-seq in their lab I think this experiment should be performed if not already done (parenthetically they should reference the two original papers that developed Patch-seq Cadwell et al., and 2015 Fuzik et al., in Nature Biotechnology).
- 2) The quantification of the overlap between the different transcriptomic data sets was not clear to me (Extended Data Fig 1b). Here one would want to cross validate across data sets e.g., define clusters using some of the data sets and compare the likelihood that the test and training sets can be explained from the same clusters. There are many other statistical methods to do this analysis, but it was not clear what exactly the authors did here. The Extended Data Fig 1b show an aggregate UMAP of all the data sets and the colors coding the different data sets show clustering (i.e., colors are not salt and paper. This indicates that these data sets do not fully agree with each other.

Referee #2 (Remarks to the Author):

This manuscript by De Ceglia et al. describes a deep interrogation of the ability of astrocytes to release glutamate, modify synaptic efficacy and behavior. Using primarily manipulations in mice, the authors explore this controversial hypothesis using a combination of computational analyses of existing and new transcriptional information, physiological studies in acute brain slices and behavioral assessments of mice in which the vesicular glutamate transporter VGLUT1 was selectively deleted from some astrocytes. Through a combined analysis of transcriptional data, the authors identify one cluster (“7”) of astrocytes that contained RNAs encoding a combination of proteins that would support vesicular release of glutamate. They provide evidence that some astrocytes in human RNAseq data exhibit similar features. New functional evidence for the release of glutamate is provided by focal increases in glutamate sensor fluorescence following chemogenetic activation of astrocytes in brain slices. The authors then use viral and transgenic approaches to disrupt vesicular glutamate transporter (VGLUT1) expression in astrocytes and perform slice recordings, showing that LTP in the dentate gyrus was somewhat inhibited. Subsequent behavioral tests revealed that fear conditioning was reduced (expression of contextual memory), and counterintuitively, that seizure magnitude was enhanced in these mice. To explore whether this phenomenon occurs beyond the dentate gyrus, the authors then turn their attention to the substantial nigra dopamine pathway. They report evidence of similar “glutamatergic astrocytes” in RNAseq data from this region and, using the conditional mice, and find that SNpc DA neurons have higher mEPSC frequency, larger amplitudes and a lower PPR – an effect that appears to be mediated by loss of inhibition of presynaptic mGluRs. Finally, they provide evidence that there is greater amphetamine induced dopamine release in these mice, a phenomenon that they suggest is due to less presynaptic inhibition of SNpc DA neurons.

There is accumulating evidence that astrocytes are genotypically and phenotypically heterogeneous, but the consequences of these differences remain to be determined. This study focuses on addressing whether a subset of astrocytes release glutamate through vesicular fusion to impact glutamatergic synaptic transmission and behavior. The studies extend anatomical and physiological studies performed by this lab previously and are remarkable in their breadth. The authors should be commended for tackling this question with this series of advanced experimental manipulations. They provide new information about the diversity of responses and additional evidence that astrocytes can modulate synaptic function through this mechanism. The studies suffer a bit from this excessive breadth, particularly in the inclusion of the nigrostriatal data, where there is much less direct physiological evidence for glutamate release by astrocytes. Moreover, there are logical leaps made without sufficient evidence that make links between the datasets tenuous or not fully supported. Despite the extensive nature and logical progression of the investigations, the lack of *in vivo* physiological validation of vesicular glutamate release by astrocytes reduces the impact of the studies to the field.

Major comments

Transcriptional analyses

1. It is increasing possible to cluster cells according to transcriptional differences using various PCA algorithms. The validity of this approach depends critically on demonstrating that such profiles exist *in vivo*. Evidence for “glutamatergic astrocytes” is provided by patch-Seq experiments, but the studies lack clear evidence of the incidence and distribution of these cells in the brain using *in situ* or fluorescent reporter studies. Obviously, signals for synaptic proteins are likely to be dominated by

neurons, so a novel approach, such as cell specific tagging would be required. It would be helpful to show that sampling of astrocytes in the CA regions of the hippocampus does not yield similar transcripts. The authors should provide additional information about the age and brain regions used in these prior studies.

2. It is common to detect “anomalous” transcripts in cells. The reasons for this phenomenon are actively being debated, but one possibility is that these represent engulfment or fusion of RNA containing exosomes. Astrocytes have been shown to engulf synapses/synaptic material and this might be more likely to occur in the dentate gyrus where new neurons are continually integrated and their inputs refined. None of the data presented are inconsistent with this alternative hypothesis.

3. The cell clustering presents a complex picture with several inconsistencies and limited validation. One consistent observation is there appears to be only a small subset of astrocytes with this transcriptional profile – between 10 and 90 cells (average 41 cells/database, <5% of total population). The limited sampling of this putative population raises concerns about the validity of the conclusions. In particular, Fig 1i and J show that expression of GLT-1 was also highly variable among these astrocytes, and that VGLUT1 was expressed outside Cluster 7. Moreover, glutamine synthetase transcripts were equally abundant in this population, which would be expected to dramatically reduce cytoplasmic glutamate availability. One might also expect an inverse relationship between VGLUT and GLT-1 expression, otherwise these cells would presumably be rapidly taking up the glutamate they release, but this was not observed. Thus, what emerges is evidence of some transcripts consistent with glutamate secretion in astrocyte samples, but high variability in transcript detection between astrocytes, and lack of supporting evidence that these cells have a consistent phenotype optimized for glutamate release, raise doubt on the hypothesis proposed. Even allowing for this possibility, it appears that this phenotype is expressed in a small minority of cells, which raises questions about how effective they would be in modulating circuits.

Physiological evidence for vesicular glutamate release

4. The authors rely on co-expression of a Gq DREADD and an extracellular glutamate sensor in astrocytes to assess vesicular glutamate release. These acute slice experiments were well designed and in most cases appropriate controls performed. However, the authors condense the data by reporting responding cells and binarized fluorescent responses. For example, the binarized images in Fig. 7 make it hard to assess reliability, which is a crucial aspect of the data to indicate the presence of hot spots, which would be expected for sites of vesicular release. Maps of fluorescence changes over time with amplitudes should be provided. It would also be helpful to show movies of these responses.

5. A major limitation of this investigation is the reliance on exogenous receptors for induction. Although it is reasonable to use a Gq DREADD as a first pass to determine whether such phenomena are possible as astrocytes respond to Gq metabotropic receptors with liberation of intracellular calcium, they do not provide sufficient evidence that it occurs under physiological conditions. It is critical that they provide evidence that endogenous neurotransmitter/neuromodulator release is able to trigger glutamate release from these cells in situ. There are in vivo imaging methods that can be employed to examine astrocyte activity in the dentate gyrus, and if this phenomenon is widespread, as implicated by the SNpc data, it may also be possible to examine glutamate release by astrocytes in the upper layers of the cortex through a cranial window. Although this might seem like an unreasonable request, the authors have already provided evidence of astrocyte glutamate

release in acute preparations, while others have reported contradictory findings. The field will only advance substantially by assessing this possibility under the most physiological conditions.

6. The results of conditional VGLUT1 deletion are confusing, in that biochemical results indicate that genetic deletion was achieved, yet only partial phenotypes are observed. No fluorescent recordings are provided to demonstrate the range of glutamate sensor responses in these mice (similar to those shown in Fig. 2d). In addition, there appear to be no similar data for the GFAP-CreER conditional knockout mice, the veracity of which is critical for support of the behavioral studies.

7. The authors provide a diagram to indicate how field recordings were performed in regions with and without manipulated astrocytes, but from Fig. 3d it is not clear that expression was as restricted as drawn, which would be necessary for the design of these experiments.

8. The field recording traces in Fig. 3e (response 2) appear to be contaminated with a spike wave (visible from the sharp peak of the average response). If so, it would be inappropriate to use peak amplitude as a measure. Instead, an analysis of initial fEPSP slope should be performed, as this provides a more accurate estimate of synaptic conductance. This analysis may be difficult, given the large stimulation artifacts in these recordings.

9. Some of the findings appear inconsistent with the guiding hypothesis, including the fact that the kinetics of glutamate sniffer responses were similar in the VGLUT1 knockdown astrocytes. If less glutamate is released and these sensors report the time course of extracellular glutamate, one would expect the decay kinetics to be faster. The authors report that there was a 66% reduction in the frequency of these events. This metric is hard to understand, as the sensor responses look to be monotonic following exposure to CNO (Fig. 2d). In addition, the authors find that seizure incidence increased in the VGLUT1 cKO mice, when one might expect a decrease in seizures based on the excitatory effects observed in the hippocampus.

Minor comments

1. Statements about significance (e.g. “cluster 7 exhibited among the most significantly differentially enriched transcripts”) should be accompanied by information about the statistical tests performed and significance values.

2. The statement “thus confirming that this astrocytic cell population is evolutionarily conserved” seems a large overreach, based on the lack of independent validation and evidence that in human tissue, these cells are much more widely dispersed among the astrocyte population.

3. The statement that “whether a reduced Θ -LTP of the medial entorhinal input to the hippocampus resulted in defective contextual memory processing” and other similar specific statements should be revised. As these manipulations were performed at a global level (using GFAP-CreER), it is unclear where the locus of the effect resides.

Referee #3 (Remarks to the Author):

In this manuscript, authors have investigated molecular mechanisms and functional consequences of glutamate release from astrocytes.

The study includes the use of a plethora of assays, including state-of-the-art cellular and molecular techniques, electrophysiology, chemogenetics, patch-seq of astrocytes, behavior analysis, transgenic mice, etc., as well as bioinformatic analysis of scRNAseq/snRNAseq databases.

Authors first perform a bioinformatic analysis of RNAseq databases, one subset of hippocampal astrocytes with “synaptic-like” machinery for regulated glutamate secretion, including the vesicular glutamate transporter VGLUT1. Then, using a glutamate sensor under two-photon imaging identified a subgroup of hippocampal astrocytes that release glutamate in response to chemogenetic stimulation. The deletion of VGLUT1 suppressed this glutamate release. They further analyzed the electrophysiological and behavioral phenotype of VGLUT1 mice, and a reduction of the LTP, an impairment of the expression of the hippocampal-associated fear conditioning, and an enhancement of kainate-induced epileptiform activity. Finally, they tested the effects of glutamate release in the substantia nigra (SN) by deleting the other vesicular glutamate transporter VGLUT2. They found that the frequency of spontaneous EPSCs (sEPSCs) was enhanced in VGLUT2KO mice, and that the amphetamine-induced levels of dopamine (DA) were also increased in the dorsal striatum.

This is an interesting study that adds valuable information regarding a relevant topic in current neuroscience, the regulatory role of astrocytes of brain function through gliotransmission. Conceptually, the idea of glutamatergic transmission is not especially novel but has been a largely debated. This manuscript provides further and solid support for it with elegant experiments and cutting-edge techniques. The manuscript is also innovative regarding the functional aspects of the glutamatergic gliotransmission.

The manuscript includes a large number of interesting observations that provide interesting implications for several aspects and topics of the astrocyte-neuron interaction. This could be a strength of the manuscript, but such ambition also leads to several weaknesses, as detailed in the specific comments. Indeed, the broad scope of the manuscript, which touches on astrocyte functional heterogeneity, astrocytic glutamate involvement in hippocampal synaptic plasticity and epileptiform activity, VGLUT1 in the hippocampus, VGLUT2 in SN, regulation of dopamine levels in the striatum, etc., provides interesting and novel results in these aspects, but in many cases, they would benefit of a deeper and more focused analysis.

The study is methodologically and technically adequate, having combined the use of multiple state-of-the-art techniques to provide elegant results, which are, in general, very convincing. The main conclusions reached are, in general, well-supported by the experimental data. However, in some cases (see specific comments), probably due to the broad scope, the authors' interpretation of the results is very likely, but not exclusive. As detailed below, additional feasible experiments are needed to strongly support some of the conclusions.

Therefore, I have several comments about the present manuscript. While it presents many strengths, I have also found some weaknesses that need to be addressed to support the relevant

conclusions reached and to improve the high quality of the manuscript.

As a general assessment, I found the manuscript with excessive broad scope that weaken the important message. I believe the manuscript would benefit by producing a more focused study centered on glutamatergic gliotransmission in the hippocampus. Results presented in the first part of the manuscript (Fig. 1) is a bioinformatic analysis of databases, which this reviewer found of smaller interest. Results presented at the end of the manuscript (Fig. 4) are related to SN and the control of DA release via VGLUT2 in SN, which seems insufficiently coherent with the rest of results that were related to the hippocampus and the role of VGLUT1.

Specific comments

1. In the bioinformatic analysis, Syt1 and SNAP25 were found in cluster 7, which authors consider to be astrocytic. However, Syt 1 is considered to be responsible for neuronal exocytosis in fast synaptic transmission, whereas syt4 has been proposed to mediate astrocyte glutamate release (see Zhang et al., PNAS 2004). Likewise, SNAP25 is considered to be expressed presynaptically, whereas its isoform SNAP23 is supposed to be astrocytic (see e.g, Verkhratsky et al., EMBO J 2016). I might be missing something, but having identified “neuronal genes” in “astrocytes” is worrisome and needs clarification. Either the well-accepted genes expressed by neurons is wrong or cells in cluster 7 are not astrocytes.

2. In the experiments testing Glu release by chemogenetic activation of astrocytes, the released Glu was sensed in the same astrocyte that was stimulated. The diffusion of the Glu and activation receptors of neighboring cells is an important aspect of gliotransmission. Can Glu be detected in adjacent cells that are not stimulated? Authors could inject lower levels of both viruses to attain sparse expression to test this.

3. As an example of my general assessment about the broad scope of the manuscript, that probably has prevented a deeper experimental support for important conclusions that remain hypothetical: In Page 7, line 175, authors state that CNO-evoked SF-iGluSnFR responses occurred consistently at specific locations representing “hot-spots”. This is a very important conclusion because it is indicative of specific release sites. However, it requires a more exhaustive analysis. Are these “hot-spots” functional or simply represent heterogeneous hot-spot expression of DREADDs or SF-iGluSnFr? Authors should provide a spatial quantification of the expression of these elements. Moreover, are these hot-spots reliably conserved upon successive stimulations? This is an important aspect of the study that needs a more precise analysis.

4. In the synaptic plasticity experiments (Fig 3e), LTP amplitude was indicated to be significantly lower in VGLUT1 KO. I wonder whether this an apparent observation or statistical significance. If there is statistical significance, it should be stated, and the statistical analysis needs to be reported.

5. Related to the previous point, when whole-cell recordings were performed, only 50% of the cells showed LTP reduction. This is claimed to be consistent with the patch-seq data of 50% of astrocytes expressing VGLUT1. This might be the case, but not necessarily. Furthermore, the 50-50 situation is not helpful. To demonstrate the actual cause-effect link additional experimental work should be

provided. Otherwise, authors should refrain to establish such a weak association. The link, if it exists, between these two set of separate observations remains hypothetical.

6. VGLUT1 dependence of glutamate release implies the vesicular release probably through a calcium dependent process. To accept the conclusions regarding the synaptic studies of VGLUT1 KO require basic control experiments that the upstream phenomena, like calcium excitability, has not been compromised in these transgenic mice. Authors should provide experimental evidence testing whether both the spontaneous and theta stimulation-evoked astrocyte calcium signal were affected in these mice.

7. Experiments regarding the effects of VGLUT1KO on kainate-induced epileptiform activity (Fig.3 j-n) are interesting. However, for experts on epilepsy, they could be seen as rather simple to naïvely conclude “these data reveal a “protective” role of astrocyte VGLUT1-dependent signalling against seizures *in vivo*”. To support such a strong conclusion authors should provide evidence using different experimental models of epilepsy, as well as a more detailed analysis of ictal discharges, interictal duration, etc.

8. The last part of the manuscript focuses on VGLUT2 in SN because it is claimed to be the main VGLUT in this area. Authors basically repeat what was done in the previous figures, but obviously the rigor was not as demanding as the other part of the ms (see the following comments). This change of the focus to a different brain area and a different VGLUT weakens the main message of the manuscript because it seems to be non-coherently articulated with other parts. Authors may consider to strength the main conclusion of the paper by showing the absence of effects of VGLUT1 KO in this brain area.

9. Authors claim that the lower paired pulse ratio (PPR) in VGLUT2KO than in control mice (Fig. 4f) is “consistent with a pre-synaptic inhibition of the STN excitatory input induced by the astrocyte VGLUT2-dependent pathway”. This might be the case, but it is not necessary. If EPSCs are larger, the probability of release during PPR is expected to be decreased, i.e., no such presynaptic inhibition may exist.

10. Because “the group III mGluR agonist O-phospho-L-serine (L-SOP) did not modify sEPSC frequency in control mice”. Authors conclude that a presynaptic inhibitory mechanism endogenously active that occludes L-SOP effects. Again, this is a likely explanation, but it should be supported by direct experimental evidence, e.g., testing the effects of mGluR antagonists.

11. Authors finally aimed to test the astrocyte control on nigrostriatal circuit function by examining the *in vivo* levels of dopamine (DA) in the dorsal striatum. Authors found that basal levels in control and VGLUT2KO mice were not significantly different, which contradicts the main claim of the authors, i.e., that astrocytic VGLUT2-dependent glutamate release affects nigrostriatal circuit function. Authors should make sense of this contradiction. On the other hand, and in contrast, they found that amphetamine-evoked increase of DA levels was enhanced in VGLUT2KO. This a very interesting observation, but it lacks sufficient experimental test to provide a clear mechanistic interpretation. As honestly indicated by the authors, this is “consistent with loss in these mice of presynaptic inhibition of SNpc DA neurons (and possibly of additional astrocyte controls in the dST)”,

but this interpretation remains hypothetical and requires further studies to characterize the underlying mechanisms.

Minor comments

1. Page 9, line 209. It is stated that the essay was done “few weeks later”. This vague information should be explicitly stated in more concrete terms.

Author Rebuttals to Initial Comments:

Point-by-point response to the Referees' comments

Referees' comments:

Referee #1 (Remarks to the Author):

De Ceglia and colleagues aim to demonstrate the existence of astrocytes and define their transcriptomic type that mediate glutamatergic gliotransmission. This is a question that is rather controversial in the field and this paper applies a variety of methods to try and address it. Overall, I found the paper interesting, and I was impressed by the variety of methods they used to study glutamate release including in vivo and behavior studies as well as to demonstrate the existence of this cell types across brain areas and species.

We thank the Referee for the interest and positive global appreciation of our study and, in particular, for recognizing the variety of complementary approaches that we used to address a long-term controversial topic in the field. To comply with her/his requests, as well with the requests of the other Referees, we further enlarged the experimental portfolio with four main new datasets and several new control experiments and data analyses, to provide multiple converging additional levels of evidence supporting the existence and relevance of the sub-population of glutamatergic astrocytes. The key new added datasets and their results are summarized in our response to the general comment by Referee 2 and presented in detail in the responses to the specific *ad hoc* points of each Referee.

I have two questions concerning experiments and the analysis.

1) It was not clear to me if with the in situ functional imaging studies, they established that the specific astrocytes in the DGLM population that release glutamate are the ones in cluster 7 as defined with their transcriptomic analysis. Demonstrating this for the same cells is critical since this is a major claim of the paper. This result would directly link the physiology (glutamate release) to the specific transcriptomic type of astrocytes. The authors may have done this experiment and I have missed it. One would want to perform guided Patch-seq on cells that show glutamate release (i.e., Figure 2) vs ones that do not and show which cluster number these cells belong to. Given that the group has established Patch-seq in their lab I think this experiment should be performed if not already done (parenthetically they should reference the two original papers that developed Patch-seq Cadwell et al., and 2015 Fuzik et al., in Nature Biotechnology).

We thank the Referee for her/his suggestion. In the original submission, we did not perform guided patch-seq on individual astrocytes already tested for glutamate release. However, we fully agree that this is a critical experiment that can directly link physiology to transcriptomic type of the astrocytes by verifying if glutamate-releasing cells (CNO responders) correspond to transcriptomically-predicted glutamatergic astrocytes (cluster 7), as well as if non-responders fit to other transcriptomically-predicted astrocyte clusters. Therefore, we performed this experiment: the results are shown in Extended Data Fig. 8, and reported in text at p. 10-11, lines 318-351 (and in methods p. 58-59, lines 1460-1483). Extended Data Figure 8 includes our historical patch-seq experiment with 65 cells and our newly generated guided patch-seq on individual astrocytes that have been tested for glutamate release (20 cells). We would like to emphasize the challenge that this experiment represented. The final 20 astrocytes good for analysis that we obtained are from 37 cells on which we could perform the full experiment, which, in turn, are from many more that we tested but had to drop at some point of

the procedure for technical problems. Each single astrocyte underwent the established protocol of 6 CNO/L-glutamate puffs to ascertain its capacity to release glutamate, followed by whole-cell patching for nucleus aspiration and sequencing. Additionally, transcriptomics was performed on cells that virally expressed Gq-DREADD and SF-iGluSnFR, underwent CNO and L-glutamate stimulation, and were bathed for 10-15 min in a cocktail of synaptic inhibitors. These conditions resulted in a lower efficiency of cells passing quality control: 54% (20/37) for functional patch-seq compared to 72% (65/91) with historical patch-seq. Nonetheless, among the 20 combined patch-seq cells, we identified four responders (i.e., displaying SF-iGluSnFR response to CNO) and 16 non-responders (examples are highlighted in Extended Data Fig. 8g). Transcriptomic annotation using label transfer with our reference database matched the prediction for responders (cluster 7: 3/4; cluster 2: 1/4) and non-responders (cluster 2: 9/16; cluster 4: 3/16; cluster 5: 2/16; cluster 7: 2/16 cells; Extended Data Fig. 8h), thus showing statistical significance (Fisher exact test $P = 0.0320$) for the correct prediction. We thank the Referee for her/his suggestion that strengthens our demonstration supporting a robust correlation between physiological and molecular identification of glutamatergic astrocytes. We apologize for the missing citations and added the original patch-seq studies in both main text (p. 10, line 320, Refs. 55,56) and methods (p. 57, line 1407, Refs. 55,76).

2) The quantification of the overlap between the different transcriptomic data sets was not clear to me (Extended Data Fig 1b). Here one would want to cross validate across data sets e.g., define clusters using some of the data sets and compare the likelihood that the test and training sets can be explained from the same clusters. There are many other statistical methods to do this analysis, but it was not clear what exactly the authors did here. The Extended Data Fig 1b show an aggregate UMAP of all the data sets and the colors coding the different data sets show clustering (i.e., colors are not salt and paper. This indicates that these data sets do not fully agree with each other.

We apologize for the lack of clarity in our methodology and thank the Referee for the helpful comment. Extended Data Figure 1b represents the UMAP visualization of all cells available in the eight hippocampal databases that we used for data integration. The lack of overlap is explained by the fact that each dataset has been generated using different practices in cell collection methodology (e.g., FACS-sorted vs. non-FACS-sorted, single nuclei vs. whole cells, age, etc.). This information is summarized in Extended Data Figure 1a, and more explicitly indicated in its legend (p. 30, lines 908-911), and in methods (p. 60, lines 1511-1515). Despite differences, these databases still have reasonably good overlap and coverage for hippocampal astrocytes. Their integration allows for better sensitivity in detecting populations despite the variability. We agree with the Referee that cross-validation is a good practice to ensure non-overfitting of astrocyte predictions. The analysis presented in our initial submission was already cross-validated for our cell type prediction model trained on the hippocampal database from Yao 2021, which was split into a training and a testing batch (98.4% accuracy prediction on the test dataset, Extended Data Figure 1e). Following the Referee's comment, we performed additional confirmation using a leave-one-out cross-validation by removing each individual dataset and predicting astrocytes using the others, then calculated the overall prediction efficiency. Our results show a robust and dataset-independent performance with high accuracy, sensitivity, and specificity for astrocyte prediction. The new information is presented in Extended Data Figure 1f, its legend (p. 31, lines 915-918), in main text (p. 3, lines 73-74) and in methods (p. 61, lines 1559-1563). To note, similar accuracy, sensitivity, and specificity were confirmed even after including very recently published hippocampal sc-RNAseq datasets (Endo et al., Science, 2022).

Referee #2 (Remarks to the Author):

This manuscript by De Ceglia et al. describes a deep interrogation of the ability of astrocytes to release glutamate, modify synaptic efficacy and behavior. Using primarily manipulations in mice, the authors explore this controversial hypothesis using a combination of computational analyses of existing and new transcriptional information, physiological studies in acute brain slices and behavioral assessments of mice in which the vesicular glutamate transporter VGLUT1 was selectively deleted from some astrocytes. Through a combined analysis of transcriptional data, the authors identify one cluster (“7”) of astrocytes that contained RNAs encoding a combination of proteins that would support vesicular release of glutamate. They provide evidence that some astrocytes in human RNAseq data exhibit similar features. New functional evidence for the release of glutamate is provided by focal increases in glutamate sensor fluorescence following chemogenetic activation of astrocytes in brain slices. The authors then use viral and transgenic approaches to disrupt vesicular glutamate transporter (VGLUT1) expression in astrocytes and perform slice recordings, showing that LTP in the dentate gyrus was somewhat inhibited. Subsequent behavioral tests revealed that fear conditioning was reduced (expression of contextual memory), and counterintuitively, that seizure magnitude was enhanced in these mice. To explore whether this phenomenon occurs beyond the dentate gyrus, the authors then turn their attention to the substantia nigra dopamine pathway. They report evidence of similar “glutamatergic astrocytes” in RNAseq data from this region and, using the conditional mice, and find that SNpc DA neurons have higher mEPSC frequency, larger amplitudes and a lower PPR – an effect that appears to be mediated by loss of inhibition of presynaptic mGluRs. Finally, they provide evidence that there is greater amphetamine induced dopamine release in these mice, a phenomenon that they suggest is due to less presynaptic inhibition of SNpc DA neurons.

There is accumulating evidence that astrocytes are genotypically and phenotypically heterogeneous, but the consequences of these differences remain to be determined. This study focuses on addressing whether a subset of astrocytes release glutamate through vesicular fusion to impact glutamatergic synaptic transmission and behavior. The studies extend anatomical and physiological studies performed by this lab previously and are remarkable in their breadth. The authors should be commended for tackling this question with this series of advanced experimental manipulations. They provide new information about the diversity of responses and additional evidence that astrocytes can modulate synaptic function through this mechanism.

The studies suffer a bit from this excessive breadth, particularly in the inclusion of the nigrostriatal data, where there is much less direct physiological evidence for glutamate release by astrocytes. Moreover, there are logical leaps made without sufficient evidence that make links between the datasets tenuous or not fully supported. Despite the extensive nature and logical progression of the investigations, the lack of in vivo physiological validation of vesicular glutamate release by astrocytes reduces the impact of the studies to the field.

We thank the Referee for her/his thorough evaluation of our study, identifying both strengths and remaining weaknesses to the demonstration of glutamate exocytosis from astrocytes and its physiological relevance, a topic debated in the field since 15 years. It is rewarding for us that the Referee recognizes that we here provide new evidence to the topic, using advanced experimental manipulations, and that our study has a remarkable breadth. We acknowledge the less detailed analysis in the nigro-striatal circuit, but we thought relevant presenting information also outside the hippocampal circuitry as an indication that glutamatergic astrocytes are not restricted to the

hippocampal region. On the other hand, the Referee has made several critical comments that identify specific aspects for which she/he indicates that we provided insufficient evidence to support our conclusions, highlighting in particular the lack of *in vivo* physiological validation of vesicular glutamate release by astrocytes. We have addressed all the Referee's critical comments with new experiments using additional advanced methods and analyses, highly motivated to provide a level of demonstration adequate to hopefully settle the debated issues and advance the field once for good, as indicated in the Referee's request. Specifically, together with several new control experiments and data analyses, we have enlarged the experimental portfolio of our study with four main new datasets, aimed to provide additional converging levels of evidence supporting the existence and physiological relevance of the sub-population of glutamatergic astrocytes:

1. Spatial transcriptomics using RNAscope HiPlex assay to visualize glutamatergic astrocytes and their distribution in hippocampus (Fig. 1g,h and Extended Data Fig. 2j). These data provide direct evidence for the existence of this astrocyte population in the adult brain. See details in the response to point 1 of this Referee.
2. Combined SF-iGluSnFR imaging and patch-seq of individual astrocytes to assess whether cells functionally classified as "glutamate-releasing astrocytes" correspond molecularly to the glutamatergic astrocytes sub-population (Ext. Data Fig. 8g,h). Analysis of the collected data shows statistical significance for correct prediction of correspondence. See details in the response to point 1 by Referee 1.
3. New SF-iGluSnFR imaging experiments *in situ* to define whether astrocyte glutamate release responses are induced by stimulation of endogenous Gq-GPCRs and not just of exogenously expressed Gq-DREADD. We selected for stimulation purinergic P2Y1Rs, which mediate a physiological astrocyte control on excitatory synapses in dentate gyrus (Jourdain et al, 2007; Di Castro et al., 2011). We observed glutamate responses analogous to those previously activated with CNO. Like those, the P2Y1R-mediated responses occurred only in a sub-group of the tested DGML astrocytes and were seen consistently at specific hot spots (Fig. 2h-i, m-q; Extended Data Fig. 4b,e, f,j-l; Extended Data Fig. 6e-h; extra example in response to point 4 of this Referee). See details in the response to point 5 of this Referee.
4. New SF-iGluSnFR experiments in the awake mouse, to assess the occurrence and physiological relevance of astrocyte glutamate release *in vivo*. To our knowledge, these experiments are among the first to address this topic in the living animal. They were conducted in hippocampal dentate gyrus using fiber photometry (Extended Data Fig. 7a,b) and in primary visual cortex with two-photon imaging (Fig. 2r-y and Ext Data Fig. 7d-g). The latter experiments confirm the existence of sub-groups of astrocytes that respond to chemogenetic (Gq-DREADD) and natural neuromodulator (acetylcholine, Ach) stimulation with hotspot local glutamate release in the presence of synaptic release blockers. The cholinergic pathway is known to exert a physiological glutamatergic control on the visual cortex circuit via the astrocytes (Chen et al., PNAS, 2012). See details in the response to point 5 of this Referee.

We hope that these new results complementing those in the original manuscript, together with the additional controls, new analyses, and more detailed representations in the figures, offer a convincing demonstration of the existence of a specialized population of glutamate-releasing astrocytes of physiological relevance.

Major comments

- Transcriptional analyses

1. It is increasing possible to cluster cells according to transcriptional differences using various PCA algorithms. The validity of this approach depends critically on demonstrating that such profiles exist *in vivo*. Evidence for “glutamatergic astrocytes” is provided by patch-Seq experiments, but the studies lack clear evidence of the incidence and distribution of these cells in the brain using *in situ* or fluorescent reporter studies. Obviously, signals for synaptic proteins are likely to be dominated by neurons, so a novel approach, such as cell specific tagging would be required. It would be helpful to show that sampling of astrocytes in the CA regions of the hippocampus does not yield similar transcripts. The authors should provide additional information about the age and brain regions used in these prior studies.

We thank the Referee for her/his insightful suggestion that we interpreted as focusing on the spatial distribution of glutamatergic astrocytes within the hippocampus. We agree with the Referee that identifying this population *in situ* is, at the same time, highly needed and challenging, due to the dominance of neuronal signals in the synaptic signature that distinguishes glutamatergic astrocytes from other astrocytes. To address this challenge, we performed spatial transcriptomics using RNAscope HiPlex assay in tamoxifen-treated adult *GFAP^{creERT2}tdTom^{Isl/Isl}* mice that express tdTomato (Tom) under GFAP promoter. In several of the hippocampal regions that we studied, from CA stratum radiatum and oriens, to DG molecular layer and hilus, isolated cells can be identified by DAPI-labelled nuclei, thus excluding nuclear RNA-signals overlap from different cells (i.e. neurons/astrocytes). Therefore, we could select isolated astrocytes by DAPI and concomitant expression of astrocyte markers (GS and S100 β) and tdTomato (Tom) reporter. Through the use of HiPlex, we were able to identify several such astrocytes expressing the combination of *Snap25*, *Syt1*, and *Sc17a7/VGLUT1*, *Sc17a6/VGLUT2* transcripts in all the studied hippocampal regions (Fig. 1g; Extended Data Fig. 2j). This finding provides new and direct evidence for the existence of glutamatergic astrocytes in intact tissue, complementary to the bioinformatic and patch-seq evidence, and adds new information on their spatial distribution in hippocampus. To this latter point, and with reference to the Referee’s sentence about the specificity of glutamatergic astrocytes to DG versus CA regions, we apologize for the lack of clarity. This population is present also in CA1: this was already indicated by its identification in the GSE60361 database that is part of our scRNA-seq/snRNA-seq integrated database and contains exclusively CA1 cells, and is now shown directly by our newly generated spatial transcriptomic experiment (Fig. 1g, Extended Data Fig. 2j). Another important observation of the latter experiment concerns the glutamate astrocyte population density, which we found varies along the dorsal-ventral axis, with a higher proportion in the dorsal DGML (24% of the total number of GS/S100 β -positive astrocytes) compared to the ventral region (8% of the total) (Fig. 1h). Results of the spatial transcriptomics experiments are reported in the manuscript text (p. 4-5, lines 125-144), while the detailed experimental and analysis procedure is in the Methods (p. 54-56, lines 1324-1399). Lastly, we apologize for any confusion regarding the age and brain regions of the single-cell databases previously published. This information is provided in Extended Data Fig1a, along with details about the RNA sequencing technology adopted and the dataset identification code.

2. It is common to detect “anomalous” transcripts in cells. The reasons for this phenomenon are actively being debated, but one possibility is that these represent engulfment or fusion of RNA containing exosomes. Astrocytes have been shown to engulf synapses/synaptic material and this might be more likely to occur in the dentate gyrus where new neurons are continually integrated and their inputs refined. None of the data presented are inconsistent with this alternative hypothesis.

We understand the Referee's concern regarding engulfment of synaptic material or of exosomes containing RNA as possible cause of detection of a “glutamatergic” synaptic phenotype in some DG astrocytes, particularly because these astrocytes are at proximity of excitatory synapses integrating newborn granule cells. To this point, we now provide evidence that glutamatergic astrocytes are not restricted to the neurogenic portions of the DG, but are seen throughout hippocampus (see response to previous point). Moreover, we have several pieces of evidence that rule out the artefactual possibilities indicated by the Referee. Firstly, in one of the databases that we utilized (GSE143758), we identified a significant proportion of the glutamatergic astrocyte population despite the database is from a single-nucleus experiment, which reflects only nuclear RNA and formally excludes the enrichment of cytoplasmic mRNA. Secondly, our patch-seq experiments are enriched significantly for nuclear transcripts rather than cytoplasmic ones, and provide again strong evidence for an astrocytic origin of the transcripts. Thirdly, the newly generated “functional patch-seq” data, combining glutamate imaging to patch-seq of individual astrocytes show a robust correlation between physiological and molecular identification of glutamatergic astrocytes (Extended Data Figure 8g,h). Fourthly, for engulfment of synaptic material to occur, astrocytes identified as glutamatergic should be enriched in key transcripts responsible for astrocyte phagocytosis, such as *Megf10* and *Mertk* (Chung et al., Nature, 2013; Lee et al., Nature, 2021) with respect to astrocytes belonging to other clusters, notably to other synapse-related clusters (clusters 0, 1 and 3, see Fig. 1c), which is not the case – this information is now added in the text (p. 4, lines 113-118) and in Extended Data Fig. 2i. Lastly, and importantly, we confirmed the existence of these transcripts in the nuclei of isolated astrocyte subpopulations in intact tissue using spatial transcriptomics, thus excluding the possibility that they originated from neuronal engulfment (Fig. 1g, Extended Data Fig. 2j). In this context, we want to mention a study published very recently that has attracted our attention on an additional possible source of contamination of snRNAseq data in glia, with ambient RNA of neuronal origin (Caglayan et al. , Neuron, 2022). We acknowledge that an ambient RNA contamination may lead to misinterpretation of the results of our bioinformatic analysis. Although we cannot formally exclude this possibility for our integrated database, we can fully exclude it for our observation of the transcripts in intact tissue in the spatial transcriptomics experiment. Moreover, we have experimental observations that argue against a relevance of this contamination also in snRNAseq data: (a) following the recommendation in Caglayan et al, we conducted an experiment in which we confirmed the presence of our glutamatergic population upon sorting nuclei with NeuN antibody depletion, which significantly reduces the potential neuronal contamination. (b) if astrocytes were contaminated by ambient RNA, also other populations such as vascular cells (endocytes) extracted from the brain tissue should have been contaminated. However, we did not find a “neuronal signature” contamination in this population. The specific point of ambient RNA contamination was not raised by the Referee, but we felt it was important to clarify also this aspect in the context of potential alternative explanations for the presence of “neuronal glutamatergic” transcripts in astrocytes, and data can be provided on request if needed.

3. The cell clustering presents a complex picture with several inconsistencies and limited validation. One consistent observation is there appears to be only a small subset of astrocytes with this transcriptional profile – between 10 and 90 cells (average 41 cells/database, <5% of total population). The limited sampling of this putative population raises concerns about the validity of the conclusions. In particular, Fig 1i and J show that expression of GLT-1 was also highly variable among these astrocytes, and that VGLUT1 was expressed outside Cluster 7. Moreover, glutamine synthetase transcripts were equally abundant in this population, which would be expected to dramatically reduce cytoplasmic glutamate availability. One might also expect an inverse relationship between VGLUT and GLT-1 expression,

otherwise these cells would presumably be rapidly taking up the glutamate they release, but this was not observed. Thus, what emerges is evidence of some transcripts consistent with glutamate secretion in astrocyte samples, but high variability in transcript detection between astrocytes, and lack of supporting evidence that these cells have a consistent phenotype optimized for glutamate release, raise doubt on the hypothesis proposed. Even allowing for this possibility, it appears that this phenotype is expressed in a small minority of cells, which raises questions about how effective they would be in modulating circuits.

There are several issues raised by the Referee's critical considerations on our bioinformatic analysis. To start, the Referee makes calculations based on which she/he concludes that glutamatergic astrocytes would represent <5% of the total astrocyte population. We think that use of this mathematical approach to define population distribution when dealing with single cell data analysis is not appropriate. Indeed, calculations can be considered to be realistic only if one is sure that, for each experiment that gave rise to a database, the authors sampled the total number of astrocytes existing in the tissue, i.e., they covered the total single cell variability and obtained the full representation of the single astrocytes of the given region. Alternatively, the authors should have obtained stochastically an equal representation of all the astrocytic subpopulations. However, none of the databases that we analyzed fills both pre-conditions. Important differences in the technical steps used to create the single cell preparations indicate that the astrocyte sub-populations are neither fully nor proportionally equally represented in each of the databases. These differences, which include use of different mouse models, tissue dissociation methods, regional micro-dissections, sequencing technologies and overall total cell number yield, are now explicitly mentioned in the manuscript (details in Response to point 2 of Referee 1). For example, we can easily spot some sub-populations that are overrepresented in databases using ACSA-2 immunolabeling for FACS-sorting the astrocytes (e.g., see Batiuk database in Extended Data Fig. 2b). Therefore, variability among databases due to methodological differences in the way they were generated (as indicated in Extended Data Fig. 1a), calls for caution in drawing quantitative conclusions. Despite these factors and the large "background technical noise" that they determine, we consider an important result that we could identify consistently the glutamatergic astrocyte "phenotype" in all the databases (a conclusion further supported by the cross-validation analysis requested by Referee 1, see response to point 2 of Referee 1). Indeed, consistency of a phenotype in single cell studies should not be searched via the consistent presence of individual genes, vulnerable to the "dropout" phenomenon, but should be based on the capacity to consistently detect a signature of differentially-expressed genes that characterizes the cell population (e.g., Kharchenko et al., Nat Meth., 2014). In complement, we present compelling new evidence supporting the existence of this population, including its presence in intact tissue as revealed by spatial transcriptomics (Fig. 1g). Moreover, we highlight differences in density of the population along the dorsal-ventral axis of the hippocampus (see response to point 1 of this Referee), going from 24% in the dorsal DGML to 8% in the ventral region (Fig. 1h). The existence of an unequal instead of regular distribution of glutamatergic astrocytes may help explain the observed variations in the scRNA-seq/snRNA-seq databases composing our integrated database. This said, we can broadly agree with the Referee that the glutamatergic astrocytic population here identified corresponds to a specialized subpopulation of hippocampal astrocytes, that is *on average* quantitatively limited. Nonetheless, the high or low *mean* density of a population is not necessarily correlated to its efficiency in modulating *specific* neuronal circuits. For example, a single astrocyte in the mouse hippocampus can functionally interact with several neurons, hundreds of dendrites (Halassa et al., J Neurosci. 2007) and hundred thousand synapses (Bushong et al, J Neurosci. 2002). Concretely, a study using cell-specific CB1 receptor knock-outs found that astrocytic rather than neuronal CB1 mediates a prominent behavioral phenotype of memory impairment induced by cannabis assumption (Han et al., Cell, 2012) despite that only a

minimal fraction of CB1 receptors in hippocampus are expressed in astrocytes (and most of them are in neurons). Likewise, in our opinion, the Referee expresses logical but theoretical considerations when she/he argues that the abundance of glutamine synthase (GS) and GLT1 expression in glutamatergic astrocytes are problematic, because GS would prevent these cells from releasing glutamate and GLT-1 from producing significant effects in neurons due to immediate reuptake. We would like to present existing experimental data that challenge such expectations. For example, in past work we compared by immunogold electron microscopy L-glutamate labeling in nerve terminal and neighboring astrocytic profiles in the dentate gyrus: this approach permitted to extract cytosolic concentration values for L-glutamate: the value in the astrocytic cytosol, albeit lower than the neuronal one, was still in the low millimolar range (Bergersen et al., Cerebral cortex, 2012), despite the presumed local presence of GS, and is compatible with vesicular storage and release of the transmitter. More to this point, a recent study in ciliary epithelial cells of the eye, reported cross-regulation between GS and bestrophin-2, an anion channel that can also release glutamate, providing an unexpected mechanism by which molecules for enzymatic degradation and release of glutamate can coexist and operate in the same microenvironment (Owij et al., Nature, 2022). Concerning local coexistence of glutamate release and uptake mechanisms, we showed by immunogold EM in the DG that VGLUT-labeled synaptic-like microvesicles in astrocytes are present at sites expressing GLT1/GLAST on the plasma membrane and facing extra-synaptic sites of perforant path terminals (Bezzi et al., Nat Neurosci., 2004). We also showed that the neuronal sites express pre-synaptic NMDAR subunits (the target of the astrocyte release), clustered at a distance from the astrocyte vesicles resembling that of the synaptic cleft (Jourdain et al., Nat Neurosci., 2007; Savtchouk et al., PNAS, 2019). These immunolocalization data support a form of focal vesicular glutamate release and transmission from astrocytes to neurons, controlled but not repressed by glutamate uptake (see effect of TBOA in Santello et al., Neuron, 2011), which is in line with the several astrocyte-mediated functional effects that we reported on synaptic functions and plasticity mediated by presynaptic NMDAR in the dentate gyrus (Jourdain et al., *ibid*; Di Castro et al., Nat Neurosci., 2011; Savtchouk et al., *ibid*.).

- Physiological evidence for vesicular glutamate release

4. The authors rely on co-expression of a Gq DREADD and an extracellular glutamate sensor in astrocytes to assess vesicular glutamate release. These acute slice experiments were well designed and in most cases appropriate controls performed. However, the authors condense the data by reporting responding cells and binarized fluorescent responses. For example, the binarized images in Fig. 7 make it hard to assess reliability, which is a crucial aspect of the data to indicate the presence of hot spots, which would be expected for sites of vesicular release. Maps of fluorescence changes over time with amplitudes should be provided. It would also be helpful to show movies of these responses.

We thank the Referee for acknowledging the quality of our experimental design and adequacy of most controls done. Given the controversial topic and contradictory astrocyte iGluSnFR literature, this was a necessary starting point. The Referee is totally right that we greatly condensed the results. The graphical representation of the data was challenging for space reasons and this led us to the “condensation”, including the generation of binary maps, a simplified representation allowing us to summarize the key information for all cells (a total of 47 cells in the original submission, summing up to 85 now with the new experiments addressing point 5 of this Referee). We agree, however, that the selected graphical modalities did not allow full evaluation of important aspects of the responses, including their spatial reliability (hot spots). For this reason, after getting green light by the Editor about

spaces availability, we completely reorganized the presentation of the SF-iGluSnFR data, both those already present in the original submission and those added to this revised version, so to now show the information requested by the Referee and respond to this and other points of this Referee (point 5) and of Referee 3 (point 3). Overall, all the glutamate release data *in situ* and *in vivo* are now presented in much more detail in one main figure (Fig. 2), four Extended Data figures (Figs. 4-7) and 1 supplemental video (Video 1). Additional examples not included in the manuscript are provided in this response for the Referees' evaluation. For the new *in vivo* data we send to the response to point 5 of this Referee. For the *in situ* data, we present:

1. Time series of pixel-by-pixel fluorescence intensity maps (z-score color scale) throughout the FOV in the 240 msec preceding and following each one of the 6 applications of the stimulus (CNO for formerly presented experiments, 2MeSADP for new experiments, see response to point 5). This representation allows spatial visualization and comparison of the sequential responses to stimuli and is provided in Fig. 2 (panels d,h) for one representative responder astrocyte/type of stimulus. A second set of examples is provided here below. A third one is presented for a CNO responder in Extended Data Fig. 5. The time series are complemented by standard deviation (STD) projections that display the signal variance across the 6 drug stimulations. Areas of high variance reflect subregions of the FOV that repeatedly exhibit stimulus-evoked SF-iGluSnFR responses, namely "hot spots" (Fig. 2, panels c,g). The same STD map is shown for the control L-Glut application (same panels in Fig. 2), while the corresponding L-Glut time series are shown in Extended Data Fig. 4, panels a,b).
2. Traces providing kinetics and amplitude (in z-score scale) of responses from 2 representative hot spots for the 6 applications of CNO and 2MeSADP (Fig. 2, panels e, i, and example here below).
3. A movie showing the z-scored raw fluorescence responses from another CNO responder example (Supplementary video 1).
4. Time-series and STD projections like those in point 1 above, but representing stimuli applied in VGLUT1^{GFAP-KO} (CNO) and P2Y1R^{GFAP-KO} (2MeSADP) astrocytes, reporting the abolished effect of the two stimuli in the KO cells (time-series: Fig. 2, panels l,o; STD projections: Fig. 2, panels k, m), together with the maintained L-Glut responses (Fig. 2, panels k,m; Extended Data Fig. 4, panels d,e). See additional examples in the Figure here below.
5. Representative examples of cells not responding to CNO or 2MeSADP (but responding to L-Glut) in the same conditions in which others respond, and of a cell still responding to CNO in VGLUT1^{GFAP-KO}, with visual representation organized as described in point 1 (Extended Data Fig. 4, panels g-i for CNO non-responder; panels j-l for 2MeSADP non-responder; panels m-o for CNO responder in VGLUT1^{GFAP-KO}).
6. In complement to the information in 1-5, we present mean projection images of Gq-DREADD-mCherry expression pattern in the astrocyte in the FOV, allowing for comparison with SF-iGluSnFR CNO-responding regions (see point 3 of Referee 3), and mean projection images of EBFP2-iCre or mCherry-iCre nuclear expression in the astrocyte in the FOV, indicating that the cell underwent *Cre* recombination (Gq-DREADD m-cherry: Fig. 2, panels c,k; Extended Data Fig. 4, panel h,n; iCre-reporter: Fig. 2, panels k,n; Extended Data Fig. 4, panel n; for both see also example for Referees here below)
7. A figure that explains the analytical steps that we used to generate the final binarized maps from which we define responder vs non-responder astrocytes and extract quantitative information about the hot spot-responding regions (Ext. Data Fig. 5, panels a-h).

8. Binary maps reporting the agonist response and corresponding L-Glut response for each of the 85 analyzed cells, accompanied by the mean kinetics of the responses for the 4 experimental groups (CNO/2MeSADP stimulations in WT/KO mice). (Extended Data Fig. 6 a-h)

We hope that this new representation of the *in situ* results is satisfactory and provides all the elements for a thorough evaluation of the veracity of our statements in the manuscript.

Additional examples of representative of FOVs: *top, left*: a Gq-DREADD-expressing astrocyte responding to CNO in a wild-type mouse; *top, right*: an astrocyte responding to 2MeSADP in a wild-type mouse; *bottom, left*: a Gq-DREADD-expressing astrocyte not responding to CNO in a *Slc17a7^{fl/fl}* mouse injected with *AAV5-hGFAP-EBFP2-iCre* (*VGLUT1^{GFAP-KO}*); *bottom, right*: an astrocyte not responding to 2MeSADP in a *P2ry1^{fl/fl}* mouse injected with *AAV5-hGFAP-mCherry-iCre* (*P2Y1R^{GFAP-KO}*). All cells are injected with *AAV5-hGFAP-SF-iGluSnFR(A184S)*. For detailed legends see identical panels in Fig. 2.

5. A major limitation of this investigation is the reliance on exogenous receptors for induction. Although it is reasonable to use a Gq DREADD as a first pass to determine whether such phenomena are possible as astrocytes respond to Gq metabotropic receptors with liberation of intracellular calcium, they do not provide sufficient evidence that it occurs under physiological conditions. It is critical that they provide evidence that endogenous neurotransmitter/neuromodulator release is able to trigger glutamate release from these cells *in situ*. There are *in vivo* imaging methods that can be employed to examine astrocyte activity in the dentate gyrus, and if this phenomenon is widespread, as implicated by the SNpc data, it may also be possible to examine glutamate release by astrocytes in the upper layers of the cortex through a cranial window. Although this might seem like an unreasonable request, the authors have already provided evidence of astrocyte glutamate release in acute preparations, while others have

reported contradictory findings. The field will only advance substantially by assessing this possibility under the most physiological conditions.

We fully agree with the spirit of the Referee's comments and, as already indicated, it is in the aim of the present study to advance the field over a long-lasting controversy. Therefore, we performed experiments to our knowledge unprecedented for astrocytes, notably *in vivo*, trying to push the level of demonstration of astrocyte glutamate release to the highest realistic standards (the Referee her/himself is aware of the challenge intrinsic to her/his request ("*although this might seem like an unreasonable request...*"). Our strategy has been to: (a) stimulate endogenous receptors already shown in the literature to mediate physiological controls by astrocytes, *in situ* (P2Y1R) and *in vivo* (cholinergic, presumably muscarinic); (b) use as stimulus in *in vivo* two-photon experiments a natural neuromodulator, acetylcholine (ACh); (c) perform the stimulations, including *in vivo*, in the presence of synaptic release blockers; (d) produce parallel data using Gq-DREADD astrocyte-specific stimulation, to be in the position of comparing the spatial-temporal properties of the endogenous and chemogenetic astrocyte SF-iGluSnFR responses. With the above approach we produced the following new evidence:

1. In *in situ* experiments, we stimulated P2Y1R, an endogenous Gq-GPCR that we previously demonstrated mediates a physiological control by astrocytes on PP-GC synapses in dentate gyrus. The control was revealed by the effect of P2Y1R antagonists on both basal (Di Castro et al., Nat Neurosci, 2011) and evoked release at these synapses (Jourdain et al., Nat Neurosci, 2007). For the demonstration that the control is astrocyte-specific, see point 2 here below. In these experiments we used the selective P2Y1R agonist, 2MeSADP, as stimulus, and applied the same 6 puffs protocol used for CNO stimulations, including the 6 control L-Glut puffs. This approach allowed us comparing the responses evoked by endogenous Gq-GPCR stimulation to those evoked by Gq-DREADD stimulation. Responses were very similar: only a subgroup of astrocytes responded to 2MeSADP stimulation, like for CNO, and responses were consistently seen at specific "hot spot" locations. We noticed just a few differences in number and size of the hot spots induced by the natural and chemogenetic stimulations. Results of experiments of P2Y1R stimulation in wild-type mice are presented in new text (p. 7-8, lines 232-247) and figures (Fig. 2, panels f-i,p,q; Extended Fig. 4, panels b, f, j-l, Extended Fig. 6, panels e,f; see additional example in response to point 4 of Referee 2).
2. We repeated P2Y1R stimulations in mice in which we induced P2Y1R deletion selectively in astrocytes. In this case, none of the astrocytes responded to 2MeSADP with glutamate release, demonstrating that the endogenous Gq-GPCR signaling that produces the release is indeed in astrocytes. Results of these experiments are presented in new text (p. 8, lines 247-255) and figures (Fig. 2, panels m-q; Extended Fig. 4, panels e,f; Extended Fig. 6, panels g,h; see additional example in response to point 4 of Referee 2).
3. In parallel, we developed *in vivo* approaches, as encouraged by the Referee. At first, we tried to measure SF-iGluSnFR signals in the dorsal dentate gyrus of awake mice by fiber photometry, an approach immediately available in a collaborating lab (Manuel Mameli), where it is applied to neuronal studies. We tested the use of an optofluid cannula, i.e. a cannula connected to the optic fiber, to locally infuse drugs in the FOV. To our knowledge, this approach had never been tried before for recording locally-evoked SF-iGluSnFR signals changes in astrocytes *in vivo*, although a paper reporting fiber photometry SF-iGluSnFR measures from astrocytes appeared in BioRxiv at almost the same time (<https://doi.org/10.1101/2022.05.12.491656>). The first test was in Gq-DREADD-expressing mice where we locally applied CNO in the presence of synaptic blockers. We had mixed expectations, suspecting that the SF-iGluSnFR signal change would have been tenuous if not undetectable, as it would have reported release from only a

subpopulation of the astrocytes present in the imaged FOV. However, we were able to record tiny, but statistically significant responses to CNO, which were not evoked by infusion of the CNO vehicle in the same FOV (Extended Data Fig. 7a-b). We reasoned that if we could capture minimal but visible glutamate responses to CNO with a non-ideal “population fluorescence” approach as fiber photometry, it was worth trying higher resolution two-photon studies, as proposed by the Referee. Results of these new fiber photometry experiments are presented in the text (p. 8, lines 256-265). Methodology and analysis detail are presented at p. 72-73, lines 1952-1983.

4. We performed *in vivo* two-photon astrocyte SF-iGluSnFR imaging in awake mice, an ongoing approach in our lab, albeit limited so far to Ca²⁺ imaging studies. We decided to target the visual cortex because we obtained transcriptomics analytical support that analogous glutamatergic astrocytes (cluster 7) are present in this area (Extended Data Fig. 7c) and, importantly, because previous work from Migranka Sur lab described a physiological role for astrocyte signalling in this area. Thus, astrocytes are activated by the cholinergic afferents from nucleus basalis and control the excitatory visual cortex circuit via NMDAR activation, presumably by Ca²⁺-dependent glutamate release upon muscarinic activation (Chen et al., PNAS, 2012). In the two photon experiments, we first infused synaptic blockers via the cranial window and observed strong suppression of the spontaneous SF-iGluSnFR signals in the awake mouse (Extended Data Fig. 7d-f). Subsequently, we increased the local Ach concentration via targeted puffs and observed astrocyte SF-iGluSnFR responses very similar to those evoked by 2MeSADP *in situ*, with only some of the tested astrocytes responding to the stimulus, and responses being restricted to small local astrocyte domains that were in good part conserved upon repetition of the stimulus. Finally, we tested CNO stimulations in Gq-DREADD-expressing mice and obtained results analogous to those with Ach, strengthening the astrocyte-specificity of the observed responses. All these new *in vivo* two-photon data, like the *in situ* data, are presented as fluorescence intensity, z-scored color maps, together with traces of peaks in ROIs in the main figure 2 (panels r-y), while z-scored color maps of all the tested cells are shown in Extended Data Fig. 7g. The results of these experiments are presented in the text (p. 8-10, lines 266-317), the experimental methodology at p. 66-68, lines 1736-1786, and the data analysis approach at p. 71-72, lines 1894-1950.

We believe that the several new data added to the study, showing consistency in the type of responses evoked by endogenous stimuli with respect to those previously shown for chemogenetic stimulations, as well as consistency of the responses observed *in vivo* in the awake animal with those observed *in situ*, provide strong support for the existence of astrocyte sub-populations releasing glutamate (not limited to the hippocampus) and for a broad physiological relevance of this phenomenon.

6. The results of conditional VGLUT1 deletion are confusing, in that biochemical results indicate that genetic deletion was achieved, yet only partial phenotypes are observed. No fluorescent recordings are provided to demonstrate the range of glutamate sensor responses in these mice (similar to those shown in Fig. 2d). In addition, there appear to be no similar data for the GFAP-CreER conditional knockout mice, the veracity of which is critical for support of the behavioral studies.

We regret not having highlighted in the original manuscript the presence in DGML astrocytes not only of VGLUT1 but also, to a lesser extent, of VGLUT2, which is very likely responsible of the residual “partial phenotype” in the few (3/23) VGLUT1^{GFAP-KO} astrocytes that responded to CNO despite the VGLUT1 deletion. We agree with the Referee that omission of this information has been confusing, because the data seemed to identify a contradiction between the genomic PCR demonstration of

achieved VGLUT1 deletion and the functional responses. We first reported the presence of VGLUT2 in DGML astrocytes several years ago in Bezzi et al., Nat Neurosci., 2004. We showed by both immunogold EM and single-cell PCR the presence of both VGLUT1 and VGLUT2 in these astrocytes. Quantitatively, VGLUT2-positive astrocytes (scPCR) and VGLUT2-positive astrocytic processes (immuno-EM) were less numerous than VGLUT1-positive ones (range 15-37%). The spatial transcriptomic and patch-seq data in the present study further corroborate the initial information by showing expression of VGLUT2 in cluster 7 astrocytes, albeit at lower levels and in a lower number of astrocytes than VGLUT1 (spatial transcriptomics: Fig. 1g; Extended Data Fig. 2j; patch-seq: Extended Data Fig. 8f). This information is in line with the observations in residual CNO responders in VGLUT1^{GFAP-KO} cells (see Extended Data Fig. 4, panels m-o). We have modified the text to clarify the issue, making appropriate reference to the present and past data on VGLUT2 expression in DGML astrocytes (p. 7, lines 227-228).

In light of the above information, we believe there is no real contradiction between our SF-iGluSnFR and genomic PCR data. The latter clearly show that *Cre*-dependent VGLUT1 genetic deletion was achieved both when virally-induced in *Slc17a7^{fl/fl}* mice (Extended Data Fig 4c) and when TAM-induced in *GFAP^{CreERT2}Slc17a7^{fl/fl} tdTom^{Isl/Isl}* mice (Fig. 3b). In the latter mice, *Cre*-dependent expression of tdTomato fluorescence allowed visualization of the astrocytes that underwent recombination. tdTomato expression was also exploited for FACS sorting the fluorescent astrocyte population and directly prove they had VGLUT1 genetic deletion, as well as for the design of the LTP experiments shown in Fig. 3e-f (see also response to point 7 here below). However, it was not appropriate for conducting SF-iGluSnFR experiments in this mouse line. Thus, the wide fluorescence emission profile of tdTomato, made hardly compatible its combined use with SF-iGluSnFR, prompting us to use the alternative viral strategy in *Slc17a7^{fl/fl}* mice lacking tdTomato.

In conclusion, we believe that the genomic PCR evidence, combined with the above explanations, indicates that VGLUT1 KO was achieved in astrocytes of *GFAP^{CreERT2}Slc17a7^{fl/fl} tdTom^{Isl/Isl}* mice, supporting the veracity of the phenotypes observed in these mice in synaptic physiology, behavior and in acute seizures experiments.

7. The authors provide a diagram to indicate how field recordings were performed in regions with and without manipulated astrocytes, but from Fig. 3d it is not clear that expression was as restricted as drawn, which would be necessary for the design of these experiments.

We think that this comment of the Referee depends on she/he missing the images that we presented in Extended Data Fig 8e of the original manuscript, now become Extended Data Fig. 9e in the revised manuscript. In addition to the scheme reported in Fig. 3d, we presented in that panel the exact experimental setting for the θ -LTP, including fluorescence images with Tom-positive and Tom-negative astrocytes. In particular, the positions of the stimulating electrode and of the two LFP recording electrodes and their interdistances - 200 μ m - in the DGML are evident from the lower magnification bright-field and fluorescence images, whereas the location of the recording electrodes, one in proximity of a Tomato fluorescent astrocyte (astro-Tom+) and the other of a non-fluorescent astrocyte (astro) without fluorescent astrocytes in the direct surroundings, is observed in the higher magnification images. This experimental setting is described in Methods (p. 74, lines 2008-2015).

8. The field recording traces in Fig. 3e (response 2) appear to be contaminated with a spike wave (visible from the sharp peak of the average response). If so, it would be inappropriate to use peak amplitude as a measure. Instead, an analysis of initial fEPSP slope should be performed, as this provides a more

accurate estimate of synaptic conductance. This analysis may be difficult, given the large stimulation artifacts in these recordings.

We thank the Referee for this comment. We believe that in our stimulation conditions and fEPSP recordings, contamination of fEPSPs due to population spikes is quite marginal, since also at the highest stimulation intensities used to perform the I/O curve they are rarely overt. Nonetheless, we agree with the Referee that initial fEPSP slope provides a more accurate measure of synaptic transmission than amplitude. Thus, we have performed slope fEPSPs analysis and confirmed the observation that θ -LTP magnitude of fEPSP recorded in proximity of VGLUT1^{GFAP-KO} astrocytes is significantly lower with respect to θ -LTP recorded in the domain of wildtype astrocytes. Results of the new slope analysis are presented in the revised Fig. 3e-f and in the main text (p. 11, lines 375-380).

9. Some of the findings appear inconsistent with the guiding hypothesis, including the fact that the kinetics of glutamate sniffer responses were similar in the VGLUT1 knockdown astrocytes. If less glutamate is released and these sensors report the time course of extracellular glutamate, one would expect the decay kinetics to be faster. The authors report that there was a 66% reduction in the frequency of these events. This metric is hard to understand, as the sensor responses look to be monotonic following exposure to CNO (Fig. 2d). In addition, the authors find that seizure incidence increased in the VGLUT1 cKO mice, when one might expect a decrease in seizures based on the excitatory effects observed in the hippocampus.

The Referee lists here some apparent inconsistencies of our data with the guiding hypothesis. Concerning the kinetics of glutamate sniffer responses, the Referee is surprised that astrocyte VGLUT1 deletion did not modify such kinetics, which should have faster decay because less glutamate should be released in this condition. We agree with the Referee's prediction. In addition to what discussed in response to point 6 about the nature of such residual responses, let us highlight that we reported a mean value of 350.90 ± 45.66 ms for the FWHM of the glutamate release responses in VGLUT1^{GFAP-KO} astrocytes vs 445.20 ± 57.34 ms in wild-type astrocytes, as well as a value of 266.5 ± 29.96 ms for the decay time in VGLUT1^{GFAP-KO} astrocytes vs 352.10 ± 51.21 ms in wild-type astrocytes, i.e. FWHM and decay time were, respectively, 21% and 25% lower in the KO astrocytes, in line with the Referee's prediction. These reductions did not reach, however, statistical significance because of the large unbalance in the sampled populations, i.e. in the VGLUT1^{GFAP-KO} astrocytes we had only 3 responders out of 23 sampled astrocytes, compared to 9 out of 24 in wild-type astrocytes.

Concerning our statement that there was a 66% reduction in the frequency of the SF-iGluSnFR responses in VGLUT1^{GFAP-KO} astrocytes, we realized from the Referee's comment that the statement was misleading and difficult to understand. Indeed, we used inappropriately the term "frequency" to indicate the likelihood of encountering an astrocyte capable of glutamate release. By thinking that "frequency" instead referred to the interevent interval of SF-iGluSnFR responses, the Referee was obviously surprised. We are sorry for causing the misunderstanding and have now rephrased the text (page 7, lines 213-215) to: "*In this case, of 23 tested cells, only 3 showed reliable SF-iGluSnFR responses to CNO, with a 66% reduction in the number of responding astrocytes compared to wild-type mice (Fig. 2j-l,p, Extended Data Figs. 4m-o and 6c)*".

Finally, the Referee was surprised, and found counterintuitive, that the seizures incidence increased instead of decreasing in VGLUT1^{GFAP-KO} mice, given the excitatory effect of glutamatergic astrocytes in hippocampus. This is a logical conclusion. However, the excitatory effect that we documented in hippocampus refers specifically to the medial entorhinal cortex-dentate circuit function in a physiological plasticity paradigm, whereas the acute seizures generated by peripheral kainate

administration involve activation of multiple circuits in a pathological context. Indeed, our EEG electrodes, placed in cortex and hippocampus, recorded discharges in both regions, accounting for a widespread phenomenon. Let us also say that the inhibitory effect that we report for astrocyte VGLUT2-dependent signaling on the excitatory input to nigral DA neurons via presynaptic group III mGluRs, shows that astrocyte glutamatergic signaling can also produce inhibition of excitation. Therefore, we think that one can hardly predict *a priori* the global impact of astrocyte VGLUT1 KO on the E/I balance. To make realistic predictions one should know more about the location of glutamatergic astrocytes in the several circuits involved in the epileptic seizures phenomenon, as well as about the types of influence these astrocytes exert at the different circuit levels. Consequently, we think that our EEG data, albeit surprising, are not necessarily inconsistent with the guiding hypothesis.

Minor comments

1. *Statements about significance (e.g. “cluster 7 exhibited among the most significantly differentially enriched transcripts”) should be accompanied by information about the statistical tests performed and significance values.*

We agree with the Referee’s comment and have added the requested statistics information (significance values: page 4, lines 110-112; statistical test performed: p. 79, lines 2161-2162).

2. *The statement “thus confirming that this astrocytic cell population is evolutionarily conserved” seems a large overreach, based on the lack of independent validation and evidence that in human tissue, these cells are much more widely dispersed among the astrocyte population.*

We agree with the Referee’s comment and have toned down our text accordingly. Now it reads (page 4, lines 119-124): *“To evaluate whether this cluster is preserved among species, we interrogated three recent databases of human hippocampal cells³⁵⁻³⁷. We annotated them using our integrated astrocytic database as reference to perform label transfer, and confirmed that human hippocampal cells had transcriptional features typical of astrocytes and molecularly similar to several of the clusters previously identified, including the “synaptic glutamate exocytosis” cluster 7 (Fig. 1f)”.*

3. *The statement that “whether a reduced Θ -LTP of the medial entorhinal input to the hippocampus resulted in defective contextual memory processing” and other similar specific statements should be revised. As these manipulations were performed at a global level (using GFAP-CreER), it is unclear where the locus of the effect resides.*

Although contextual memory encoding is known to involve specific circuits, including the medial entorhinal cortex input for spatial representation, we agree with the Referee that the model that we used leads to GFAP-dependent VGLUT1 gene manipulation throughout the CNS and does not allow to firmly establish a causative link between reduced Θ -LTP at the medial entorhinal input and reduced context recall performance. Consequently, we have rephrased the sentence (page 12, lines 383-384): *“Next, we asked whether the induction of VGLUT1^{GFAP-KO} astrocytes could also affect hippocampal memory processing⁵⁹”.*

Referee #3 (Remarks to the Author):

In this manuscript, authors have investigated molecular mechanisms and functional consequences of glutamate release from astrocytes.

The study includes the use of a plethora of assays, including state-of-the-art cellular and molecular techniques, electrophysiology, chemogenetics, patch-seq of astrocytes, behavior analysis, transgenic mice, etc., as well as bioinformatic analysis of scRNAseq/snRNAseq databases. Authors first perform a bioinformatic analysis of RNAseq databases, one subset of hippocampal astrocytes with “synaptic-like” machinery for regulated glutamate secretion, including the vesicular glutamate transporter VGLUT1. Then, using a glutamate sensor under two-photon imaging identified a subgroup of hippocampal astrocytes that release glutamate in response to chemogenetic stimulation. The deletion of VGLUT1 suppressed this glutamate release. They further analyzed the electrophysiological and behavioral phenotype of VGLUT1 mice, and a reduction of the LTP, an impairment of the expression of the hippocampal-associated fear conditioning, and an enhancement of kainate-induced epileptiform activity. Finally, they tested the effects of glutamate release in the substantia nigra (SN) by deleting the other vesicular glutamate transporter VGLUT2. They found that that the frequency of spontaneous EPSCs (sEPSCs) was enhanced in VGLUT2KO mice, and that the amphetamine-induced levels of dopamine (DA) were also increased in the dorsal striatum.

This is an interesting study that adds valuable information regarding a relevant topic in current neuroscience, the regulatory role of astrocytes of brain function through gliotransmission. Conceptually, the idea of glutamatergic transmission is not especially novel but has been a largely debated. This manuscript provides further and solid support for it with elegant experiments and cutting-edge techniques. The manuscript is also innovative regarding the functional aspects of the glutamatergic gliotransmission.

The manuscript includes a large number of interesting observations that provide interesting implications for several aspects and topics of the astrocyte-neuron interaction. This could be a strength of the manuscript, but such ambition also leads to several weaknesses, as detailed in the specific comments. Indeed, the broad scope of the manuscript, which touches on astrocyte functional heterogeneity, astrocytic glutamate involvement in hippocampal synaptic plasticity and epileptiform activity, VGLUT1 in the hippocampus, VGLUT2 in SN, regulation of dopamine levels in the striatum, etc., provides interesting and novel results in these aspects, but in many cases, they would benefit of a deeper and more focused analysis.

The study is methodologically and technically adequate, having combined the use of multiple state-of-the-art techniques to provide elegant results, which are, in general, very convincing. The main conclusions reached are, in general, well-supported by the experimental data. However, in some cases (see specific comments), probably due to the broad scope, the authors' interpretation of the results is very likely, but not exclusive. As detailed below, additional feasible experiments are needed to strongly support some of the conclusions.

Therefore, I have several comments about the present manuscript. While it presents many strengths, I have also found some weaknesses that need to be addressed to support the relevant conclusions reached and to improve the high quality of the manuscript.

As a general assessment, I found the manuscript with excessive broad scope that weaken the important message. I believe the manuscript would benefit by producing a more focused study centered on glutamatergic gliotransmission in the hippocampus. Results presented in the first part of the manuscript (Fig. 1) is a bioinformatic analysis of databases, which this reviewer found of smaller interest. Results presented at the end of the manuscript (Fig. 4) are related to SN and the control of DA release via VGLUT2 in SN, which seems insufficiently coherent with the rest of results that were related to the hippocampus and the role of VGLUT1.

We thank the Referee for the thorough assessment of our study and for highlighting several merits in it, from bringing new information to a relevant and debated topic in current neuroscience, to providing several new interesting results that not only further support glutamatergic gliotransmission but have implications for additional aspects of astrocyte-neuron interactions, and for doing so with elegant experiments and cutting edge techniques. We also acknowledge the positive global assessment of our work by the Referee who considers our methods adequate, the results very convincing and the main conclusions well-supported by the experimental data, in general. The Referee also indicates, however, that our study has perhaps excessively broad scope, particularly in its final part on the nigro-striatal circuit, that she/he finds not fully integrated with the core observations in hippocampus - as remarked also by Referee 2. The Referee identifies in the broad scope also the cause of some weaknesses that need to be addressed to support the relevant conclusions reached and to improve the high quality of the manuscript. We addressed all the points raised by the Referee, including most of them with new experiments and deeper analyses, including those concerning the nigro-striatal circuit. These new experiments (together with the many others requested by Referees 1 and 2) hopefully resolve the identified weaknesses and strengthen all our conclusions.

*1. In the bioinformatic analysis, *Syt1* and *SNAP25* were found in cluster 7, which authors consider to be astrocytic. However, *Syt 1* is considered to be responsible for neuronal exocytosis in fast synaptic transmission, whereas *syt4* has been proposed to mediate astrocyte glutamate release (see Zhang et al., PNAS 2004). Likewise, *SNAP25* is considered to be expressed presynaptically, whereas its isoform *SNAP23* is supposed to be astrocytic (see e.g, Verkhartsy et al., EMBO J 2016). I might be missing something, but having identified “neuronal genes” in “astrocytes” is worrisome and needs clarification. Either the well-accepted genes expressed by neurons is wrong or cells in cluster 7 are not astrocytes.*

We understand the Referee’s comment. However, we would like to highlight that the bioinformatic analysis that we performed to identify cluster 7 relies on the correlated expression of multiple genes (a signature) rather than just on expression of “key” individual genes such as *Syt1* and *Snap25*. Indeed, in single cell studies, individual genes are vulnerable to the “dropout” phenomenon and may not be present in individual cells, while these cells still belong to a given population. Therefore, identification of a cell population relies on the capacity to consistently detect a signature of differentially-expressed genes that characterizes the cell population (e.g., Kharchenko et al., Nat Meth., 2014). In this respect, we could consistently identify the glutamatergic astrocyte signature in all the individual databases that we integrated, despite the large variability between them and the large “background technical noise” this variability determines. Secondly, while the classification of *Snap25* or *Syt1* as neuronal genes is based on several studies showing their clear enrichment in neuronal populations, this evidence is not enough to rule out the presence of *Snap25* and *Syt1* in astrocytic subpopulations. These proteins are functionally defined as essential players in exocytosis of synaptic vesicles rather than cell-type specific markers. Thus, it is not unexpected that they are detected in a “peculiar” subpopulation of astrocytes

exhibiting glutamatergic regulated secretion. As for *Snap23* and *Syt4*, according to our integrated hippocampus database, they are not as enriched in cluster 7 as are *Syt1* and *Snap25*, rather they are quite spread out in astrocyte clusters (Extended Data Fig. 2f,g). To confirm expression of *Snap25*, *Syt1* and *Scl17a7/VGLUT1*, *Scl17a6/VGLUT2* transcripts by a subpopulation of astrocytes, we performed spatial transcriptomics using RNAscope HiPlex assay. The glutamate exocytosis markers were conspicuously more abundant in a specific subset of astrocytes that were present in all regions of the hippocampus (Fig. 1g , Extended Data Fig. 2j). We refer to our responses to points 1 and 2 of Referee 1 and to point 3 of Referee 2 for additional information relevant to this point of Referee 3.

2. In the experiments testing Glu release by chemogenetic activation of astrocytes, the released Glu was sensed in the same astrocyte that was stimulated. The diffusion of the Glu and activation receptors of neighboring cells is an important aspect of gliotransmission. Can Glu be detected in adjacent cells that are not stimulated? Authors could inject lower levels of both viruses to attain sparse expression to test this.

We agree with the Referee that astrocyte-to-astrocyte glutamate signal propagation is an important aspect of gliotransmission, and might be one of the mechanisms sustaining e.g., heterosynaptic modulation by astrocytes via spatial transfer of information between non directly connected synapses. We also acknowledge that the highly “controlled methodology” that we set-up to measure astrocyte glutamate release can, *in principle*, be further developed to assess the question raised by the Referee. We want to point out, however, that this experiment is not technically straightforward and has strict requirements to be unequivocally interpreted. If we consider Gq-DREADD stimulation by CNO puff, the first necessary pre-condition is that sparse “goldilocks” viral expression is achieved, so to have microenvironments in which an astrocyte expressing Gq-DREADD-mCherry + SF-iGluSnFR (astrocyte 1) sits contiguously to another astrocyte that only expresses SF-iGluSnFR (astrocyte 2). To note that astrocyte 2 needs to not express even a small amount of Gq-DREADD-mCherry, which would be almost invisible in terms of fluorescence but still possibly capable of inducing direct glutamate release from astrocyte 2. In our experience, chances to obtain this exact configuration are dim. This pre-condition can be, however, eliminated, by using natural Gq-GPCR stimulations. Thus, during manuscript revision, we found that endogenous P2Y1R stimulation with the selective agonist 2MeSADP produces release responses similar to chemogenetic stimulations (Fig. 2). Therefore, experiments with P2Y1R stimulations need just SF-iGluSnFR expression in two contiguous astrocytes. Nonetheless, other requirements are to be met: 2MeSADP puff must be spatially restricted to astrocyte 1, excluding any leak to astrocyte 2, which could cause its direct activation. In complement, the exact border between the membranes of the two astrocytes - both expressing SF-iGluSnFR – must be recognizable. Otherwise, it will be difficult to establish if responses in cell 2 are secondary to activation of cell 1 or depend on direct activation of cell 2. Overall, we think that this experiment, albeit undoubtedly interesting, is demanding and, in our opinion, beyond the scope of the present study, when considering on balance technical challenges, chances of success and extra information brought to the main message of the story.

3. As an example of my general assessment about the broad scope of the manuscript, that probably has prevented a deeper experimental support for important conclusions that remain hypothetical: In Page 7, line 175, authors state that CNO-evoked SF-iGluSnFR responses occurred consistently at specific locations representing “hot-spots”. This is a very important conclusion because it is indicative of specific release sites. However, it requires a more exhaustive analysis. Are these “hot-spots” functional or simply represent heterogeneous hot-spot expression of DREADDs or SF-iGluSnFr? Authors should provide a spatial quantification of the expression of these elements. Moreover, are these hot-spots

reliably conserved upon successive stimulations? This is an important aspect of the study that needs a more precise analysis.

This point by Referee 3 is similar to point 4 by Referee 2. We agree that the issue of the release occurring at hot-spots is highly relevant as indicative of specific release sites, and the case should have been presented with deeper experimental and analytical support. This is, hopefully, amended in the current version of the manuscript, which incorporates new experiments, new analyses and a more exhaustive data presentation in the figures. We send the Referee to our detailed responses to points 4 and 5 of Referee 2. In synthesis, we now provide several additional experimental datasets in which we show responses to Gq-DREADD as well as to endogenous Gq-GPCRs stimulation, both *in situ* (P2Y1R in dentate gyrus) and *in vivo* (cholinergic receptors- likely muscarinic – in visual cortex) – see revised Fig. 2 and new Extended Data Figs. 4-7. In all cases, responses occur recurrently at specific hot-spots that overall occupy only a small fraction (10-20%) of the FOV containing a single astrocyte. Since spotty responses are elicited not only by Gq-DREADD stimulation but also by stimulation of endogenous receptors, i.e. in the absence of any artificial Gq-DREADD expression, we exclude that “spottiness” of the responses depends on the pattern of Gq-DREADD expression. Moreover, for each Gq-DREADD expressing cell that we stimulated, we now show the pattern of Gq-DREADD expression (mCherry fluorescence) that can be compared to the pattern of iGluSnFR responses (Fig. 2, panels c,k,w; Extended Data Fig. 4, panel h,n; extra examples in response to point 4 of Referee 2). The two never overlap, with the mCherry pattern occupying a portion of the FOV larger than that of the spotty SF-iGluSnFR responses. We also exclude that SF-iGluSnFR hot-spots depend on defined patterns of SF-iGluSnFR expression. Thus, for each individual cell *in situ* that we stimulated with either CNO or 2MeSADP, we show the corresponding response elicited by L-Glut application (Fig. 2, panels c,g; Extended Data Fig. 4, panels a,b; Extended Data Fig. 6a,e; extra examples in response to point 4 of Referee 2). In all cases, the L-Glut response occupies >70% of the FOV (Extended Data Fig. 4f). This response highlights all the sites in the given cell/FOV that express iGluSnFR and are potentially responsive to stimuli, thereby excluding that hot spots can be due to limited SF-iGluSnFR expression. Moreover, by adding Alexa 594 to the stimuli solutions and monitoring its fluorescence diffusion during puffs, we could exclude that hot spots are due to the incapacity of puffed agents to widely diffuse in the FOV (Supplemental Video 1). Finally, we now provide visual documentation in representative cells that SF-iGluSnFR responses to repeated Gq-GPCR stimulations occur in a spatially reliable manner. Specifically, for each FOV/cell, we now show pixel-by-pixel maps of all the individual responses to the consecutive receptor stimulations (6 responses in *in situ* experiments and 2 *in vivo*, Fig. 2, panels d,h for *in situ* responses, and panels u,x for the *in vivo* ones; Supplemental Video 1; extra examples in response to point 4 of Referee 2). Such representation allows visualizing the spatial pattern and intensity of each of the responses and their comparison. Moreover, the standard deviation map that we add shows in color code the signal variance across the 6 stimulations, highlighting areas of high variance that identify sites of repeated responses, i.e. hot spots (Fig. 2, panels c,g; Extended Data Fig. 4n; extra examples in response to point 4 of Referee 2). We believe that, taken together, all the information provided in the revised manuscript makes a compelling case for the genuine biological nature of the hot-spot responses, most likely signaling specific sites (regions) of glutamate release.

4. In the synaptic plasticity experiments (Fig 3e), LTP amplitude was indicated to be significantly lower in VGLUT1 KO. I wonder whether this an apparent observation or statistical significance. If there is statistical significance, it should be stated, and the statistical analysis needs to be reported.

The observed Θ -LTP reduction is statistically significant. We are sorry that this point was not sufficiently documented in the original manuscript. To address the Referee’s comment, we have added the

statistical symbol in Figs. 3e and f, as well as information about the significance level ($p < 0.05$) and the test used (two-tailed paired t test) in the results description (p. 11, lines 374-375) and in the figure legend. To note that, following Referee's 2 request (see response to point 8), we have reanalyzed the data and compared slopes (rather than amplitudes) of fEPSPs recorded in proximity of VGLUT1^{GFAP-KO} astrocytes with those of fEPSPs recorded in the domain of wildtype astrocytes. Therefore, the data in the revised version of Fig. 3e-f are from the slopes analysis.

5. Related to the previous point, when whole-cell recordings were performed, only 50% of the cells showed LTP reduction. This is claimed to be consistent with the patch-seq data of 50% of astrocytes expressing VGLUT1. This might be the case, but not necessarily. Furthermore, the 50-50 situation is not helpful. To demonstrate the actual cause-effect link additional experimental work should be provided. Otherwise, authors should refrain to establish such a weak association. The link, if it exists, between these two set of separate observations remains hypothetical.

We agree with the Referee's criticism. During revision, we have obtained new information useful to the topic such as the first percent counting (24%) and distribution of glutamatergic astrocytes in the dorsal DGML, the region where we performed the Θ -LTP experiments. Moreover, we see from the examples presented in Fig. 1g, that some of the Tom+ astrocytes from tamoxifen-treated *GFAP^{creERT2}tdTom^{ls/ls}* mice are not glutamatergic. These data are in line with the idea that part of Tom+ astrocytes will not affect LTP in their domain, but are still insufficient to establish a precise quantitative relation with the proportion of responders that we observed in the LTP experiments. Therefore, we decided to address the Referee's critique by removing the "50% responders issue" and its interpretation from the text, focusing only on the statistically proven data, the difference in mean LTP magnitude in the domain of VGLUT1^{GFAP-KO} astrocytes compared to wildtype astrocytes resulting from the analysis of all the tested pairs. The revised text describing and interpreting the Θ -LTP experiments is at pages 11-12, lines 370-382.

6. VGLUT1 dependence of glutamate release implies the vesicular release probably through a calcium dependent process. To accept the conclusions regarding the synaptic studies of VGLUT1 KO require basic control experiments that the upstream phenomena, like calcium excitability, has not been compromised in these transgenic mice. Authors should provide experimental evidence testing whether both the spontaneous and theta stimulation-evoked astrocyte calcium signal were affected in these mice.

We agree with this comment of the Referee and have performed the requested control experiments to confirm that suppression of glutamate release in VGLUT1^{GFAP-KO} mice is specifically due to deletion of the transporter and not to a generic impairment of the astrocyte signaling. We measured Ca²⁺ dynamics both under basal conditions/low level of stimulation, and in response to Θ -LTP protocols in the MPP, comparing responses in *Slc17a7^{fl/fl}* mice injected with either AAV5-*hGFAP-mCherry-iCre* virus (VGLUT1^{GFAP-KO}) or AAV5-*GFAP-mCherry* virus (controls). Despite the very different type of astrocyte Ca²⁺ signals observed under basal conditions and Θ -LTP stimulation, in neither of the two situations we observed evident differences between the KO and control groups, excluding unspecific signaling perturbation in the VGLUT1^{GFAP-KO} mice. The new data are shown in Extended Data Fig. 9g-l, and described in the text (page 11, lines 377-380), while experimental and analysis details are reported in figure legend and in the Methods section (experimental information: page 66, lines 1705-1713; analysis: page 70, lines 1861-1867).

7. Experiments regarding the effects of VGLUT1KO on kainate-induced epileptiform activity (Fig.3 j-n) are interesting. However, for experts on epilepsy, they could be seen as rather simple to naively conclude “these data reveal a “protective” role of astrocyte VGLUT1-dependent signalling against seizures *in vivo*”. To support such a strong conclusion authors should provide evidence using different experimental models of epilepsy, as well as a more detailed analysis of ictal discharges, interictal duration, etc.

We agree with the Referee that our former conclusion: “these data reveal a “protective” role of astrocyte VGLUT1-dependent signaling against seizures *in vivo*” sound too strong and general being supported by data obtained in a single model of experimental seizures. Therefore, we rephrased it bringing it down to its context (“these data reveal a “protective” role of astrocyte VGLUT1-dependent signalling against KA-induced acute seizures *in vivo*” (page 12, lines 411-414). We maintained emphasis on the protective role, because our main intention originally was to highlight the novelty of the observed protective effect and its potential therapeutic interest, rather than generalizing the relevance of our findings to human epilepsy. Indeed, our results were unexpected (counterintuitive for Referee 2 in her/his general comment) as they indicate that astrocyte VGLUT1-dependent signaling limits amplification of kainate-induced seizures *in vivo*, at odds with the pro-excitatory roles reported for astrocytes by most of the current literature (see “expectations” of Referee 2, point 9, and our response there). Of course, as indicated by Referee 3, deeper studies in multiple models are needed in order to consider the current findings of therapeutic relevance in human epilepsy, and this is in our opinion a subject for a follow-up publication. To respond to the more immediate request of the Referee, i.e., to strengthen our conclusions with a more detailed analysis of current data, we added two parameters, time first-to-last seizure and interictal duration. While the former was not changed in VGLUT1^{GFAP-KO} mice compared to controls, the latter was reduced, as expected by the more numerous and longer-lasting seizure episodes observed in the VGLUT1^{GFAP-KO} mice. The new results are shown in Fig. 3n,p and described in the text at page 12, lines 407-411.

8. The last part of the manuscript focuses on VGLUT2 in SN because it is claimed to be the main VGLUT in this area. Authors basically repeat what was done in the previous figures, but obviously the rigor was not as demanding as the other part of the ms (see the following comments). This change of the focus to a different brain area on a different VGLUT weakens the main message of the manuscript because it seems to be non-coherently articulated with other parts. Authors may consider to strength the main conclusion of the paper by showing the absence of effects of VGLUT1 KO in this brain area.

We understand the Referee’s reasoning on the last part of our study in SN and can agree to a large extent. Let us, however, reiterate what indicated in the response to point 1: consistency of the glutamatergic astrocyte phenotype does not depend on the consistent presence of a specific marker, VGLUT1 in this case, but on a signature of several differentially-expressed genes, including VGLUT2, that characterizes the cell population. In this perspective, switching studies from VGLUT1^{GFAP-KO} in hippocampus to VGLUT2^{GFAP-KO} in SN, does not change in our view the coherence of our investigation on glutamatergic astrocytes. Rather it tries to address a conceptually important issue, if this cell population is restricted to one brain region, or it plays larger roles in brain function. In our opinion, this issue should be addressed within the study that provides the first description of these cells. This said, we admit that in pursuing this conceptual goal we have enlarged the scope of the study, exposing it to appear somehow incoherent/not well articulated among its components. Beyond the scope issue, the Referee indicates that the SN part appeared less accurate than the others and was introduced too briskly. To amend these weaknesses, we have performed all the experiments proposed by Referee 3 in

this and following points. To start, we extended the initial electrophysiological experiments in SNpc DA neurons to VGLUT1^{GFAP-KO} mice. The results indicate that removing astrocyte VGLUT1 has no effect on sEPSC frequency, in contrast with the large effect seen upon removal of VGLUT2. As a corollary, this result provides a clearer rationale, supported by experimental data, for switching studies in SN from VGLUT1^{GFAP-KO} to VGLUT2^{GFAP-KO} mice, thereby hopefully helping the SN part look better connected to the preceding ones. The new data are shown in Extended Data Fig. 10f,g, and described in the text within a new narrative of the passage from VGLUT1 to VGLUT2 studies (page 13, lines 421-439).

9. Authors claim that the lower paired pulse ratio (PPR) in VGLUT2KO than in control mice (Fig. 4f) is “consistent with a pre-synaptic inhibition of the STN excitatory input induced by the astrocyte VGLUT2-dependent pathway”. This might be the case, but it is not necessary. If EPSCs are larger, the probability of release during PPR is expected to be decreased, i.e., no such presynaptic inhibition may exist.

10. Because “the group III mGluR agonist O-phospho-L-serine (L-SOP) did not modify sEPSC frequency in control mice”. Authors conclude that a presynaptic inhibitory mechanism endogenously active that occludes L-SOP effects. Again, this is a likely explanation, but it should be supported by direct experimental evidence, e.g., testing the effects of mGluR antagonists.

Points 9 and 10 of the Referee focus on our interpretation of the changes observed in ePSCs/PPR (point 9) and sEPSCs (point 10) in VGLUT2^{GFAP-KO} mice with respect to controls. The two points are conceptually linked and are answered here together. Thus, the Referee expresses in both cases the concern that our interpretation of the results is not the only one possible and indicates that we should perform additional experiments to validate the hypothesis that astrocyte VGLUT2-dependent signaling acts as endogenous presynaptic inhibitory mechanism of the excitatory input to SNpc DA neurons via activation of group III mGluRs. We performed such experiments, and compared the effects of a gr. III mGluR agonist (L-SOP) and a gr. III mGluR antagonist (MSOP) on both sEPSCs and ePSCs/PPR in VGLUT2^{GFAP-KO} mice vs related controls. If pre-synaptic gr. III mGluRs are the endogenous target of astrocyte VGLUT2-dependent glutamate release, responses to mGluR III drugs should be different in control and KO mice. In particular, agonist responses should be observed in KO mice (putatively lacking the endogenous VGLUT2-dependent activation of group III mGluRs) and occluded in control mice; for antagonist responses, the opposite should be true. The experimental results that we obtained are consistent with the predictions and we think that they significantly strengthen the evidence in support of our original conclusion that VGLUT2-dependent astrocyte signaling is a physiological endogenous activator of gr. III mGluRs inducing presynaptic inhibition of the excitatory input to SN DA neurons. The new data are presented in Fig. 4 d-f, described in the text at p. 13-14, lines 448-462, with experimental detail and statistical analysis reported in the figure legend.

11. Authors finally aimed to test the astrocyte control on nigrostriatal circuit function by examining the in vivo levels of dopamine (DA) in the dorsal striatum. Authors found that basal levels in control and VGLUT2KO mice were not significantly different, which contradicts the main claim of the authors, i.e., that astrocytic VGLUT2-dependent glutamate release affects nigrostriatal circuit function. Authors should make sense of this contradiction. On the other hand, and in contrast, they found that amphetamine-evoked increase of DA levels was enhanced in VGLUT2KO. This a very interesting observation, but it lacks sufficient experimental test to provide a clear mechanistic interpretation. As honestly indicated by the authors, this is “consistent with loss in these mice of presynaptic inhibition of SNpc DA neurons (and possibly of additional astrocyte controls in the dST)”, but this interpretation remains hypothetical and requires further studies to characterize the underlying mechanisms.

In our understanding, there are two linked points in this comment by the Referee: (1) Authors should make sense of the contradiction that basal DA levels in control and VGLUT2^{GFAP-KO} mice were not significantly different, which contradicts their main claim, i.e., that astrocytic VGLUT2-dependent glutamate release affects nigrostriatal circuit function. (2) Given the first issue, authors should perform further studies to understand by which mechanism amphetamine affects DA levels in astrocyte VGLUT2^{GFAP-KO} mice. Concerning the first point, we wrote in the original manuscript: “Basal levels in the two groups were not significantly different, *despite a trend to be higher in the eluates from VGLUT2^{GFAP-KO} mice*”. The trend to higher levels in VGLUT2^{GFAP-KO} mice was visible in each of the three time-points of the basal measures (see inset to Fig. 4i in the original manuscript). Stimulated by the Referee’s comment, we looked carefully at this dataset, and realized that we did a mistake in our previous statistical analysis. Assuming the data distribution was normal for all groups and time-points, we utilized parametric tests (repeated measures two-ways ANOVA followed by Fisher’s test). In fact, by checking with *ad hoc* tests (Shapiro-Wilk and Chen-Shapiro), the data turned out to not respect the normal distribution, implying the need of non-parametric statistics for their analysis. Thus, we reanalyzed the dataset combining Friedman ANOVA, the non-parametric equivalent to repeated measures ANOVA, with post-hoc Wilcoxon signed rank test (for comparing time-points within each genotype) or Kolmogorov-Smirnov test (for comparisons between the two genotypes). This analysis provided different results, most notably basal DA levels in VGLUT2^{GFAP-KO} resulted significantly higher than in VGLUT2^{GFAP-WT} mice at each of the 3 sampled points ($p = 0.048, 0.039$ and 0.012 for Basal 1, Basal 2 and Basal 3, taken 60, 40 and 20 min before amphetamine administration, respectively). We are very sorry for the mistake that led us to an incorrect (and detrimental) statement on our results, which in turn triggered the negative comment and request of clarifications by the Referee. We have now corrected the statistical information: in the figure (now Figure 4h), legend, in the statistics methods section (p. 79, lines 2150-2151 and 2157-2160) and in the text (p. 14, lines 465-466). In view of the relevance of these data, and to give them better visibility, we have regrouped the 3 time-points and presented them in a separate panel (Fig. 4h) as histogram of mean basal DA levels in VGLUT2^{GFAP-KO} vs VGLUT2^{GFAP-WT} mice, rather than showing them as part of the time-course of the amphetamine effect as done in the original manuscript. The second point raised by the Referee stems from the first one and should be largely resolved by the above revised results, which let now logically interpret the amphetamine effect on DA levels in VGLUT2^{GFAP-KO} mice as potentiation of the effect already seen in basal condition in these mice, in line with the known mode of this drug action on the dopamine system. Nonetheless, we agree with the Referee that we cannot firmly conclude that relief of presynaptic inhibition of SN DA neurons is the only mechanism underlying the strong potentiation of DA release induced by amphetamine in VGLUT2^{GFAP-KO} mice, as we already indicated in the original manuscript. Survey of the current literature provided hints of possible additional primary or secondary contributory pathways in dST (e.g., Adermark et al. Neuropsychopharmacol., 2021), but no defined mechanistic hypothesis that we could easily test within the current study, already somehow criticized by both Referee 2 and 3 for its very broad scope. Rather, we see the need for a wide range of investigations, which in our opinion should be the subject of a dedicated future study.

Minor comments

1. Page 9, line 209. It is stated that the essay was done “few weeks later”. This vague information should be explicitly stated in more concrete terms.

The time window used for these experiments is “6-8 weeks after viral injection”, and was stated in the Methods section of the original manuscript but not in the main text. We have now substituted the vague information in the main text with the specific information (page 6, line 212).

Reviewer Reports on the First Revision:

Referees' comments:

Referee #1 (Remarks to the Author):

The authors have addressed all of my original concerns. In particular, I commend them for conducting the targeted Patch-seq experiments, despite encountering some technical difficulties. Although the number of cells they were able to test was not high, their results support the hypotheses linking glutamate release to specific transcriptomic cell types. Additionally, they have enhanced their statistical analysis concerning the overlap between the different transcriptomic datasets used for the analysis.

Referee #3 (Remarks to the Author):

Present manuscript is a revised version of a manuscript previously submitted.

As indicated in the assessment of the previous version, "This is an interesting study that adds valuable information regarding a relevant topic in current neuroscience, the regulatory role of astrocytes of brain function through gliotransmission. Conceptually, the idea of glutamatergic transmission is not especially novel but has been a largely debated. This manuscript provides further and solid support for it with elegant experiments and cutting-edge techniques. The manuscript is also innovative regarding the functional aspects of the glutamatergic gliotransmission." I also indicated "the results presented are novel and relevant, and of broad interest for the scientific community."

I also indicated: "The study is methodologically and technically adequate, having combined the use of multiple state-of-the-art techniques to provide elegant results, which are, in general, very convincing. The main conclusions reached are, in general, well-supported by the experimental data." However, I expressed some specific concerns that authors have addressed in this revised version. Below is the evaluation of authors' reply as well as the general assessment of the revised version of the manuscript.

Evaluation of authors reply to the specific major comments:

1. Authors have provided a reasonable explanation that addresses my original concern.
2. The question still remains, but I agree with the authors that the suggested experiments are very demanding and that the question is somehow beyond the scope of the present study. I m satisfied with the response and the lack of the further information gained with the experiments does not compromise the main conclusions of the manuscript.
3. The new experiments and analysis now provided have adequately addressed my previous concern and support the author's conclusions.

4. Authors have adequately addressed the concern.

5. Authors have accepted the criticism, removing the concerning issue of the 50%. The issue has been clarified without compromising the major conclusions.

6. New control data presented in Extended Data Fig. 9g-l suggest that the upstream astrocyte calcium signal was unaffected. The reported data is purely qualitative, and a more exhaustive quantitative analysis would have been desirable. Potential quantitative differences could not likely account for the observed effects. Therefore, I found the reported results sufficiently convincing.

7. Authors have tone down the initial strong conclusion of the protective role of astrocyte VGLUT1-dependent signaling against seizures. The additional analysis presented now has strengthen the results. Authors responded to the suggested deeper studies on epilepsy as a subject for a follow-up publication, which seems reasonable. Nevertheless, current data have a value as a starting initial description. In any case, I believe authors should acknowledge this point with a short sentence. Altogether, the authors' responses to these concerns are considered satisfactory.

8. Authors have adequately addressed the concern with well-reasoned explanation and new data shown in Extended Data Fig. 10f,g.

9 and 10. Authors have adequately addressed the concerns providing concncing new data and analysis.

11. First, this reviewer appreciates the re-analysis performed by the authors to address the concern, which has been now adequately clarified. The reasoning provided by the authors about the potential mechanistic interpretation as satisfactory. Moreover, the statement in the text "Independent of the specific mechanism(s), these data revealed that astrocyte VGLUT2-dependent signaling regulates nigrostriatal DA pathway function in vivo" is considered very fair and balanced.

In summary, I believe authors have adequately addressed my specific concerns about the previous version of the manuscript, providing reasonable justifications and adding novel and convincing experimental data and analysis. The manuscript has been therefore substantially improved.

In conclusion, I confirm my previous general assessment of the manuscript: "This is an interesting study that adds valuable information regarding a relevant topic in current neuroscience, the regulatory role of astrocytes of brain function through gliotransmission" and "The manuscript includes a large number of interesting observations that provide interesting implications for several aspects and topics of the astrocyte-neuron interaction". Such positive initial assessment of the manuscript is now further enhanced about the revised version.

I have therefore not further comments and I recommend the publication of the present manuscript.

Alfonso Araque

Author Rebuttals to First Revision:

Point-by-point response to Referees

First we want to thank all the Referees for their final positive evaluation of our study and for the constructive comments during the review process that have certainly led us to improve the quality of our manuscript.

In blue our responses to the residual points:

Referee #1 (Remarks to the Author):

The authors have addressed all of my original concerns. In particular, I commend them for conducting the targeted Patch-seq experiments, despite encountering some technical difficulties. Although the number of cells they were able to test was not high, their results support the hypotheses linking glutamate release to specific transcriptomic cell types. Additionally, they have enhanced their statistical analysis concerning the overlap between the different transcriptomic datasets used for the analysis.

We thank Referee 1 for her/his positive evaluation of the additional experiments that we performed on her/his suggestions. We do not see residual responses to give.

Referee #3 (Remarks to the Author):

Present manuscript is a revised version of a manuscript previously submitted.

As indicated in the assessment of the previous version, “This is an interesting study that adds valuable information regarding a relevant topic in current neuroscience, the regulatory role of astrocytes of brain function through gliotransmission. Conceptually, the idea of glutamatergic transmission is not especially novel but has been a largely debated. This manuscript provides further and solid support for it with elegant experiments and cutting-edge techniques. The manuscript is also innovative regarding the functional aspects of the glutamatergic gliotransmission.” I also indicated “the results presented are novel and relevant, and of broad interest for the scientific community.”

I also indicated: “The study is methodologically and technically adequate, having combined the use of multiple state-of-the-art techniques to provide elegant results, which are, in general, very convincing. The main conclusions reached are, in general, well-supported by the experimental data.” However, I expressed some specific concerns that authors have addressed in this revised version. Below is the evaluation of authors’ reply as well as the general assessment of the revised version of the manuscript.

We thank the Referee for reiterating a positive general evaluation of our study and for indicating that we have addressed in the revised manuscript her/his specific concerns.

Evaluation of authors reply to the specific major comments:

1. Authors have provided a reasonable explanation that addresses my original concern.
2. The question still remains, but I agree with the authors that the suggested experiments are very demanding and that the question is somehow beyond the scope of the present study. I m satisfied

with the response and the lack of the further information gained with the experiments does not compromise the main conclusions of the manuscript.

We thank the Referee for understanding our point.

3. The new experiments and analysis now provided have adequately addressed my previous concern and support the author's conclusions.

4. Authors have adequately addressed the concern.

5. Authors have accepted the criticism, removing the concerning issue of the 50%. The issue has been clarified without compromising the major conclusions.

6. New control data presented in Extended Data Fig. 9g-l suggest that the upstream astrocyte calcium signal was unaffected. The reported data is purely qualitative, and a more exhaustive quantitative analysis would have been desirable. Potential quantitative differences could not likely account for the observed effects. Therefore, I found the reported results sufficiently convincing.

We agree with the Referee that the data presented in Fig. 9g-l is mostly qualitative, although we repeated the same observation more than once. We never saw in VGLUT1^{GFAP-KO} a dramatic reduction of the Ca²⁺ signaling evoked by Gq-DREADD stimulation in astrocytes. Being the Referee's concern that this upstream effect could explain the suppression of glutamate release observed in this line, we think that our experiments anyway address her/his concern.

7. Authors have toned down the initial strong conclusion of the protective role of astrocyte VGLUT1-dependent signaling against seizures. The additional analysis presented now has strengthened the results. Authors responded to the suggested deeper studies on epilepsy as a subject for a follow-up publication, which seems reasonable. Nevertheless, current data have a value as a starting initial description. In any case, I believe authors should acknowledge this point with a short sentence. Altogether, the authors' responses to these concerns are considered satisfactory.

We took the point of the Referee and added the requested sentence to the present main text: "More specifically, they reveal a "protective" function of astrocyte VGLUT1-dependent signalling against KA-induced acute seizures *in vivo*, notably opposing the mechanism/s causing seizure amplification. ***This function is worth exploring further in chronic epilepsy models for possible therapeutic perspectives***".

8. Authors have adequately addressed the concern with well-reasoned explanation and new data shown in Extended Data Fig. 10f,g.

9 and 10. Authors have adequately addressed the concerns providing concerning new data and analysis.

11. First, this reviewer appreciates the re-analysis performed by the authors to address the concern, which has been now adequately clarified. The reasoning provided by the authors about the potential mechanistic interpretation as satisfactory. Moreover, the statement in the text "Independent of the specific mechanism(s), these data revealed that astrocyte

VGLUT2-dependent signaling regulates nigrostriatal DA pathway function in vivo” is considered very fair and balanced.

We thank the Referee for appreciation of the analysis performed and the statement in the text

In summary, I believe authors have adequately addressed my specific concerns about the previous version of the manuscript, providing reasonable justifications and adding novel and convincing experimental data and analysis. The manuscript has been therefore substantially improved.

In conclusion, I confirm my previous general assessment of the manuscript: “This is an interesting study that adds valuable information regarding a relevant topic in current neuroscience, the regulatory role of astrocytes of brain function through gliotransmission” and “The manuscript includes a large number of interesting observations that provide interesting implications for several aspects and topics of the astrocyte-neuron interaction”. Such positive initial assessment of the manuscript is now further enhanced about the revised version.

I have therefore not further comments and I recommend the publication of the present manuscript.

We thank the Referee for thorough evaluation of our study with constructive indications, and for the final positive evaluation.